# 3D mapping of compositional gradients of core-shell structures in AgIn$_x$Ga$_{1-x}$S$_2$ quantum dots by atom probe tomography

Byeong-Gyu Chae[1,4], Mihye Lim[2,3,4], Junho Lee [2], Nayoun Won[2], Soo Kyung Kwon[2], Ara Jo[2], Dong Jin Yun [1], Sangjun Lee [1], Jwa-Min Nam [3] ✉, Soohwan Sul [1] ✉ & Tae-Gon Kim [2] ✉

Colloidal quantum dots (QDs), which exhibit tunable band gaps depending on their size and composition, are widely studied for light-emitting and optoelectronic applications. AgIn$_x$Ga$_{1-x}$S$_2$-based QDs are particularly promising due to their pure green emission, high blue absorption, and environmental friendliness. However, a comprehensive understanding of these quaternary QDs remains challenging because of the difficulty in examining their complex compositional structure. Here, we three-dimensionally characterize quaternary QDs (AgIn$_x$Ga$_{1-x}$S$_2$-based heterostructured QDs with core/shell and core/shell/shell structures) on the atomic scale using atom probe tomography. We reveal that both the AgIn$_x$Ga$_{1-x}$S$_2$/AgGaS$_2$ QDs with and without an outer ZnS shell have compositional gradients at their interfaces and elemental inhomogeneity among their cores. Furthermore, an Ag-deficient AgIn$_y$Ga$_{1-y}$S$_2$ layer is identified on the outer surface of the AgGaS$_2$ shell, where the stoichiometric fractions satisfy $x \gg y$, arising from differences in the precursor reactivity. Meanwhile, in the AgIn$_x$Ga$_{1-x}$S$_2$/AgGaS$_2$/ZnS QDs, the outer ZnS shell evolves into Zn$_{1-3/2x}$Ga$_x$S through a cation exchange process, ensuring structural and chemical compatibility with the inner shell. Our findings uncover the internal architecture and nanoscale elemental distributions of quaternary QDs, providing guidance for the future development of QDs.

Over the past several decades, colloidal semiconductor quantum dots (QDs) have garnered significant attention owing to their versatility in applications, including optoelectronic and biomedical devices[1–11]. QDs are particularly attractive because of their high photoluminescence (PL) performance, size-tunable optoelectronic characteristics, and cost-effectiveness[1–14]. The introduction of a core/shell structure, where the core is encapsulated by a shell, stabilizes QDs and increases their PL efficiency[1,6,9–11,14–17]. Recently, multinary I–III–VI-based QDs, such as AgInS$_2$ and CuInS$_2$, have emerged as promising environmentally

friendly alternatives to conventional Cd- or In-based QDs for next-generation applications because their high compositional flexibility enables band-gap tuning[1,7,14–16,18–24]. Notably, the incorporation of Ga atoms into AgInS$_2$ cores has produced green-emitting AgIn$_x$Ga$_{1-x}$S$_2$-based QDs with enhanced color purity and strong blue absorption[18–24].

Despite these advantages, AgIn$_x$Ga$_{1-x}$S$_2$-based QDs still face critical challenges, including an inherently lower PL quantum yield (QY) and poor photochemical stability[20,21,24]. These issues may stem from factors such as the internal structure, compositional distribution,

[1]Analytical Engineering Group, Samsung Advanced Institute of Technology, Samsung Electronics Co., Ltd., Suwon, Republic of Korea. [2]Display Solution Platform, Samsung Advanced Institute of Technology, Samsung Electronics Co., Ltd., Suwon, Republic of Korea. [3]Department of Chemistry, Seoul National University, Seoul, Republic of Korea. [4]These authors contributed equally: Byeong-Gyu Chae, Mihye Lim. ✉e-mail: jmnam@snu.ac.kr; soohwan.sul@samsung.com; taegon2.kim@samsung.com

defects, elemental diffusion, and residual impurities. However, little is known about the atomistic mechanisms, such as the incorporation and diffusion of elements within $AgIn_xGa_{1-x}S_2$-based QDs. Several key questions remain unanswered: Is the QD structure a core/shell configuration? What is the three-dimensional (3D) compositional distribution within each layer? How do elements diffuse in heterostructured QDs? Are the interfaces and layers compositionally discrete or gradual? How do these elemental distributions affect the properties of QDs? What happens during shell growth on these complex structures?

Although the research on nanometer-sized core/shell QDs has advanced significantly through techniques such as transmission electron microscopy (TEM), X-ray photoelectron spectroscopy (XPS), and Fourier-transform infrared spectroscopy, these questions remain unresolved[25–30]. In particular, directly mapping the 3D distribution of constituent elements is a significant challenge. This difficulty might be due to the restricted spatial resolution and sensitivity of these instruments, as well as their inherent inability to analyze the 3D distribution in such complex structures. Atom probe tomography (APT) provides a potential solution by enabling direct 3D atomic imaging with a spatial resolution on the sub-nanometer scale[31–35]. In APT, an electric or laser pulse directed at a needle-shaped specimen ($< 100$ nm apex) causes individual ions to evaporate and travel to a position-sensitive detector, where their time-of-flight yields the mass-to-charge ratio, enabling 3D elemental mapping[31–35]. However, applying APT to nanostructures, especially colloidal QDs, remains technically challenging because of their porosity, which complicates the preparation of needle-shaped specimens and often leads to structural instability.

In this study, we employ APT analysis to achieve the direct 3D mapping of two types of quaternary $AgIn_xGa_{1-x}S_2$-based heterostructured QDs: (1) $AgIn_xGa_{1-x}S_2$ core/$AgGaS_2$ shell QDs and (2) $AgIn_xGa_{1-x}S_2$ core/$AgGaS_2$ inner shell/ZnS outer shell QDs. The direct 3D observation of individual atoms enables us to uncover the complex internal structure of these QDs. We reveal that both types of QDs exhibit a compositional gradient across the layers and interfaces, rather than discrete boundaries. For the $AgIn_xGa_{1-x}S_2$/$AgGaS_2$ QDs, we identify the formation of an Ag-deficient shell on the surface of the $AgGaS_2$ shell during synthesis because of differences in the precursor reactivity. This Ag-deficient shell may sufficiently confine charge carriers while maintaining the crystalline structure. Interestingly, during the ZnS precursor step to form an outer shell on the $AgIn_xGa_{1-x}S_2$/$AgGaS_2$ QDs, a $Zn_{1-3/2x}Ga_xS$ outer shell spontaneously forms instead, driven by $Ag^+$ and $Ga^{3+}$ to $Zn^{2+}$ cation exchange. This $Zn_{1-3/2x}Ga_xS$ shell exhibits structural and chemical compatibility with the $AgGaS_2$ inner shell and is expected to induce a Type-I band alignment. Additionally, APT analysis directly demonstrates noticeable compositional inhomogeneity among different $AgIn_xGa_{1-x}S_2$ cores, with off-stoichiometry in the core material. In addition to the effects of the size distribution, this nonuniform composition may also contribute to the inhomogeneous spectral broadening of the ultraviolet–visible (UV–Vis) absorption and PL band. These findings highlight the potential of APT to deepen our understanding of complex nanostructures and guide improvements in their performance.

## Results and discussion
### Structural and optical characteristics
Figure 1a, d schematically illustrates the internal structure, band diagram, UV–Vis absorption, and PL characteristics of the $AgIn_xGa_{1-x}S_2$/$AgGaS_2$ and $AgIn_xGa_{1-x}S_2$/$AgGaS_2$/ZnS QDs, respectively. The $AgIn_xGa_{1-x}S_2$/$AgGaS_2$ QDs were synthesized by modifying a method described in refs. 18–20. Both types of QDs exhibit bright green luminescence, with emission peaks at ≈530 nm. Notably, the QY increases from 85% to 92% after the ZnS precursor step. Furthermore, the stability of QD–acrylate composite films under ambient conditions was evaluated in terms of the change in the QY after 48 h with respect to the initial QY. As shown

in Supplementary Fig. 1, the ZnS shell significantly improves this normalized QY from 66% to 97%. The enhanced optical properties can be attributed to the effective confinement of electrons and holes provided by the wide-bandgap ZnS encapsulating the $AgIn_xGa_{1-x}S_2$/$AgGaS_2$. In addition to the dominant green emission, the $AgIn_xGa_{1-x}S_2$/$AgGaS_2$/ZnS QDs exhibit a long-wavelength emission tail in the 600–700 nm range (Fig. 1d). The area ratio of this long-wavelength emission, defined as the fraction of the total PL intensity in the long-wavelength region ($\lambda > \lambda_{PL\ max} + 50$ nm), increases from ≈3.7% in $AgIn_xGa_{1-x}S_2$/$AgGaS_2$ QDs to ≈14.9% in $AgIn_xGa_{1-x}S_2$/$AgGaS_2$/ZnS QDs. Such long-wavelength emission is generally attributed to defect- or trap-assisted recombination rather than band-edge excitonic transitions in $AgInS_2$-based QDs and related I–III–VI₂-based QDs[36–40].

The synthesized $AgIn_xGa_{1-x}S_2$/$AgGaS_2$ core/shell and $AgIn_xGa_{1-x}S_2$/$AgGaS_2$/ZnS core/shell/shell structured QDs were characterized using high-angle annular dark-field scanning transmission electron microscopy (HAADF–STEM). Figure 1b, e displays the HAADF–STEM images of the $AgIn_xGa_{1-x}S_2$/$AgGaS_2$ QDs before and after the formation of the ZnS outer shell, respectively. The ZnS shell, which facilitates the formation of Type-I structures and improves the PL QY and stability, was synthesized through thermal decomposition[1,4,5,11,15,21]. The $AgIn_xGa_{1-x}S_2$/$AgGaS_2$ QDs primarily exhibit a round-cornered polyhedral or spherical shape, with an average diameter of 6.14 ± 0.76 nm. Interestingly, after the ZnS precursor step for the outer shell formation, the QDs grow only negligibly, despite the addition of more precursors, and remain similar in both size and shape, with an average diameter of 6.11 ± 1.03 nm (Supplementary Fig. 2; see also Supplementary Fig. 3 for HAADF–STEM images of the core-only $AgIn_xGa_{1-x}S_2$ QDs). Moreover, the high-resolution STEM results reveal that the QDs are structurally similar before and after the ZnS precursor step, with all crystal planes showing a lattice spacing of ≈0.29 nm, which almost corresponds to the interplanar spacing of the (200) planes of chalcopyrite-structured $AgGaS_2$. The lattice fringes are clearly evident not only inside the $AgIn_xGa_{1-x}S_2$-based QDs but also at the edges, indicating the high crystallinity across the entire QD structure (Fig. 1b, e). X-ray diffraction (XRD) measurements were further performed on the core-only $AgIn_xGa_{1-x}S_2$, $AgIn_xGa_{1-x}S_2$/$AgGaS_2$, and $AgIn_xGa_{1-x}S_2$/$AgGaS_2$/ZnS QDs to corroborate the structural similarity before and after shell formation. All XRD patterns are consistent with a chalcopyrite-based tetragonal structure, and the overall diffraction features remain essentially unchanged, even after the Zn precursor step (Supplementary Fig. 4). This comparative analysis indicates that the crystal structure is preserved throughout the shell formation process.

Meanwhile, HAADF–STEM imaging did not provide sufficient contrast to clearly distinguish the core from the shell. This limitation arises from the lack of significant mass differences between the constituent elements, as well as the similar crystallinity extending from the core to the shell. Specifically, for $AgIn_xGa_{1-x}S_2$-based QDs ($x ≈ 0.5$) synthesized for green emission in this study, the limited Z-contrast makes the core visually indistinguishable from the shell. The STEM–energy-dispersive X-ray spectroscopy (STEM–EDS) mapping showed that Zn atoms are distributed over a broader region than Ag and In after the Zn precursor step (Fig. 1c, f and Supplementary Fig. 5). However, the current STEM results remain insufficient for resolving the detailed internal and local compositional information of such 3D nanostructures. To identify the factors affecting the optical properties of the QDs and to further improve their performance, their internal structure must be more comprehensively understood.

### 3D compositional distribution of $AgIn_xGa_{1-x}S_2$/$AgGaS_2$ QDs
To characterize and quantify the 3D atomic distribution of the $AgIn_xGa_{1-x}S_2$-based core/shell structured QDs, APT was employed. For the APT analysis in this work, dried QDs were prepared in powder form and analyzed following a method of ref. 41. In that study, we demonstrated that APT could resolve core/shell architectures and 3D

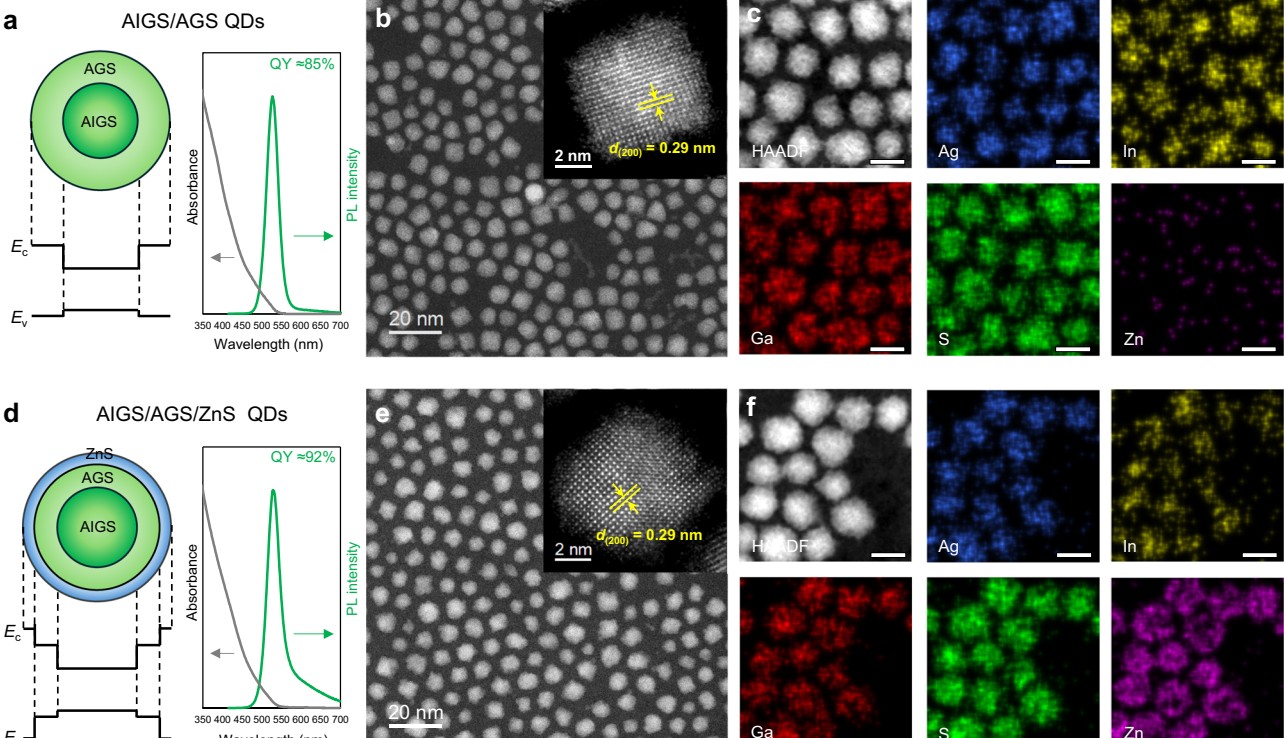

**Fig. 1 | Comparison of AgIn$_x$Ga$_{1-x}$S$_2$-based QDs before and after the ZnS precursor step. a**, **d** Schematic representation of the internal structure, band diagrams, and UV–vis absorption, and PL characteristics of AgIn$_x$Ga$_{1-x}$S$_2$/AgGaS$_2$ (AIGS/AGS) and AgIn$_x$Ga$_{1-x}$S$_2$/AgGaS$_2$/ZnS (AIGS/AGS/ZnS) QDs. Both QDs exhibit bright green luminescence at ≈530 nm, with the QY increasing from 85% to 92% after the ZnS precursor step to form the outer shell. In addition, AgIn$_x$Ga$_{1-x}$S$_2$/AgGaS$_2$/ZnS QDs exhibit a long-wavelength emission tail in the ≈600–700 nm range. $E_c$ and $E_v$ denote the conduction band minimum and valence band maximum, respectively. **b**, **e** HAADF–STEM image of AgIn$_x$Ga$_{1-x}$S$_2$-based QDs before and after the ZnS precursor step, which negligibly changed the size of the QDs, with average diameters of 6.14 ± 0.76 and 6.11 ± 1.03 nm, respectively. Size analysis was performed on more than 300 randomly selected QDs ($n > 300$). Inset: high-magnification HAADF–STEM image. Similar lattice fringes were consistently observed across multiple particles. **c**, **f** STEM–EDS elemental maps of AgIn$_x$Ga$_{1-x}$S$_2$-based QDs before and after the ZnS precursor step. Colors: Ag, blue; In, yellow; Ga, red; S, green; Zn, violet. (Scale bars: 7 nm in (**c**, **f**)).

compositional distributions in commercially available CdSe/ZnS core/shell QDs, in good agreement with TEM observations. Specimen preparation strategies, potential APT artifacts, and reconstruction considerations specific to CdSe/ZnS core/shell QDs were systematically examined, thereby establishing the applicability of APT to colloidal QD nanostructures. More broadly, the feasibility of APT in the analysis of metallic nanoparticles has been demonstrated in numerous prior studies, including ligand-resolved analyses and combined experiment–simulation approaches[42–47]. On this basis, we report a 3D atomic-scale characterization of AgIn$_x$Ga$_{1-x}$S$_2$-based QDs using APT, providing detailed insight into their internal compositional structure.

Figure 2a shows the 3D reconstructed APT atom maps of the AgIn$_x$Ga$_{1-x}$S$_2$/AgGaS$_2$ QDs, representing a hemispherical APT specimen containing five QDs (see Supplementary Fig. 6 for dashed guides indicating the approximate outlines of individual QDs). The QDs appear as slightly ellipsoidal features in the APT reconstruction, and a slice-view image along the $x$–$z$ plane is presented in Fig. 2b to visualize their internal elemental distribution. We note that APT analyses of nanostructures may exhibit reconstruction artifacts, such as apparent elongation along the analysis ($z$) direction and interfacial broadening, primarily arising from local magnification and trajectory aberrations[42–52]. Accordingly, the slightly ellipsoidal morphology observed here is attributed to such effects rather than the intrinsic particle shape. However, compositional distributions along the $z$-direction have been shown to provide more reliable quantitative information than lateral directions because the reduced trajectory overlap suppresses artificial intermixing[42–44,51,52]. Under these

conditions, compositional distributions and interfacial characteristics closer to the intrinsic distributions can be more evaluated.

APT shows that the detected Ga atoms are more broadly distributed than the Ag and In atoms in the QDs, and S atoms appear in both the core and shell regions of the QDs. Because the QDs are closely packed in a non-collinear configuration within the APT specimen and may therefore partially overlap in projection views, 3D reconstructions with C- and H-related ions removed are also provided in Supplementary Fig. 7 to better visualize the individual QDs. The 3D concentration profile obtained from an iso-concentration surface (proxigram) for Ag atoms reveals the average compositional trends across the five QDs (Fig. 2c). This profile more clearly indicates a higher concentration of Ag and In atoms toward the QD core, whereas Ga atoms increase toward the outer shell, confirming the formation of an AgIn$_x$Ga$_{1-x}$S$_2$ core/AgGaS$_2$ shell heterostructure.

To more precisely analyze the atomic-scale elemental distribution within individual QDs, two representative AgIn$_x$Ga$_{1-x}$S$_2$/AgGaS$_2$ QDs were extracted from the 3D reconstruction map. Figure 3a, d shows the 3D atom maps and the cross-sectional regions cropped by the green boxes for each QD. The cross-sectional maps of individual elements are presented in Fig. 3b, e, providing detailed insights into the internal structure. The 2D compositional profiles along two different directions (Fig. 3c, f) further confirm the formation of the AgIn$_x$Ga$_{1-x}$S$_2$ core/AgGaS$_2$ shell heterostructure with a compositional gradient. From the extracted profiles, we primarily focus on $z$-direction compositional analyses, cross-validated by proxigrams and lateral profiles. Ag and In atoms are highly concentrated at the center of the QD, at more than 20

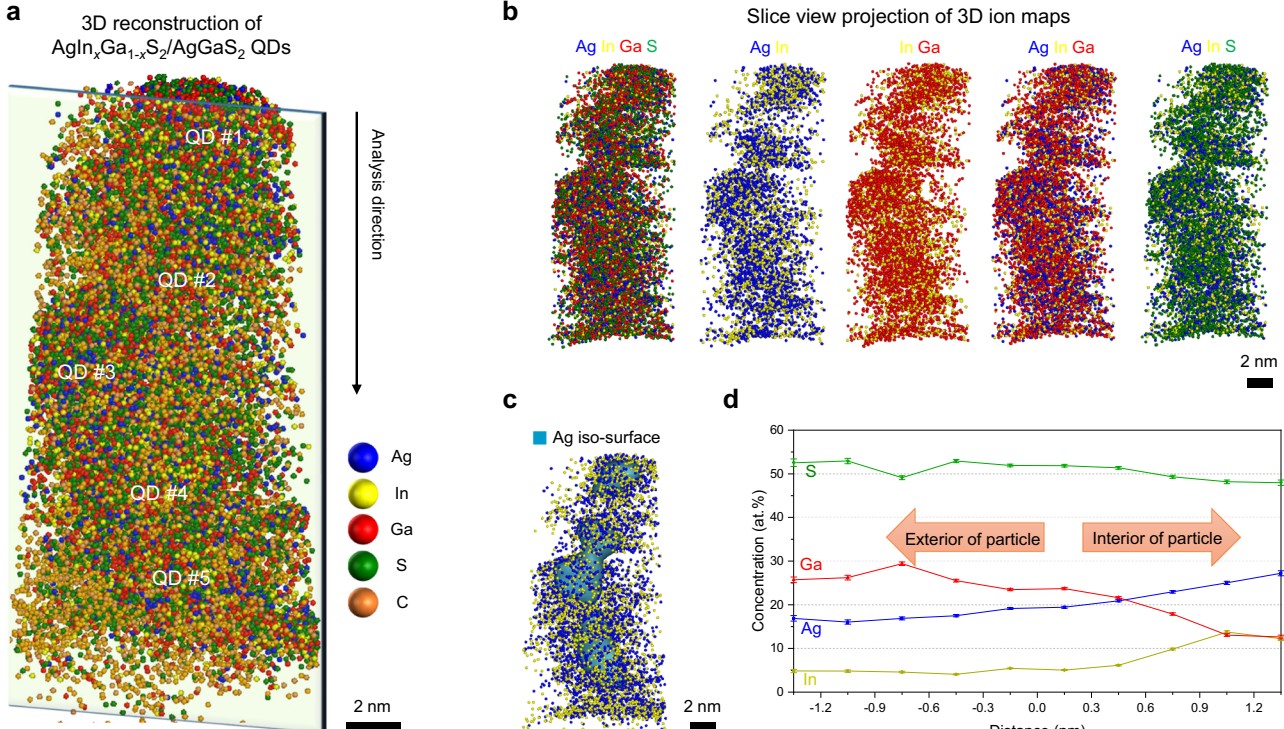

**Fig. 2 | 3D reconstruction and compositional analysis of AgIn$_x$Ga$_{1-x}$S$_2$/AgGaS$_2$ QDs using APT. a** Reconstructed 3D atom map of the entire analyzed volume, showing five slightly ellipsoidal QDs. **b** Slice-view atom maps of Ag, In, Ga, and S, illustrating their internal spatial distributions, shown as 3 nm-thick slices. Ga and S atoms are more broadly distributed than Ag and In. **c** Ag iso-surface map and **d** proxigram showing 3D compositional trends; the proxigram was obtained from five QDs. The Ag and In contents increase toward the QD core, while the Ga content increases toward its outer surface, confirming the core/shell heterostructure. Colors: Ag, blue; In, yellow; Ga, red; S, green; C, orange. (Scale bars: 2 nm in (**a**–**c**)) Error bars in (**d**) indicate one-sigma counting statistics based on the detected ion counts within each sampling bin. The analysis is based on a single dataset without independent replicate measurements. Source data are provided as a Source Data file.

and 10 at.%, respectively, but their concentrations gradually decrease toward the shell. Conversely, Ga atoms are less concentrated in the QD center but significantly more present in the outer shell. In addition, trace amounts of In atoms remain present in the shell region, leading to a shell composition closer to AgIn$_y$Ga$_{1-y}$S$_2$, where $x$ denotes the fraction of In in the core and $y$ represents the residual In fraction in the outer layer, with $x \gg y$, rather than a pure AgGaS$_2$ shell. This is likely due to the outward diffusion of In atoms from the AgIn$_x$Ga$_{1-x}$S$_2$ core to vacant sites in AgGaS$_2$, driven by the concentration gradient or thermal budget during synthesis. Nevertheless, the resulting AgIn$_y$Ga$_{1-y}$S$_2$ shell is expected to effectively confine charges within the AgIn$_x$Ga$_{1-x}$S$_2$ core owing to its higher band gap, consistent with a Type-I heterostructure, which promotes radiative recombination.

Another notable feature of the AgIn$_x$Ga$_{1-x}$S$_2$/AgGaS$_2$ QDs revealed by the APT analysis is the Ag-deficient outermost shell on the AgGaS$_2$ shell, which, strictly speaking, corresponds to the AgIn$_y$Ga$_{1-y}$S$_2$ shell. Comparing the cross-sectional atomic maps of Ag and Ga (Fig. 3b, e), Ga shows a broader distribution than Ag. The 2D compositional profiles also indicate a higher concentration of Ga atoms in the outermost region of the QDs, although distinguishing individual layers is challenging because of the compositional gradient (Fig. 3c, f). The unexpected formation of the Ag-deficient AgIn$_y$Ga$_{1-y}$S$_2$ outermost shell is associated with the relatively lower reactivity of Ga$^{3+}$ compared with that of Ag$^+$ during the AgGaS$_2$ shell synthesis. The lower reactivity between S$^{2-}$ and group 13 elements, such as Ga$^{3+}$, compared with that of group 11 elements, e.g., Ag$^+$, has been observed when synthesizing group 11, 13, and 16 semiconductor QDs; hence, the reaction conditions must be precisely controlled to achieve the desired shell formation[53]. To compensate for the limited incorporation of Ga caused by its lower reactivity, we introduced an excess of Ga precursors

during shell growth. Given that Ag$^+$ is more reactive than Ga$^{3+}$, Ag preferentially reacts with S during the initial stage of AgGaS$_2$ shell growth. The excess Ga, which may not be readily incorporated during the early stage of the reaction, contributes to forming the Ag-deficient AgIn$_y$Ga$_{1-y}$S$_2$ shell at a later stage. The resulting Ag-deficient outermost shell is expected to effectively passivate surface defects without introducing additional defect levels into the band gap, despite its Ag deficiency, thereby preserving the crystalline structure and stabilizing the QDs.

To demonstrate the role of differential precursor reactivity, we performed controlled syntheses in which the Ag precursor amount was increased at a fixed Ga level, a condition under which increasing the Ag precursor amount directly promotes the further growth of the AgGaS$_2$ shell. STEM−EELS analyses showed the pronounced thickening of the AgGaS$_2$ shell and consistently increased Ag incorporation, supporting the role of precursor reactivity in the formation of the shell (Supplementary Fig. 8 and Supplementary Table 1). These changes were accompanied by an increased contribution from trap-related emission and a decreased PL QY, indicating that Ag-driven shell growth directly impacts the optical properties of the QDs.

**3D compositional distribution of AgIn$_x$Ga$_{1-x}$S$_2$/AgGaS$_2$/ZnS QDs**
Following the ZnS precursor step to form the outermost shell, APT was conducted to investigate the compositional evolution of AgIn$_x$Ga$_{1-x}$S$_2$/ AgGaS$_2$/ZnS QDs. Figure 4a shows a reconstructed APT 3D atom map containing five nearly spherical AgIn$_x$Ga$_{1-x}$S$_2$/AgGaS$_2$/ZnS QDs (see Supplementary Fig. 9 for dashed guides indicating the approximate outlines of individual QDs). To further clarify the presence of these five QDs within the reconstructed volume and their internal structures, slice-view images along the $x$−$z$ planes are provided in Fig. 4b.

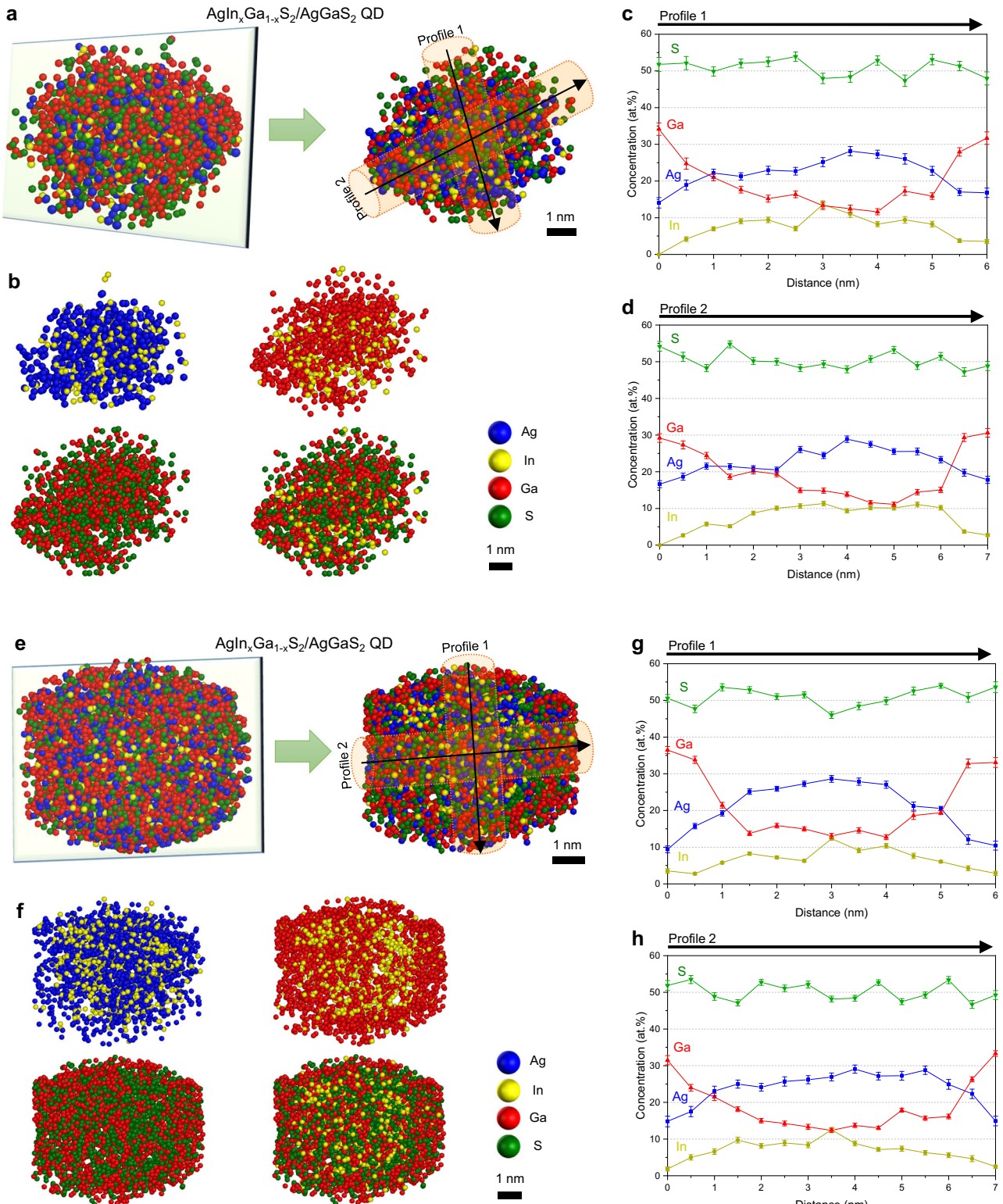

**Fig. 3 | Compositional distributions of individual AgIn$_x$Ga$_{1-x}$S$_2$/AgGaS$_2$ QDs.**
**a**, **e** 3D atom map and cross-sectional atom map cropped by the green box.
**b**, **f** Cross-sectional atom maps showing the spatial distribution of each element. Ga and S are broadly distributed, while Ag and In are present in small amounts in the shell regions. Colors: Ag, blue; In, yellow; Ga, red; S, green; C, orange. **c**, **d**, **g**, **h** 2D compositional profiles confirming the AgIn$_x$Ga$_{1-x}$S$_2$ core/AgGaS$_2$ shell structure and

a compositional gradient. An Ag-deficient AgIn$_y$Ga$_{1-y}$S$_2$ layer appears in the outermost region. (Scale bars: 1 nm in (**a**, **b**, **e**, **f**)) Error bars in (**c**, **d**, **g**, **h**) indicate one-sigma counting statistics based on the detected ion counts within each sampling bin. The analysis is based on a single dataset without independent replicate measurements. Source data are provided as a Source Data file.

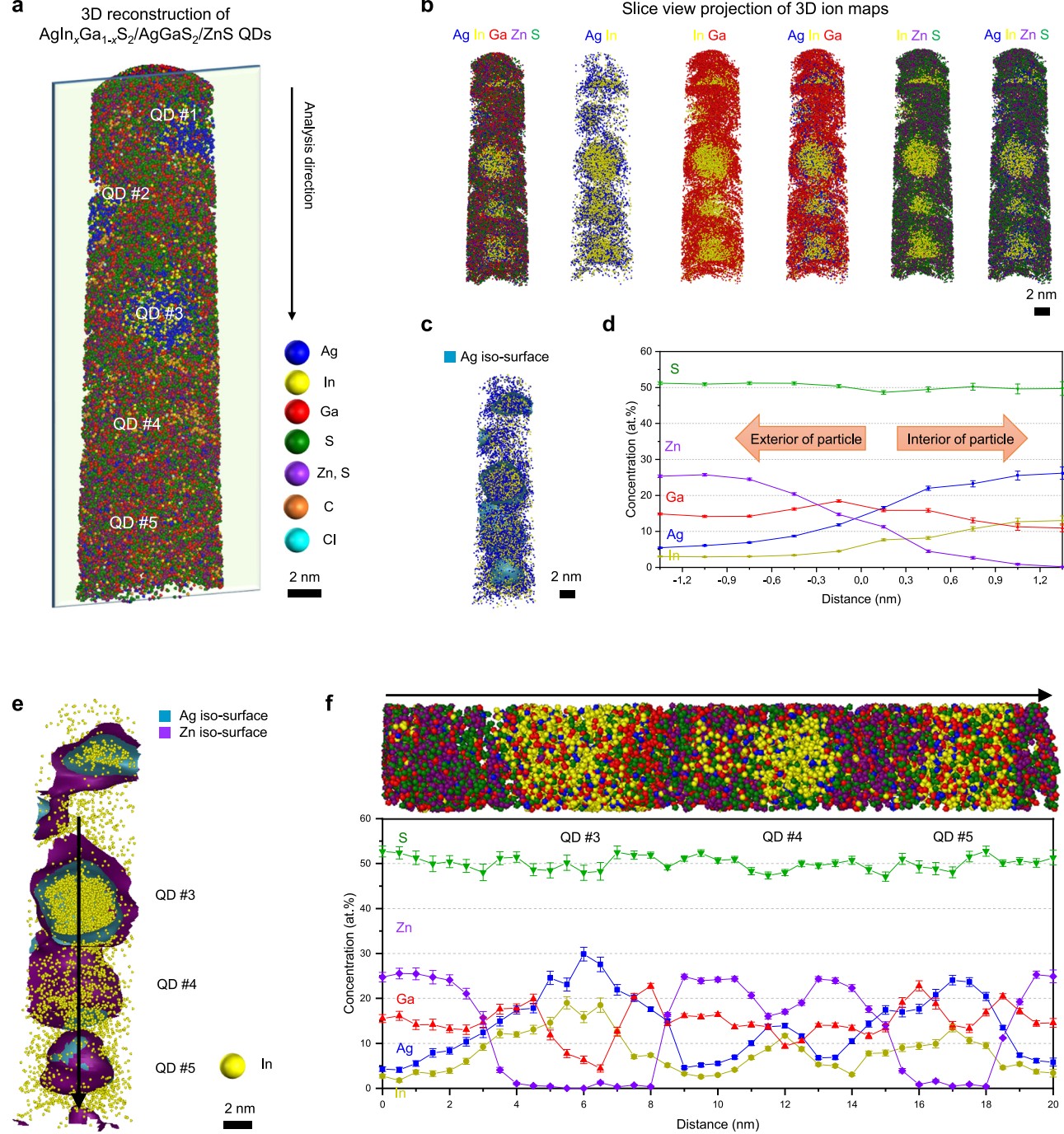

**Fig. 4 | 3D reconstruction and compositional analysis of AgInₓGa₁₋ₓS₂/AgGaS₂/ZnS QDs using APT. a** Reconstructed 3D atom map of the entire analyzed volume, showing five slightly ellipsoidal QDs. **b** Slice-view atom maps of Ag, In, Ga, Zn, and S, illustrating their internal spatial distributions, shown as 3 nm-thick slices. Although most Ag and In atoms are in the core regions, a significant amount of Ga is distributed over the shell region. Zn atoms predominantly exist in the outer shell. **c** Ag iso-surface and **d** proxigram obtained from Ag iso-surfaces; the proxigram was obtained from five QDs. The Ag and In contents increase toward the QD core, while the content of Zn increases toward the outer shell of the QD. A significant amount of Ga also appears in the outer shell. **e** Ag and Zn iso-surface and **f** 2D compositional profile extracted from a cylindrical volume ($3 \times 3 \times 21 \, \text{nm}^3$) through the 3D reconstruction, confirming the core/shell/shell structure. Colors: Ag, blue; In, yellow; Ga, red; S, green; Zn, violet; C, orange; Cl, cyan. (Scale bars: 2 nm in (**a**–**c**, **e**)). The slight variations in the compositional distributions from QD to QD arise from the spatial offsets of individual QDs within the APT specimen. Error bars in (**d**, **f**) indicate one-sigma counting statistics based on the detected ion counts within each sampling bin. The analysis is based on a single dataset without independent replicate measurements. Source data are provided as a Source Data file.

Although the 3D distribution of individual elements varies slightly among QDs depending on the cross-section position, all QDs consistently exhibit the AgInₓGa₁₋ₓS₂ core/AgGaS₂ inner shell/Zn₁₋₃/₂ₓGaₓS outer shell structure, as shown in Fig. 4b. Further, the 3D atom maps show that Zn atoms, primarily located in the shell regions, are more broadly distributed than Ag and In. In APT, some Zn and S isotopes overlap in the mass spectrum. However, distinct peaks, such as S⁺, allow imaging with minimal interference from Zn isotopes (Supplementary Fig. 10). Interestingly, the Ga atoms, which are expected to reside primarily in the core and inner shell, are more broadly distributed throughout the QD than initially anticipated (Supplementary Fig. 11).

The proxigram from an iso-concentration surface of Ag atoms, which describes the five QDs, reveals that In atoms are more narrowly distributed than Ag atoms (Fig. 4c and Supplementary Fig. 12). Zn atoms are predominantly present in the outer shell, as evidenced by their increasing concentration toward the exterior. Ga atoms are primarily abundant in the AgGaS$_2$ inner shell but are also present in significant amounts within the ZnS outer shell. The 2D concentration profile along the analysis direction offers additional insights into the internal structure of each QD (Fig. 4d). Although the elemental concentrations vary slightly depending on the profiling path owing to the spatial offset of individual QDs in the APT specimen, the 2D profile confirms the significant presence of Ga atoms in the outer shell, albeit at lower concentrations than Zn atoms. C-related signals, likely originating from ligands, were also observed on the outside of the QDs (Supplementary Fig. 13). Nevertheless, because of the difficulty in quantifying these signals, our discussion focuses on the distribution of the main elements (see Supplementary Fig. 14 for the 3D reconstructions with the C- and H-related ions removed for clarity). Notably, this work provides the first 3D atomic-scale insights into heterostructured quaternary QDs with a core/shell/shell geometry using APT.

To validate the internal structure and elemental distribution of AgIn$_x$Ga$_{1-x}$S$_2$/AgGaS$_2$/ZnS QDs, two representative QDs were extracted from the 3D reconstruction map (Fig. 5). Figure 5a, d shows the extracted 3D atom maps and the cross-sectional regions cropped by the green boxes for each QD. According to the z-direction compositional profiles cross-validated by proxigrams and lateral profiles, Ag atoms are more widely distributed than In atoms in the 3D atom maps (Fig. 5b, e). However, both Ag and In atoms are at their highest concentrations in the central region of the QDs, with values exceeding 20 and 10 at.%, respectively. The concentration of In atoms remains lower than that of Ag atoms outside the core, substantiating the formation of an inner shell closer to AgIn$_y$Ga$_{1-y}$S$_2$ or AgGaS$_2$ (Fig. 5c,f). Similar to those of the AgIn$_x$Ga$_{1-x}$S$_2$/AgGaS$_2$ QDs, the shell regions of the AgIn$_x$Ga$_{1-x}$S$_2$/AgGaS$_2$/ZnS QDs also exhibit trace amounts of In atoms, likely due to diffusion during synthesis. Zn atoms show the highest concentrations in the outer shell surrounding the AgIn$_x$Ga$_{1-x}$S$_2$/AgGaS$_2$ core/inner shell. Interestingly, the compositional distributions of Ag, In, and Ga are gradual rather than discrete across the layers. From a structural perspective, this compositional gradient is expected to offer advantages such as minimizing lattice mismatch between the core and inner shell and effectively reducing the likelihood of internal defect formation. Such structural features—often referred to as compositional gradient shells or gradient alloy shells—have been widely reported in InP- and CdSe-based QDs to contribute to enhancing both the QY and stability[54–58]. As observed in the reconstructed maps and proxigram, Ga atoms are also present not only in the AgIn$_x$Ga$_{1-x}$S$_2$ core and AgGaS$_2$ inner shell, but also in the ZnS outer shell, thus forming a Zn$_{1-3/2x}$Ga$_x$S shell (Fig. 5). The Ga content is highest in the AgGaS$_2$ inner shell and lower than the Zn content in the outer shell. Because APT inherently probes only a limited number of individual QDs and may therefore suffer from limited statistical representativeness, the compositional features identified by APT were further examined using complementary techniques. XPS analyses performed on the same samples reveal consistently higher Ga-to-Ag ratios after the Zn precursor step, evaluated from multiple core-level regions, including Ag MNN Auger transition (where MNN denotes an Auger transition involving the M and N shells) vs. Ga 2p and Ag 4d vs. Ga 3d. This systematic increase in the Ga-to-Ag ratios supports the formation of Ga-containing outer shells and is consistent with the Zn$_{1-3/2x}$Ga$_x$S shell formation revealed by APT (Supplementary Fig. 15 and Supplementary Table 2). Inductively coupled plasma analyses performed on the same samples show overall elemental compositions consistent with the APT results (Supplementary Tables 3 and 4). Taken together, these results indicate that the structural and compositional features identified by

APT are not limited to a small number of individual QDs but are representative of overall sample.

Interestingly, directly comparing two individual QDs reveals compositional differences, even among co-synthesized QDs. These differences are particularly pronounced in the core region, with noticeable variations in the Ag, In, and Ga contents; some QDs exhibit cores with a very low Ga content (Figs. 4d and 5c). Similar results were observed in the AgIn$_x$Ga$_{1-x}$S$_2$/AgGaS$_2$ QDs, albeit to a lesser extent. The pronounced inhomogeneity in the cores likely originates from the intrinsic complexity of the quaternary Ag–In–Ga–S system. The synthesis of the core simultaneously involves multiple cation precursors with different reactivities, and the cores are formed under a continuous heating process. Previous studies have also shown that AgIn$_x$Ga$_{1-x}$S$_2$ core formation proceeds through intermediate phases and cation exchange processes rather than a simple single-step growth process[18,53,59]. More specifically, AgGaS$_2$ or AgInS$_2$ seeds form initially, followed by the incorporation of In and Ga to generate the chalcopyrite AgIn$_x$Ga$_{1-x}$S$_2$ lattice, which is likely to promote local compositional fluctuations within the core. Furthermore, the subsequent ZnS precursor step introduces an additional thermal treatment, which can promote cation diffusion and redistribution, thereby amplifying pre-existing compositional fluctuations within the cores.

Consistent with the APT observations, STEM–EDS line profiles collected from multiple QDs reveal noticeable particle-to-particle variations in the relative Ag, In, and Ga distributions (Supplementary Figs. 16 and 17). For each QD, several independent line profiles were extracted and compared, all of which show dot-to-dot compositional fluctuations. These results indicate that the observed compositional inhomogeneity is an intrinsic feature of the synthesized quaternary QDs, thereby supporting the features revealed by APT. A deviation from the ideal stoichiometry in the core can change the emission wavelengths. Thus, such compositional inhomogeneity in the quaternary semiconductor cores may contribute to the inhomogeneous spectral broadening of the UV–Vis absorption and PL bands for both QDs, along with other factors such as the size distribution. Our APT analysis directly shows compositional inhomogeneity among the cores and highlights the importance of controlling such variations, along with defects, to improve the color purity of the green emission.

Our analytical findings have thus far proved that the outer shell in AgIn$_x$Ga$_{1-x}$S$_2$/AgGaS$_2$/ZnS QDs is indeed a Zn$_{1-3/2x}$Ga$_x$S shell rather than a ZnS shell. The unintended formation of this Zn$_{1-3/2x}$Ga$_x$S shell can be ascribed to cation exchange between Zn$^{2+}$ and both Ag$^+$ and Ga$^{3+}$, with Zn$^{2+}$ ions partially replacing both Ag$^+$ and Ga$^{3+}$ ions in the AgGaS$_2$ shell during the ZnS precursor step (Supplementary Fig. 18). During the initial stages of this process, Zn$^{2+}$ ions approach the AgGaS$_2$ surface and replace Ag$^+$ and Ga$^{3+}$ ions, which occupy tetrahedral coordination sites in the AgGaS$_2$ lattice. These two ions are progressively substituted by Zn$^{2+}$ ions while maintaining charge neutrality, leading to a gradual transition in the composition of the outer shell into Zn$_{1-3/2x}$Ga$_x$S. This process forms a compositionally mixed Zn$_{1-3/2x}$Ga$_x$S outer shell with a chalcopyrite structure, which is the overall crystallographic framework of the AgGaS$_2$ inner shell[18,53,60–65]. Fortuitously, the formation of Zn$_{1-3/2x}$Ga$_x$S with a chalcopyrite-like structure, which is the same as that of AgGaS$_2$, is crystallographically preferable over zincblende-structured ZnS. This structural compatibility between Zn$_{1-3/2x}$Ga$_x$S and AgGaS$_2$ is expected to minimize lattice mismatch, thus reducing lattice distortion and residual strain. Furthermore, this explains why the STEM–HAADF analysis revealed that the overall size of the QDs remained nearly unchanged after the ZnS precursor step, while their crystallinity was preserved throughout the QDs, thus verifying the formation of the Zn$_{1-3/2x}$Ga$_x$S shell, which is structurally similar to ZnGaS$_2$, via cation exchange. The emergence of the Zn$_{1-3/2x}$Ga$_x$S outer shell via cation exchange offers a plausible origin for the long-wavelength emission tail observed after the ZnS precursor step. In our system, the substitution of Ag$^+$ and Ga$^{3+}$ by Zn$^{2+}$ during shell formation

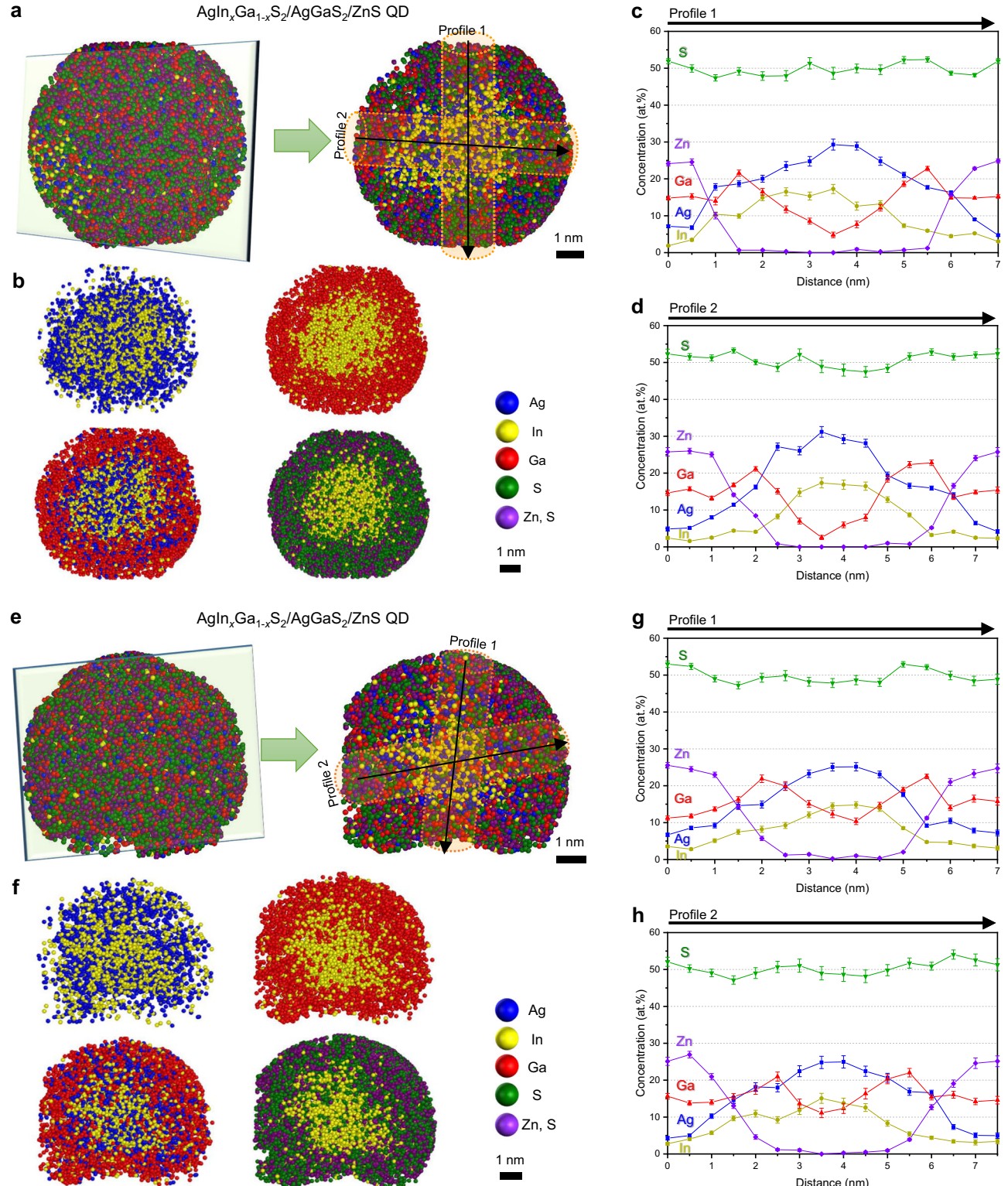

**Fig. 5 | Compositional distributions of individual AgIn$_x$Ga$_{1-x}$S$_2$/AgGaS$_2$/ ZnS QDs. a**, **e** 3D atom map and cross-sectional atom map cropped by the green box. **b**, **f** Cross-sectional atom maps showing the spatial distribution of each element. Zn atoms are more broadly distributed than Ag and In. Ga atoms exhibit a broad distribution across the QDs, extending beyond the inner shell. Colors: Ag, blue; In, yellow; Ga, red; S, green; Zn, violet. **c**, **d**, **g**, **h** 2D compositional profiles confirming the AgIn$_x$Ga$_{1-x}$S$_2$ core/AgGaS$_2$ inner shell/ZnS outer shell structure with

a gradual compositional gradient. Ga is observed in all layers, including the outer shell, which thus comprises Zn$_{1-3/2x}$Ga$_x$S rather than ZnS. Compositional differences between individual QDs, particularly in the core region, are also evident. (Scale bars: 1 nm in (**a**, **b**, **e**, **f**)) Error bars in (**c**, **d**, **g**, **h**) indicate one-sigma counting statistics based on the detected ion counts within each sampling bin. The analysis is based on a single dataset without independent replicate measurements. Source data are provided as a Source Data file.

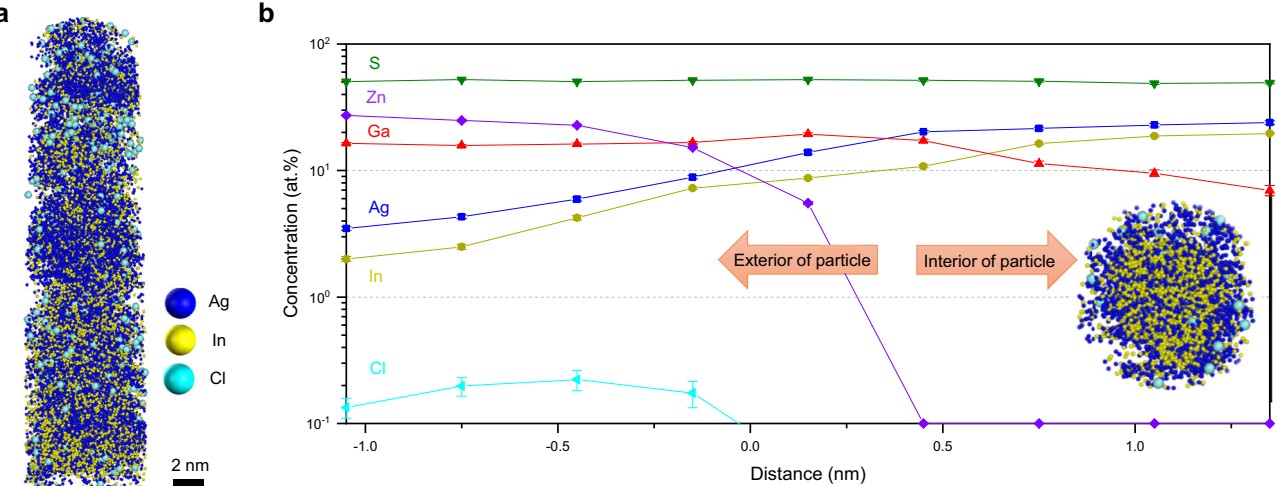

**Fig. 6 | Cl residues in AgIn$_x$Ga$_{1-x}$S$_2$/AgGaS$_2$/ZnS QDs. a** Cl atoms distribution in 3D atom map and **b** proxigram obtained from Ag iso-surface. Cl atoms, originating from the ZnCl$_2$ precursor, are detected in trace amounts, primarily in the outermost part of the Zn$_{1-3/2x}$Ga$_x$S shell. Inset: Cross-sectional 3D atom map. Colors: Ag, blue; In, yellow; Ga, red; S, green; Zn, violet; Cl, cyan. Scale bars: 2 nm in (**a**). Error bars in (**b**) indicate one-sigma counting statistics based on the detected ion counts within each sampling bin. The analysis is based on a single dataset without independent replicate measurements. Source data are provided as a Source Data file.

inherently introduces cation vacancies and local charge-compensating defects. Consistent with our APT observations, the inward diffusion of Zn$^{2+}$ into the interior of the QDs further induces cation exchange and promotes the formation of such defects. When these defects form in the vicinity of In-rich regions, where carriers are preferentially confined, they can introduce intragap trap states that locally perturb the confinement potential and promote trap-assisted recombination. These defect-related states are commonly associated with donor–acceptor-pair-like recombination pathways that give rise to long-wavelength emission in I–III–VI$_2$ semiconductors[23,38–40,66–68].

In terms of the band gap energy, the Zn$_{1-3/2x}$Ga$_x$S shell typically exhibits a slightly lower band gap than ZnS ($\approx 3.5$ eV), even at the nanoscale[62,65,69–72]. The formation of the compositionally mixed Zn$_{1-3/2x}$Ga$_x$S layer is therefore expected to adequately preserve the quantum efficiency, with negligible weakening of charge confinement, consistent with Type-I-like band alignment. Moreover, the spontaneous formation of the Zn$_{1-3/2x}$Ga$_x$S shell could significantly reduce crystallographic or interfacial defects that might arise from the structural differences between the inner and outer shells. By creating a more seamless interface with the AgGaS$_2$ inner shell, the Zn$_{1-3/2x}$Ga$_x$S shell may enhance the stability of the QDs under ambient conditions and contribute to a more stable and efficient green luminescence. This optical stability is particularly advantageous for optoelectronic applications where long-term emission reliability is critical.

## Trace detection of residues in AgIn$_x$Ga$_{1-x}$S$_2$/AgGaS$_2$/ZnS QDs

To investigate the possibility of trace elements remaining in the nanoscale QDs, we evaluated the distribution of Cl atoms. As shown in Fig. 6 and Supplementary Fig. 19, Cl atoms are detected only in the AgIn$_x$Ga$_{1-x}$S$_2$/AgGaS$_2$/ZnS QDs and are localized in the outer shell region in only trace amounts. The Cl atoms, derived from the ZnCl$_2$ precursor used in the ZnS precursor step, show a strong spatial correlation with Zn atoms in the APT analysis, whereas no comparable correlation is observed with Ag, In, or Ga atoms. XPS measurement further confirms that Cl is absent in AgIn$_x$Ga$_{1-x}$S$_2$ cores and nearly absent in AgIn$_x$Ga$_{1-x}$S$_2$/AgGaS$_2$ QDs, whereas the Cl signal is pronounced in the AgIn$_x$Ga$_{1-x}$S$_2$/AgGaS$_2$/ZnS QDs, exhibiting metal–Cl chemical states attributable to Zn–Cl-related species (Supplementary Fig. 20 and Supplementary Table 2). These Cl species can be both detrimental and beneficial to the AgIn$_x$Ga$_{1-x}$S$_2$/AgGaS$_2$/ZnS QDs. As for

the detrimental effects, residual Cl may introduce lattice defects and surface trap states, which can serve as non-radiative recombination pathways, ultimately reducing the QY and optical stability of the QDs[73,74]. Conversely, in the absence of such detrimental effects, Cl can play a beneficial role by acting as a surface passivation agent or inorganic ligand[75–77]. Given these opposing effects, the incorporation of residual elements such as Cl must be carefully controlled by monitoring both their amounts and spatial distribution. Achieving an optimal balance in the incorporation of residues can maximize their passivating and stabilizing effects while minimizing potential adverse impacts on QD performance.

Although our study only demonstrated the 3D distribution of trace elements remaining after synthesis, APT is expected to be readily applicable to a broad range of nanostructures, particularly those with complex heterostructures. Such applications will enable detailed elucidation of the spatial distribution, segregation behavior, and interfacial incorporation of trace elements or dopants, thereby providing critical insights and design guidelines for the structural and electronic optimization of advanced nanomaterials.

In this study, we elucidated the internal structure and 3D elemental distribution of AgIn$_x$Ga$_{1-x}$S$_2$-based QDs, which have a heterostructure in core/shell geometry, via APT analysis. Both AgIn$_x$Ga$_{1-x}$S$_2$/AgGaS$_2$ and AgIn$_x$Ga$_{1-x}$S$_2$/AgGaS$_2$/ZnS QDs exhibited compositional gradients across layers and interfaces rather than discrete boundaries. Such gradient interfaces minimize lattice mismatch between adjacent layers, thereby reducing potential internal defects and enabling the realization of highly luminescent AgIn$_x$Ga$_{1-x}$S$_2$-based QDs. For AgIn$_x$Ga$_{1-x}$S$_2$/AgGaS$_2$ core/shell structured QDs, we identified an Ag-deficient shell on the surface of the AgGaS$_2$ shell, which could be attributed to the differences in the precursor reactivity during synthesis. For the AgIn$_x$Ga$_{1-x}$S$_2$/AgGaS$_2$/ZnS core/shell/shell structured QDs, we revealed the presence of compositional inhomogeneity among AgIn$_x$Ga$_{1-x}$S$_2$ cores. Additionally, a Zn$_{1-3/2x}$Ga$_x$S outer shell was found to form spontaneously through cation exchange during the ZnS precursor step for outer shell formation, and this shell effectively confined charge carriers within the core with its Type-I band alignment. These findings provide insights into the atomic-scale incorporation and diffusion behavior in QDs, offering guidance for optimizing the design and performance for advanced optoelectronic applications.

## Methods

### Materials

Gallium(III) acetylacetonate (Ga(acac)$_3$, 99.99%, Sigma-Aldrich), gallium(III) chloride (GaCl$_3$, 98%, Allegra), silver acetate (AgOAc, 99.99% trace metal basis, Sigma-Aldrich), 1-octadecene (ODE, 90%, Sigma-Aldrich), sulfur powder (99.98%, Sigma-Aldrich), n-trioctylphosphine (TOP, 90%, Strem), oleylamine (OAm, ≥98%, Allgre), toluene (anhydrous, 99.8%, Sigma-Aldrich), N,N'-dimethylthiourea (DMTU, 97% Tokche), indium acetate (In(OAc)$_3$, 99.99%, Sigma-Aldrich), and ethanol (99.5%, Samchun Chemical) were used as purchased.

### Precursor preparation

A 0.06 M Ag/OAm stock solution was prepared by dissolving AgOAc (6 mmol) in 100 mL of OAm, and the mixture was evacuated at 50 °C. In addition, a 1 M S/OAm stock solution was prepared by dissolving S powder (0.1 mol) in 100 mL of OAm under vacuum. Finally, 1 M GaCl$_3$/TOP and 0.5 M ZnCl$_2$/TOP stock solutions were prepared by separately dissolving GaCl$_3$ (0.1 mol) and ZnCl$_2$ (0.05 mol) powders in 100 mL of TOP, respectively.

### Synthesis of the AgIn$_x$Ga$_{1-x}$S cores and AgIn$_x$Ga$_{1-x}$S$_2$/AgGaS$_2$ QDs

AgIn$_x$Ga$_{1-x}$S$_2$-core QDs were synthesized by injecting a S precursor into a mixture of Ag, In, and Ga precursors, following a procedure similar to methods reported in refs. [18–20]. First, AgOAc, In(OAc)$_3$, and Ga(acac)$_3$ were mixed in 100 mL of OAm in a reaction flask, and the mixture was evacuated at room temperature for 10 min. The flask was then filled with N$_2$ gas, and 40 mL of the 1 M S/OAm stock solution was swiftly injected into the flask. Next, the flask was heated to 240 °C, and the temperature was maintained until the desired particle size was achieved, while monitoring the reaction progress using UV–Vis absorption and PL spectrometry. The resulting AgIn$_x$Ga$_{1-x}$S cores were centrifuged with acetone and re-dispersed in toluene. The molar ratio of Ga to In precursors was adjusted between 1.0 and 1.4 to tune the PL peak wavelength to ≈530 nm.

Subsequently, to coat the cores with the AgGaS$_2$ shell, a flask containing $2.3 \times g$ of DMTU and 200 mL of an OAm/ODE (1:1) solvent mixture was evacuated at 120 °C for 10 min. Then, it was filled with N$_2$ gas and heated to 260 °C. When it reached this temperature, the 1 M GaCl$_3$/TOP solution, the 0.06 M Ag/OAm solution, and the AgIn$_x$Ga$_{1-x}$S core QDs in toluene (≈5 mL) were injected into the flask, and the temperature was maintained for 100 min. The final AgIn$_x$Ga$_{1-x}$S$_2$/AgGaS$_2$ QDs were centrifuged at $3760 \times g$ with the addition of excess ethanol and re-dispersed in toluene for subsequent coating with ZnS. The Ag/Ga and Ga/S precursor ratios were 0.04 and 0.8, respectively.

### Synthesis of the AgIn$_x$Ga$_{1-x}$S$_2$/AgGaS$_2$/ZnS QDs

First, $0.25 \times g$ of DMTU and 80 mL of OAm were loaded into a reaction flask, and the mixture was evacuated at 120 °C and heated to 210 °C under N$_2$ flow. Then, the 0.5 M ZnCl$_2$/TOP solution and the AgIn$_x$Ga$_{1-x}$S$_2$/AgGaS$_2$ core/shell QDs in toluene (≈10 mL) were quickly injected into the flask, and the temperature was maintained for 100 min. The Zn and S precursors were injected in equal molar amounts.

### Optical property characterization

UV–Vis and PL measurements were conducted with a UV-2600 (Shimadzu) and F-7100 (HITACHI), respectively. The absolute PL QYs of the QDs dispersed in toluene and QD–acrylate composite films were measured using a quantum efficiency measurement system (Otsuka, QE-2100) equipped with an integrating half sphere with excitation at 450 nm.

### STEM observations

Colloidal AgIn$_x$Ga$_{1-x}$S$_2$/AgGaS$_2$ and AgIn$_x$Ga$_{1-x}$S$_2$/AgGaS$_2$/ZnS QDs were dispersed in hexane and deposited on TEM grids with a carbon support film (Tedpella, P/N 01824). Spherical aberration (Cs)-corrected STEM (FEI-Titan Cubed) was performed in the HAADF mode at 300 kV. High-resolution STEM images were acquired using an 80 pA current probe. STEM–EDS elemental maps were obtained using a 130 pA current probe over a total acquisition time of 10 min. The average particle size was determined by measuring the diameters of more than 300 randomly selected QDs from HAADF–STEM images using ImageJ 1.53 software.

### APT specimen preparation

Needle-shaped specimens for APT analysis were prepared from dried QDs in powder form using the focused ion beam (FIB) lift-out method (Helios5 HX, Thermo Fisher Scientific). Initially, a 100 nm-thick Pt layer was deposited using an electron beam, followed by depositing a 1 μm-thick Pt layer with a Ga-ion beam to passivate a region of interest measuring $12 \times 1.7$ μm on the sample surface. The target region was then extracted and mounted onto a sharpened W tip. The specimens on the W tip were further shaped into a needle-like geometry using an annular milling pattern (30 kV, 80 pA). To refine the samples and reduce damage induced by the Ga-ion beam, the region of interest was thinned again with an annular pattern at 5 kV and 8 pA. The morphology and dispersion of the QDs prepared by this procedure were confirmed by HAADF–STEM (Supplementary Fig. 21).

### APT analysis

APT analysis was conducted using a CAMECA atom probe (Invizo 6000) operating in the deep-UV laser mode ($\lambda = 257.5$ nm) with a pulse repetition rate of 200 kHz and a base temperature of 50 K. A laser pulse energy of 100 pJ was applied, achieving a detection rate of 0.01 atoms pulse$^{-1}$. The acquired data were reconstructed and analyzed using CAMECA AP Suite 6.3 software. All elemental species constituting the AgIn$_x$Ga$_{1-x}$S$_2$-based QDs, as well as C-related ions, were identified from the mass spectra and assigned before reconstruction. The reconstructions were performed using a shank-angle-based protocol. SEM and HAADF–STEM images were utilized to guide the reconstruction and constrain the specimen geometry, thereby enhancing the accuracy of the APT reconstruction process. No artificial geometric constraints were imposed (e.g., forcing spherical particle shapes) during the reconstruction process. The slightly ellipsoidal morphologies observed in the APT datasets were therefore preserved as reconstructed rather than being artificially forced into an idealized geometry and are attributed to known trajectory aberration and local magnification effects.

### Reporting summary

Further information on research design is available in the Nature Portfolio Reporting Summary linked to this article.

## Data availability

The data that support the findings of this study are available from the corresponding authors upon request. Source data are provided with this paper.

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

## Acknowledgements

This work was supported by the National Research Foundation of Korea (NRF) grant funded by the Korea government (MSIT) (RS-2026-25469493).

## Author contributions

B.G.C., M.L., J.M.N. S.S., and T.G.K. conceived and designed the study. B.G.C. and M.L. wrote the manuscript. M.L., N.W., S.K.K. and A.J. synthesized the QDs and measured the UV–vis absorption and PL characteristics. J.L. conducted the STEM experiments. D.J.Y. carried out the XPS analyses, and S.L. conducted the XRD measurements. B.G.C. performed the FIB and APT experiments. J.M.N., S.S. and T.G.K. supervised the study. All authors discussed the results and provided feedback on the manuscript.

## Competing interests

The authors declare no competing interests.
