## [Transparent Peer Review file · Nature Communications]

3D mapping of compositional gradients of core-shell structures in $\text{AgIn}_x\text{Ga}_{1-x}\text{S}_2$ quantum dots by atom probe tomography

Corresponding Author: Dr Tae-Gon Kim

Version 0:

Reviewer comments:

Reviewer #1

(Remarks to the Author)

This manuscript presents an impressive and highly novel application of atom probe tomography (APT) to elucidate the 3D atomic structure of complex $\text{AgIn}_x\text{Ga}_{1-x}\text{S}_2$ -based quaternary quantum dots. The use of APT to map these materials at this scale is a significant technical achievement, and the findings—particularly regarding compositional gradients, precursor reactivity effects, and the spontaneous formation of a $\text{Zn}_{(1-3/2x)}\text{Ga}_x\text{S}$ shell via cation exchange—are of considerable interest to the nanomaterials and optoelectronics communities.

The work is thorough and the data is compelling. However, there are several key issues regarding data interpretation, visualization, and the substantiation of claims that require major revision before this manuscript can be considered for publication.

Below are specific points that must be addressed:

1. The APT technique is fundamentally destructive, "disassembling" the specimen atom-by-atom via high-field evaporation. A critical and unaddressed point is how the authors can be certain that the resulting 3D reconstruction faithfully represents the original, pre-analysis structure of the QDs. High electric fields and laser pulses can, in some cases, induce artifacts such as atomic diffusion or re-ordering during the analysis. The authors must include a discussion on the potential for such artifacts and provide a justification (e.g., by referencing reconstruction parameters, evaporation behavior, or literature on similar materials) for why the presented 3D maps are considered a reliable representation of the as-synthesized QD structure.
2. The manuscript's optical analysis (e.g., Figure 1) focuses on the bright green emission peak at ~530 nm. However, the provided PL spectra appear to show a correspondingly enhanced fluorescence in the long-wavelength region (approx. 600-700 nm) after the ZnS/cation exchange step. This feature is not discussed. The authors must provide a clear hypothesis or explanation for the origin of this long-wavelength emission. Is this attributable to unpassivated defect states, newly formed interfacial trap states, or a specific structural component? A complete optical analysis requires addressing all significant features in the spectra.
3. The 3D atomic map in Figure 2a is intended to show the five QDs analyzed, but the presence of C atoms (presumably from ligands or the matrix) makes it very difficult to discern the specific shapes or interfaces of the QDs themselves. To improve image clarity, it is strongly recommended to remove the C atoms (and any other extraneous matrix-related atoms) from the visualization in Figure 2a (and Figure 4a). This would provide a much clearer view of the actual QD structures being analyzed.
4. The authors provide a convincing chemical argument that the Ag-deficient $\text{AgIn}_y\text{Ga}_{1-y}\text{S}_2$ outermost shell is associated with the relatively lower reactivity of Ga^{3+} compared to Ag^+ . This is a key finding. To strengthen this claim, have the authors attempted (or can they propose) an experiment to confirm this mechanism? For example, using a low-activity Ag precursor should, in principle, lead to a more uniform elemental distribution. A discussion of this point, or ideally further experimental evidence, is needed to more robustly validate the proposed formation mechanism.
5. A close inspection of the 2D compositional profile in Figure 4d reveals an anomaly. For QD#4, Zn atoms (blue squares) are shown to be present within the QD core, not just in the outer shell. This observation is not discussed by the authors, whose analysis focuses on cation exchange at the shell interface. The authors must address this finding.
6. There is a fundamental contradiction in the manuscript's description of the final shell. The text frequently refers to

"synthesizing the ZnS shell", "ZnS shell coating", or "shell growth". However, two key pieces of the authors' own data refute this:

- TEM Data: The authors state the QDs "grow only negligibly" and remain "similar in both size and shape" after the ZnS precursor step. This is inconsistent with the epitaxial growth of a new shell.
- APT Data: The authors' own excellent APT analysis proves the process is cation exchange, forming a $\text{Zn}(1-3/2x)\text{Ga}_x\text{S}$ shell.

The authors must revise the manuscript to consistently describe this process as cation exchange, as their own evidence strongly contradicts the narrative of simple "shell growth."

7. The authors claim that the final $\text{Zn}(1-3/2x)\text{Ga}_x\text{S}$ shell "maintains the charge carrier confinement with a Type-I band alignment". This is a critical conclusion for the QD's performance, but it is not experimentally proven. This claim appears to be an inference based on the band gaps of related bulk materials. Given the novel, complex, and compositionally graded structure revealed by APT, such an assumption is not sufficient. The authors must provide spectroscopic evidence (e.g., time-resolved PL or other measurements) to conclusively demonstrate the Type-I alignment. If such evidence is unavailable, this claim must be significantly toned down and presented as a well-reasoned hypothesis, not a confirmed fact.

Reviewer #2

(Remarks to the Author)

The present manuscript simply reported the measurement of AgInGaS QDs with atom probe tomography (APT). Although atom resolution images were obtained, no relationship between the nanostructure of QDs and their photochemical properties were reported. Thus, no novel results enough to publish in highly ranked journals were contained in the present manuscript.

Especially, following points were very weak.

1. As mentioned in the introduction, the authors claimed that they three-dimensionally characterize quaternary QDs ($\text{AgIn}_x\text{Ga}_{1-x}\text{S}_2$ -based heterostructured QDs with core/shell and core/shell/shell structures in this case) on the atomic scale for the first time using APT. If so, the authors must show at first the reliability of the APT method for measurements of QD nanostructures, using well known QDs, such as CdSe QDs and CdSe/ZnS core-shell structured QDs.
2. No relationship between the nanostructure of QDs and their photochemical properties were reported.
3. No definition of the x and y values were indicated. No explanation of $x \gg y$ was made, because each formula only contained x or y.
4. The charge neutrality in single QDs and its spatial modulation should be discussed.
5. $\text{AgIn}_x\text{Ga}_{1-x}\text{S}_2/\text{AgGaS}_2$ had quite Ag-deficient surface layer. What is the crystal structure of this layer? Ag-doped Ga_2S_3 ??
6. Also, what is the crystal structure of the surface of $\text{AgIn}_x\text{Ga}_{1-x}\text{S}_2/\text{AgGaS}_2/\text{ZnS}$? The authors assumed the ZnGaS alloy layer was formed on the surface. However, no difference of lattice fringes between core and shell in the corresponding HAADF-STEM images (Fig. 1) was observed.

Thus, the present manuscript does not contain novel results. The manuscript may not be suitable for the publication in the preset journal.

Reviewer #3

(Remarks to the Author)

The authors present an important study on the atomistic level for Quantum Dots using APT. The interface design and the intermixing of the components is still a great uncertainty for quantum dots, therefore, a knowledge of the distribution of the components is very important. APT is very promising for obtain more important and exciting results about the intermixing which can use for a better tuning of the synthesis of QDs. All in all, the manuscript is very important contribution in this field. But there are some points which should be clarified before publication:

In Figs. 2, 4, and 6 different particles are shown in (a), in Figs. 2 and 4 there are five. It is unclear if all particles of them were further evaluated or only one of them was evaluated. In Figs. 4 and 6 the five particles are not homogeneous, in QD 4 and QD 5 (Fig.4) Ag seems be not so present, in Fig. 6 In was inhomogeneously distributed. Was the Cl correlated with some of the cations? I am missing a statement about this question?

The general problem of all microscopic studies are the statistical relevance. Some comments to this issue would be very helpful for the readership of this manuscript.

It would be helpful, to mark the single particles in Fig. 2, 4 and 6, then the reader can see the borders of the particles.

Reviewer #4

(Remarks to the Author)

The paper highlights an additional pathway to uncover the complex internal structure of quaternary AgInGaS core-shelled quantum dots using atom probe tomography (APT). It is a well-written paper, providing elemental compositional profiles across nanocrystals to obtain detailed information about the compositional variations of core-shell and core-shell-shell nanocrystals. Furthermore, insights are provided about relative precursor reactivity and potential substitution reactions occurring, as to why and how some of these layers were formed. I recommend publication in nature communications, with minor revisions.

--The authors make broad claims about how the compositional gradient would improve basically every QD property (quantum yield, stability, longevity, etc). The authors should provide some literature references that have shown this or indicate that this should be the case.

--A prominent tail is seen in the PL spectra after the addition of a ZnS shell (or alloyed ZnGaS layer as the authors later conclude). Could the others discuss the origin of this tail? Is it related to the compositional gradients observed in the QDs.

--In Figures 2c and 4c: is the iso-surface proxigram shown just for one quantum dot? How does it differ between the 5 QDs identified?

--What happens if quantum dots aggregate? Is it possible to differentiate using the APT technique aggregated regions of QDs from individual qds that have clearly defined domains?

--For the core-shell-shell structure (with ZnS), is the first shell still Ag-deficient like with the core-shell structure (as mentioned in line 76)?

--Since the nanocrystal size does not change after ZnS shelling, does this mean the core-size decreases after addition of the ZnS shell? Or does the thickness of the first shell (AgGaS₂ shell) decrease instead? Is it possible to estimate the diameter of the core and the thickness of two shells using this technique?

--As mentioned in line 235 to 245, why was the compositional inhomogeneity more pronounced in the core versus the shell for the samples presented? Also, why does this inhomogeneity get more noticeable/severe after shelling with ZnS?

--X-ray diffraction pattern not included. Can the XRD of the QDs before and after ZnS shelling be included as supporting information? This would further corroborate the high-resolution STEM imaging showing that the particles are structurally similar before and after coating with ZnS (lines 114 to 119).

--Do the authors have elemental analysis (such as ICP-OES or ICP-MS) of the sample to compare/confirm the elemental composition obtained from atomic probe tomography (APT)?

--Method section (line 330) indicates that core-only AgInGaS were synthesized as well. How does the size, morphology, crystal structure and elemental composition between the core-only and core-shell (and core-shell-shell) nanocrystals differ? Was APT conducted on the core-only QDs? If so, are the elemental composition trends seen for the core-only, similar to the cores of the shelled QDs? Is inhomogeneity between different nanocrystals seen also for the core-only samples? Or is it a by-product of shelling?

Version 1:

Reviewer comments:

Reviewer #1

(Remarks to the Author)

The quality of the revised ms has been improved. All the concerns have been addressed. I now recommend acceptance of this work in Nature Communications.

Reviewer #2

(Remarks to the Author)

The present manuscript has been appropriately revised and may be suitable for publication.

Reviewer #3

(Remarks to the Author)

The authors revise their manuscript appropriately and address all my concerns.

Reviewer #4

(Remarks to the Author)

This is a very interesting manuscript, addressing the challenge of compositional analysis and mapping in individual nanocrystals with chemical complexity--alloys, doped nanocrystals, core/shell, etc. The technique provides information that is not readily accessed using other techniques. I think that the authors have adequately addressed the reviewer comments in their revision. This manuscript should continue to inspire researchers in the field to utilize this technique to analyze their materials.

RESPONSE TO REVIEWERS' COMMENTS

Reviewer #1 (Remarks to the Author):

General Comment: This manuscript presents an impressive and highly novel application of atom probe tomography (APT) to elucidate the 3D atomic structure of complex $\text{AgIn}_x\text{Ga}_{1-x}\text{S}_2$ -based quaternary quantum dots. The use of APT to map these materials at this scale is a significant technical achievement, and the findings—particularly regarding compositional gradients, precursor reactivity effects, and the spontaneous formation of a $\text{Zn}(1-3/2x)\text{Ga}_x\text{S}$ shell via cation exchange—are of considerable interest to the nanomaterials and optoelectronics communities. The work is thorough and the data is compelling. However, there are several key issues regarding data interpretation, visualization, and the substantiation of claims that require major revision before this manuscript can be considered for publication.

Answer to General Comment: We sincerely thank the reviewer for the careful evaluation of our manuscript and for recognizing the novelty and technical significance of applying atom probe tomography to elucidate the 3D atomic structure of complex $\text{AgIn}_x\text{Ga}_{1-x}\text{S}_2$ -based quaternary quantum dots. We particularly appreciate the reviewer's positive and constructive assessment of our findings. We have carefully addressed all raised concerns and revised the manuscript to improve data interpretation and visualization, as well as the substantiation of our key claims, as detailed below.

Below are specific points that must be addressed:

Comment 1. The APT technique is fundamentally destructive, "disassembling" the specimen atom-by-atom via high-field evaporation. A critical and unaddressed point is how the authors can be certain that the resulting 3D reconstruction faithfully represents the original, pre-analysis structure of the QDs. High electric fields and laser pulses can, in some cases, induce artifacts such as atomic diffusion or re-ordering during the analysis. The authors must include a discussion on the potential for such artifacts and provide a justification (e.g., by referencing reconstruction parameters, evaporation behavior, or literature on similar materials) for why the presented 3D maps are considered a reliable representation of the as-synthesized QD structure.

Answer to Comment 1: We appreciate the reviewer's important point regarding the inherently destructive nature of APT and the possibility that high electric fields and laser pulses may induce artifacts such as atomic diffusion or reconstruction distortion during analysis. We acknowledge that such effects must be carefully considered, particularly when APT is applied to nanoscale systems such as quantum dots (QDs).

The applicability and reliability of APT for 3D compositional analysis of nanoparticles have been demonstrated in multiple prior studies. In particular, our previous work on CdSe-based core/shell QDs (Chae et al. ACS Nano, 2018), while focusing on a relatively simple binary QD system, directly showed that APT can reliably resolve core/shell architectures in colloidal QDs, with results consistent with

complementary transmission electron microscopy (TEM) observations. In that work, we discussed the application of APT to commercial CdSe/ZnS core/shell QDs, the representativeness and reliability of the resulting datasets, potential artifacts and reconstruction considerations. More broadly, numerous APT studies on metallic and semiconductor nanostructures have established APT as a viable technique for nanoscale structural and compositional characterization.

We also recognize that specific artifacts have been reported in APT analyses of nanostructures and QDs, such as apparent geometric distortion along the analysis (z) direction, artificially broadened interfaces at nominally sharp interfaces, and apparent intermixing between adjacent regions. These effects are primarily associated with trajectory aberrations arising from the local magnification effect and cannot be entirely eliminated in APT analyses of nanostructures. In the revised manuscript we explicitly acknowledge these potential artifacts and cite relevant prior studies that have investigated their origin and impact.

Importantly, in APT analyses of nanostructures, the reconstruction fidelity depends on the analysis direction. Compositional results obtained along the direction perpendicular to the interfaces—corresponding to the z-direction in APT—have been reported to provide more reliable quantitative information in resolving the distributions of segregating atoms at grain boundaries as well as constituent atoms within nanoparticles or precipitates. The improved reliability of z-direction analyses originates from the reduced trajectory overlap, which effectively suppresses artificial intermixing and preserves compositional distributions and interfacial characteristics. In contrast, lateral (x/y) analyses are more susceptible to local magnification effects and thus tend to overestimate interfacial broadening.

In the present study, we therefore place particular emphasis on z-direction compositional analyses, while also cross-validating the observed trends using proxigrams and lateral (x/y) profiles. Although minor variations in profile broadening are observed depending on the analysis direction, all datasets consistently reveal the same qualitative compositional tendencies. The absence of severe trajectory overlap observed in the reconstructed datasets further suggests that intermixing artifacts are minimized in our datasets.

Furthermore, the compositional features revealed by APT were corroborated using complementary characterization techniques. TEM observations show compositional trends consistent with the APT results at the spatial resolution accessible by electron microscopy. In addition, XPS analyses support the presence of the key phenomena identified by APT, such as $Zn_{1-3/2x}Ga_xS$ formation and Cl distribution at the outer shell, further corroborating the validity of our interpretations.

We also have clarified the APT acquisition and reconstruction procedures in the Methods section. All elemental species were identified from the mass spectra prior to reconstruction, and the datasets were reconstructed using a standard shank-angle-based protocol. SEM images of the APT specimens and HAADF-STEM images of the QDs were used to guide the reconstruction and constrain the specimen geometry. This approach improves reconstruction fidelity without artificially modifying the particle shape, and these details have now been added to the revised manuscript. We have revised the manuscript to include a dedicated discussion of these limitations and justifications, clarifying why the presented 3D compositional maps can reasonably be regarded as representative of the original QD architecture.

The manuscript has been revised as follows:

Please see Page #7,

“To characterize and quantify the 3D atomic distribution of the $AgIn_xGa_{1-x}S_2$ -based core/shell structured

QDs, APT was employed. For the APT analysis in this work, dried QDs were prepared in powder form and analyzed following our previously described method.⁴¹ In that study, we demonstrated that APT could faithfully resolve core/shell architectures and 3D compositional distributions in commercially available CdSe/ZnS core/shell QDs, in good agreement with TEM observations. Specimen preparation strategies, potential APT artifacts, and reconstruction considerations specific to CdSe/ZnS core/shell QDs were systematically examined, thereby establishing the applicability of APT to colloidal QD nanostructures. More broadly, the feasibility of APT in the analysis of metallic nanoparticles has been demonstrated in numerous prior studies, including ligand-resolved analyses and combined experiment–simulation approaches.⁴²⁻⁴⁷ On this basis, we report a 3D atomic-scale characterization of $\text{AgIn}_x\text{Ga}_{1-x}\text{S}_2$ -based QDs using APT, which, to the best of our knowledge, represents the first such analysis of quaternary QDs.”

Please see Page #8,

“Figure 2a shows the 3D reconstructed APT atom maps of the $\text{AgIn}_x\text{Ga}_{1-x}\text{S}_2/\text{AgGaS}_2$ QDs, representing a hemispherical APT specimen containing five QDs (see Supplementary Fig. 6 for dashed guides indicating the approximate outlines of individual QDs). The QDs appear as slightly ellipsoidal features in the APT reconstruction, and a slice-view image along the x–z plane is presented in Fig. 2b to visualize their internal elemental distribution. We note that APT analyses of nanostructures may exhibit reconstruction artifacts, such as apparent elongation along the analysis (z) direction and interfacial broadening, primarily arising from local magnification and trajectory aberrations.⁴²⁻⁵² Accordingly, the slightly ellipsoidal morphology observed here is attributed to such effects rather than the intrinsic particle shape. However, compositional distributions along the z-direction have been shown to provide more reliable quantitative information than lateral directions because the reduced trajectory overlap suppresses artificial intermixing.^{42-44,51,52} Under these conditions, compositional distributions and interfacial characteristics closer to the intrinsic distributions can be more faithfully evaluated.”

Please see Page #9,

"The 2D compositional profiles along two different directions (Figs. 3c and 3f) further confirm the formation of the $\text{AgIn}_x\text{Ga}_{1-x}\text{S}_2$ core/ AgGaS_2 shell heterostructure with a compositional gradient. From

the extracted profiles, we primarily focus on z-direction compositional analyses, cross-validated by proxigrams and lateral profiles. Ag and In atoms are highly concentrated at the center of the QD, at more than 20 and 10 at.%, respectively, but their concentrations gradually decrease toward the shell.”

Please see Page #12,

“Figures 5a and 5d show the extracted 3D atom maps and the cross-sectional regions cropped by the green boxes for each QD. According to the z-direction compositional profiles cross-validated by proxigrams and lateral profiles, Ag atoms are more widely distributed than In atoms in the 3D atom maps (Figs. 5b and 5e). However, both Ag and In atoms are at their highest concentrations in the central region of the QDs, with values exceeding 20 and 10 at.%, respectively.”

Please see Page #13,

“Because APT inherently probes only a limited number of individual QDs and may therefore suffer from limited statistical representativeness, the compositional features identified by APT were further examined using complementary techniques. XPS analyses performed on the same samples reveal consistently higher Ga-to-Ag ratios after the Zn precursor step, evaluated from multiple core-level regions, including Ag *MNN* vs Ga *2p* and Ag *4d* vs Ga *3d*. This systematic increase in the Ga-to-Ag ratios supports the formation of Ga-containing outer shells and is consistent with the $Zn_{1-3/2x}Ga_xS$ shell formation revealed by APT (Supplementary Fig. 15 and Supplementary Table 2). Inductively coupled plasma analyses performed on the same samples show overall elemental compositions consistent with the APT results (Supplementary Tables 3 and 4). Taken together, these results indicate that the structural and compositional features identified by APT are not limited to a small number of individual QDs but are representative of overall sample.”

Please see Page #21,

“**APT analysis**

APT analysis was conducted using a CAMECA atom probe (Invivo 6000) operating in the deep-UV laser mode ($\lambda = 257.5$ nm) with a pulse repetition rate of 200 kHz and a base temperature of 50 K. A laser pulse energy of 100 pJ was applied, achieving a detection rate of 0.01 atoms pulse⁻¹. The acquired data were reconstructed and analyzed using CAMECA AP Suite 6.3 software. All elemental species

constituting the $\text{AgIn}_x\text{Ga}_{1-x}\text{S}_2$ -based QDs, as well as C-related ions, were identified from the mass spectra and assigned before reconstruction. The reconstructions were performed using a shank-angle-based protocol. SEM and HAADF-STEM images were utilized to guide the reconstruction and constrain the specimen geometry, thereby enhancing the accuracy of the APT reconstruction process. No artificial geometric constraints were imposed (e.g., forcing spherical particle shapes) during the reconstruction process. The slightly ellipsoidal morphologies observed in the APT datasets were therefore preserved as reconstructed rather than being artificially forced into an idealized geometry and are attributed to known trajectory aberration and local magnification effects.”

Please see Supplementary Fig. 15.

Supplementary Fig. 15 XPS analysis results of $\text{AgIn}_x\text{Ga}_{1-x}\text{S}_2$ -based QDs. XPS spectra of **a** Ga $2p$, **b** Zn $2p_{3/2}$, **c** In $3d$, **d** Ag $3d$, and **e** S $2s$ core levels acquired from $\text{AgIn}_x\text{Ga}_{1-x}\text{S}_2$ (AIGS), $\text{AgIn}_x\text{Ga}_{1-x}\text{S}_2/\text{AgGaS}_2$ (AIGS/AGS), and $\text{AgIn}_x\text{Ga}_{1-x}\text{S}_2/\text{AgGaS}_2/\text{ZnS}$ (AIGS/AGS/ZnS) QDs using an Al K-alpha X-ray source. All QDs contain Ag-S, In-S, Ga-S, and Zn-S chemical bonds, as designed without detectable oxidation. **f** Valence band spectra of AIGS, AIGS/AGS, and AIGS/AGS/ZnS QDs. The Ga-to-Ag ratio is consistently higher in the AIGS/AGS/ZnS QDs than in the AIGS/AGS QDs. The detailed compositions are provided in **Supplementary Table 2**. These results further support the presence of a

Ga-containing outer shell, consistent with the $Zn_{1-3/2x}Ga_xS$ shell identified by APT after the ZnS precursor step.

Please see Supplementary Fig. 20,

Supplementary Fig. 20 XPS Cl 2p core-level spectra of $AgIn_xGa_{1-x}S_2$ -based QDs. XPS spectra of Cl 2p core-level acquired from $AgIn_xGa_{1-x}S_2$ (AIGS), $AgIn_xGa_{1-x}S_2/AgGaS_2$ (AIGS/AGS), and $AgIn_xGa_{1-x}S_2/AgGaS_2/ZnS$ (AIGS/AGS/ZnS) QDs. Cl-related signals are observed only in the AIGS/AGS/ZnS QDs, indicating the presence of metal–Cl chemical states associated with the Zn-based shell introduced during the ZnS precursor step.

Please see Supplementary Table 2,

Specimen	Atomic composition (%) (XPS)								
	C (in C 1s)	O (in O 1s)	F (in F 1s)	S (in S 2s)	Cl (in Cl 2p)	Zn (in Zn 2p _{3/2})	Ga (in Ga 2p _{3/2})	Ag (in Ag 3d)	In (in In 3d _{5/2})
AIGS	76.29	1.91	0	11.54	0	0	1.1	4.46	4.7
AIGS/AGS	69.37	2.83	0	14.69	0.23	0	5.16	5.9	1.83

AIGS/AGS/ZnS	62.92	6.58	0	13.93	2.07	6.07	4.54	2.85	1.06
-------	------	---	-------	------	------	------	------	------

Specimen	Atomic composition (%) (XPS)			
	S (in S 2s)	Ga (in Ga 2p _{3/2})	Ag (in Ag 3d)	In (in In 3d _{5/2})
AIGS	52.92	5.07	20.44	21.57
AIGS/AGS	53.25	18.71	21.41	6.63
AIGS/AGS/ZnS	62.28	20.28	12.73	4.72

Specimen	Atomic composition (%) (XPS)			
	Ag (in Ag 4d)	Zn (in Zn 3d)	In (in In 4d)	Ga (in Ga 3d)
AIGS	36.1	0	37.7	26.2
AIGS/AGS	37.1	0	10.0	52.9
AIGS/AGS/ZnS	21.3	30.5	5.1	43.1

Supplementary Table 2. Atomic compositions of AgIn_xGa_{1-x}S₂/AgGaS₂ (AIGS/AGS) and AgIn_xGa_{1-x}S₂/AgGaS₂/ZnS (AIGS/AGS/ZnS) QDs measured by XPS. The Ga-to-Ag ratio is consistently higher in the AIGS/AGS/ZnS QDs than in the AIGS/AGS QDs, suggesting the formation of a Ga-containing outer shell. This trend is observed both when comparing Ag *MNN* and Ga *2p* and when comparing Ag *4d* and Ga *3d*, all of which lie in similar binding energy ranges

Please see Supplementary Table 3.

Specimen	Atomic ratio (ICP-OES)			
	Ag/S	In/S	Ga/S	Zn/S
AIGS/AGS	0.40	0.12	0.44	0
AIGS/AGS/ZnS	0.13	0.08	0.35	0.46

Supplementary Table 3. Atomic ratios measured of $\text{AgIn}_x\text{Ga}_{1-x}\text{S}_2/\text{AgGaS}_2$ (AIGS/AGS) and $\text{AgIn}_x\text{Ga}_{1-x}\text{S}_2/\text{AgGaS}_2/\text{ZnS}$ (AIGS/AGS/ZnS) determined by ICP–OES.

Please see Supplementary Table 4.

Specimen	Atomic ratio (APT)			
	Ag/S	In/S	Ga/S	Zn/S
AIGS/AGS	0.43	0.13	0.39	0
AIGS/AGS/ZnS	0.15	0.08	0.30	0.48

Supplementary Table 4. Atomic ratios of $\text{AgIn}_x\text{Ga}_{1-x}\text{S}_2 / \text{AgGaS}_2$ (AIGS/AGS) and $\text{AgIn}_x\text{Ga}_{1-x}\text{S}_2/\text{AgGaS}_2/\text{ZnS}$ (AIGS/AGS/ZnS) determined by APT.

References have been added as follows:

41. Chae, B. G. et al. Direct three-dimensional observation of core/shell-structured quantum dots with a composition-competitive gradient. *ACS Nano* **12**, 12109-12117 (2018). [10.1021/acsnano.8b05379](https://doi.org/10.1021/acsnano.8b05379).
42. Jang, K. et al. Three-dimensional atomic mapping of ligands on palladium nanoparticles by atom probe tomography. *Nat. Commun.* **12**, 4301 (2021). [10.1038/s41467-021-24620-9](https://doi.org/10.1038/s41467-021-24620-9).
43. Kim, S. H. et al. Characterization of Pd and Pd@Au core-shell nanoparticles using atom probe tomography and field evaporation simulation. *J. Alloys Compd.* **831**, 154721 (2020). [10.1016/j.jallcom.2020.154721](https://doi.org/10.1016/j.jallcom.2020.154721).
44. Tedsree, K. et al. Hydrogen production from formic acid decomposition at room temperature using a Ag–Pd core–shell nanocatalyst. *Nat. Nanotechnol.* **6**, 302-307 (2011). [10.1038/nnano.2011.42](https://doi.org/10.1038/nnano.2011.42).
45. Grenier, A. et al. 3D analysis of advanced nano-devices using electron and atom probe tomography. *Ultramicroscopy* **136**, 185-192 (2014). [10.1016/j.ultramic.2013.10.001](https://doi.org/10.1016/j.ultramic.2013.10.001).

46. Khan, M. A., Ringer, S. P. & Zheng, R. Atom probe tomography on semiconductor devices. *Adv. Mater. Interfaces* **3**, 1500713 (2016). [10.1002/admi.201500713](https://doi.org/10.1002/admi.201500713).
47. Hatzoglou, C., Radiguet, B. & Pareige, P. Experimental artefacts occurring during atom probe tomography analysis of oxide nanoparticles in metallic matrix: quantification and correction. *J. Nucl. Mater.* **492**, 279-291 (2017). [10.1016/j.jnucmat.2017.05.008](https://doi.org/10.1016/j.jnucmat.2017.05.008).
48. Philippe, T., Gruber, M., Vurpillot, F. & Blavette, D. Clustering and local magnification effects in atom probe tomography: a statistical approach. *Microsc. Microanal.* **16**, 643-648 (2010). [10.1017/S1431927610000449](https://doi.org/10.1017/S1431927610000449).
49. Lawitzki, R., Stender, P. & Schmitz, G. Compensating local magnifications in atom probe tomography for accurate analysis of nano-sized precipitates. *Microsc. Microanal.* **27**, 1–12 (2021). [10.1017/S1431927621000180](https://doi.org/10.1017/S1431927621000180).
50. Beinke, D., Oberdorfer, C. & Schmitz, G. Towards an accurate volume reconstruction in atom probe tomography. *Ultramicroscopy* **165**, 34-41 (2016). [10.1016/j.ultramic.2016.03.008](https://doi.org/10.1016/j.ultramic.2016.03.008).
51. Takahashi, J. & Kawakami, K. Position artifacts in 3D reconstruction of plate-shaped precipitates in steels depending on the analysis direction of atom probe tomography. *Surf. Interface Anal.* **53**, 982-995 (2021). [10.1002/sia.7001](https://doi.org/10.1002/sia.7001).
52. Maruyama, N., Smith, G. D. W. & Cerezo, A. Interaction of the solute niobium or molybdenum with grain boundaries in α -iron. *Mater. Sci. Eng. A* **353**, 126-132 (2003). [10.1016/S0921-5093\(02\)00678-0](https://doi.org/10.1016/S0921-5093(02)00678-0).

Comment 2. The manuscript's optical analysis (e.g., Figure 1) focuses on the bright green emission peak at ~530 nm. However, the provided PL spectra appear to show a correspondingly enhanced fluorescence in the long-wavelength region (approx. 600-700 nm) after the ZnS/cation exchange step. This feature is not discussed. The authors must provide a clear hypothesis or explanation for the origin of this long-wavelength emission. Is this attributable to unpassivated defect states, newly formed interfacial trap states, or a specific structural component? A complete optical analysis requires

addressing all significant features in the spectra.

Answer to Comment 2: We thank the reviewer for pointing out the enhanced long-wavelength emission (approximately 600–700 nm) observed after the ZnS/cation exchange process. We agree that all major spectral features should be addressed, and accordingly, we have revised the manuscript to explicitly discuss the origin of the long-wavelength emission.

Long-wavelength emission following ZnS shell coating has been widely observed in AgInS₂- and related I–III–VI₂-based quantum dots (QDs) and is generally attributed to defect- or trap-related recombination pathways rather than band-edge excitonic transitions. As the reviewer correctly notes, several mechanisms may contribute to such emission, including unpassivated defect states, newly formed interfacial trap states, and defects introduced during cation exchange. To date, however, the precise origin of this long-wavelength emission has not yet been fully established.

In the present system, we attribute the emergence of the long-wavelength (tail) emission primarily to defect formation during the ZnS precursor step and the associated cation exchange process. Our transmission electron microscopy and X-ray diffraction analyses indicate that the shell retains a chalcopyrite-related tetragonal structure rather than forming an incoherent zinc-blende-type ZnS phase, suggesting that Zn_{1-3/2x}Ga_xS is formed via cation exchange through substitution of Ag⁺ and Ga³⁺ by Zn²⁺. This intrinsically non-charge-neutral process is expected to generate a substantial density of cation vacancies, which act as intragap trap states in I–III–VI₂ semiconductors. Such defect-related intragap states are commonly associated with donor–acceptor-pair-like recombination, which has been proposed as a major origin of long-wavelength emission. Similar trap-assisted long-wavelength emissions have been reported in AgInS₂/ZnS and Zn-alloyed Ag–In–S QDs, where they are commonly attributed to defect-mediated donor–acceptor-pair-like recombination rather than band-edge transitions. In addition, multiple studies have reported the emergence of tail emission after ZnS shelling of AgInS₂-based QDs, and have shown that when Zn²⁺ is present in the core or inner shell regions, its presence is associated with defect formation and the emergence of trap states.

Consistent with our APT observations, the inward migration of Zn²⁺ species into the particle interior can trigger the further formation of cation vacancies and associated defect complexes. When these defects are introduced in the vicinity of In-rich domains, where charge carriers are preferentially localized, they can give rise to intragap trap states that locally perturb the confinement and facilitate trap-assisted recombination. We attempted to further elucidate the origin of this emission through time-resolved photoluminescence (TR-PL) measurements. However, the long-wavelength feature could not be reliably decomposed into distinct lifetime components, suggesting that it arises from a distribution of energetically broad and spatially heterogeneous trap states. Accordingly, the revised manuscript now explicitly discusses that the observed long-wavelength emission is consistent with defect-related recombination processes associated with the ZnS precursor step.

The manuscript has been revised as follows:

Please see Page #5,

“In addition to the dominant green emission, the AgIn_xGa_{1-x}S₂/AgGaS₂/ZnS QDs exhibit a long-wavelength emission tail in the 600–700 nm range (Fig. 1d). The area ratio of this long-wavelength emission, defined as the fraction of the total PL intensity in the long-wavelength region ($\lambda > \lambda_{\text{PL max}} +$

50 nm), increases from ~3.7% in $\text{AgIn}_x\text{Ga}_{1-x}\text{S}_2/\text{AgGaS}_2$ QDs to ~14.9% in $\text{AgIn}_x\text{Ga}_{1-x}\text{S}_2/\text{AgGaS}_2/\text{ZnS}$ QDs. Such long-wavelength emission is generally attributed to defect- or trap-assisted recombination rather than band-edge excitonic transitions in AgInS_2 -based QDs and related I–III–VI₂-based QDs.³⁶⁻⁴⁰

Please see Page #6.

“X-ray diffraction (XRD) measurements were further performed on the core-only $\text{AgIn}_x\text{Ga}_{1-x}\text{S}_2$, $\text{AgIn}_x\text{Ga}_{1-x}\text{S}_2/\text{AgGaS}_2$, and $\text{AgIn}_x\text{Ga}_{1-x}\text{S}_2/\text{AgGaS}_2/\text{ZnS}$ QDs to corroborate the structural similarity before and after shell formation. All XRD patterns are consistent with a chalcopyrite-based tetragonal structure, and the overall diffraction features remain essentially unchanged, even after the Zn precursor step (Supplementary Fig. 4). This comparative analysis indicates that the crystal structure is preserved throughout the shell formation process.”

Please see Page #15.

“The emergence of the $\text{Zn}_{1-3/2x}\text{Ga}_x\text{S}$ outer shell via cation exchange offers a plausible origin for the long-wavelength emission tail observed after the ZnS precursor step. In our system, the substitution of Ag^+ and Ga^{3+} by Zn^{2+} during shell formation inherently introduces cation vacancies and local charge-compensating defects. Consistent with our APT observations, the inward diffusion of Zn^{2+} into the interior of the QDs further induces cation exchange and promotes the formation of such defects. When these defects form in the vicinity of In-rich regions, where carriers are preferentially confined, they can introduce intragap trap states that locally perturb the confinement potential and promote trap-assisted recombination. These defect-related states are commonly associated with donor–acceptor-pair-like recombination pathways that give rise to long-wavelength emission in I–III–VI₂ semiconductors.^{23,38-40,66-68}”

Please see Fig. 1.

“**Fig. 1 Comparison of $\text{AgIn}_x\text{Ga}_{1-x}\text{S}_2/\text{AgGaS}_2$ QDs and $\text{AgIn}_x\text{Ga}_{1-x}\text{S}_2/\text{AgGaS}_2/\text{ZnS}$ QDs.** a, d Schematic representation of the internal structure, band diagrams, and UV–vis absorption, and PL characteristics. Both QDs exhibit bright green luminescence at ~530 nm, with the QY increasing from 85% to 92% after the ZnS precursor step to form the outer shell. In addition, $\text{AgIn}_x\text{Ga}_{1-x}\text{S}_2/\text{AgGaS}_2/\text{ZnS}$ QDs exhibit a long-wavelength emission tail in the ~600–700 nm range.”

Please see Supplementary Fig. 4.

Supplementary Fig. 4 XRD patterns of $\text{AgIn}_x\text{Ga}_{1-x}\text{S}_2$ -based QDs. XRD patterns of $\text{AgIn}_x\text{Ga}_{1-x}\text{S}_2$ (AIGS), $\text{AgIn}_x\text{Ga}_{1-x}\text{S}_2/\text{AgGaS}_2$ (AIGS/AGS), and $\text{AgIn}_x\text{Ga}_{1-x}\text{S}_2/\text{AgGaS}_2/\text{ZnS}$ (AIGS/AGS/ZnS) QDs. All XRD patterns are consistent with a chalcopyrite-based tetragonal structure, with only minor peak shifts attributable to slight lattice parameter variations.

References have been added as follows:

23. Kim, J. H. et al. Synthesis of widely emission-tunable Ag–Ga–S and its quaternary derivative quantum dots. *Chem. Eng. J.* **347**, 791-797 (2018). [10.1016/j.cej.2018.04.167](https://doi.org/10.1016/j.cej.2018.04.167).
36. Dai, M. et al. Tunable photoluminescence from the visible to near-infrared wavelength region of non-stoichiometric AgInS_2 nanoparticles. *J. Mater. Chem.* **22**, 12851-12858 (2012). [10.1039/C2JM31463K](https://doi.org/10.1039/C2JM31463K).
37. Chen, Y. et al. Green and facile synthesis of high-quality water-soluble Ag–In–S/ZnS core/shell quantum dots with obvious bandgap and sub-bandgap excitations. *J. Alloys Compd.* **753**, 364-370 (2018). [10.1016/j.jallcom.2018.04.242](https://doi.org/10.1016/j.jallcom.2018.04.242).

38. Park, S. M. et al. Red Ag-based I–III–VI quantum dots as competitive alternatives to InP emitters. *ACS Energy Lett.* **10**, 3005–3013 (2025). [10.1021/acsenergylett.5c00962](https://doi.org/10.1021/acsenergylett.5c00962).
39. Farid, A. et al. One-pot synthesis of luminescent Ag–Cu–Ga–S/ZnS quantum dots bridging the cyan gap for ultrahigh-color-rendering white-light-emitting diodes. *ACS Appl. Nano Mater.* **8**, 14703–14712 (2025). [10.1021/acsanm.5c02386](https://doi.org/10.1021/acsanm.5c02386).
40. Park, S. et al. Suppressing tail emission from AgIn_{1-x}Ga_xS₂/AgGaS₂ quantum dots by GaI₃-assisted interface reinforcement. *ACS Nano* **19**, 26831–26842 (2025). [10.1021/acsnano.5c07418](https://doi.org/10.1021/acsnano.5c07418).
66. Shen, F. et al. Photophysics and photovoltaic properties of Zn-alloyed Ag–In–S quantum dots sensitized solar cells. *J. Alloys Compd.* **922**, 166296 (2022). [10.1016/j.jallcom.2022.166296](https://doi.org/10.1016/j.jallcom.2022.166296).
67. Rivaux, C. et al. Continuous flow aqueous synthesis of highly luminescent AgInS₂ and AgInS₂/ZnS quantum dots. *J. Phys. Chem. C* **126**, 20524–20534 (2022). [10.1021/acs.jpcc.2c06849](https://doi.org/10.1021/acs.jpcc.2c06849).
68. Kameyama, T. et al. Controlling the electronic energy structure of ZnS–AgInS₂ solid solution nanocrystals for photoluminescence and photocatalytic hydrogen evolution. *J. Phys. Chem. C* **119**, 24740–24749 (2015). [10.1021/acs.jpcc.5b07994](https://doi.org/10.1021/acs.jpcc.5b07994).

Comment 3. *The 3D atomic map in Figure 2a is intended to show the five QDs analyzed, but the presence of C atoms (presumably from ligands or the matrix) makes it very difficult to discern the specific shapes or interfaces of the QDs themselves. To improve image clarity, it is strongly recommended to remove the C atoms (and any other extraneous matrix-related atoms) from the visualization in Figure 2a (and Figure 4a). This would provide a much clearer view of the actual QD structures being analyzed.*

Answer to Comment 3: We thank the reviewer for the helpful suggestion regarding the visualization of the 3D atomic maps. We agree that C atoms originating from organic ligands can obscure the clarity of the quantum dot (QD) morphology in Figures 2a and 4a. Following the reviewer’s recommendation, we revised the visualization by excluding C- and H-related ions from the 3D atom maps. The exclusion of these species does not alter the overall QD morphology or compositional trends but improves the visual clarity of the QD shapes and interfaces. We note that even after removing C- and H-related ions, some degree of apparent overlap between neighboring QDs may remain depending on the viewing direction. In addition, because the QDs are not arranged in a perfectly linear manner but instead adopt a partially close-packed, non-collinear configuration within the APT specimen, individual particles may

not always appear completely isolated in a single projection view. The corresponding 3D reconstructions excluding C- and H-related ions are provided in the Supplementary Information.

The manuscript has been revised as follows:

Please see Page #8,

“Because the QDs are closely packed in a non-collinear configuration within the APT specimen and may therefore partially overlap in projection views, 3D reconstructions with C- and H-related ions removed are also provided in Supplementary Fig. 7 to better visualize the individual QDs.”

Please see Page #12,

“C-related signals, likely originating from ligands, were also observed on the outside of the QDs (Supplementary Fig. 13). Nevertheless, because of the difficulty in quantifying these signals, our discussion focuses on the distribution of the main elements (see Supplementary Fig. 14 for the 3D reconstructions with the C- and H-related ions removed for clarity).”

Please see Supplementary Fig. 7,

Supplementary Fig. 7 APT 3D reconstructions of AgIn_xGa_{1-x}S₂/AgGaS₂ QDs after the removal of C- and H-related species. C- and H-related ions were excluded from the datasets for visual clarity.

Both the entire reconstructed volume ($11 \times 11 \times 22 \text{ nm}^3$) and a sliced 3D ion map ($6 \times 11 \times 22 \text{ nm}^3$) are shown, with dashed guides indicating the approximate boundaries of individual QDs. Because the QDs are arranged in a close-packed, non-collinear configuration within the APT specimen, some particles may appear partially overlapped in the projection views. (Scale bars: 2 nm.)

Please see Supplementary Fig. 14.

Supplementary Fig. 14 APT 3D reconstructions of $\text{AgIn}_x\text{Ga}_{1-x}\text{S}_2/\text{AgGaS}_2/\text{ZnS}$ QDs after the removal of C- and H-related species. C- and H-related ions were excluded from the datasets for visual clarity. For better visualization, the entire 3D ion map ($8.5 \times 8.5 \times 31 \text{ nm}^3$) and a sliced 3D ion map ($6 \times 8.5 \times 31 \text{ nm}^3$) are both shown, with dashed guides indicating the approximate boundaries of individual QDs. Because the QDs are closely packed within the APT specimen, some particles may appear partially overlapped in the projection views. (Scale bars: 2 nm.)

Comment 4. The authors provide a convincing chemical argument that the Ag-deficient $\text{AgIn}_y\text{Ga}_{1-y}\text{S}_2$ outermost shell is associated with the relatively lower reactivity of Ga^{3+} compared to Ag^+ . This is a key finding. To strengthen this claim, have the authors attempted (or can they propose) an experiment to confirm this mechanism? For example, using a low-activity Ag precursor should, in principle, lead to a more uniform elemental distribution. A discussion of this point, or ideally further experimental evidence, is needed to more robustly validate the proposed formation mechanism.

Answer to Comment 4: We thank the reviewer for this constructive comment and for highlighting the importance of further validating the proposed formation mechanism.

We agree that directly varying the chemical activity of the precursors would be an ideal way to probe this mechanism. In practice, however, replacing the Ag and Ga precursors with alternative precursors of different reactivity is not straightforward under our current synthetic conditions. Instead, to experimentally assess the role of relative precursor reactivity, we performed a series of controlled syntheses in which the Ga precursor amount was fixed while the Ag precursor amount was varied.

When the Ga precursor amount was fixed and the Ag precursor amount was increased, TEM measurements revealed a clear and systematic increase in the overall QD size, most notably in the thickness of the AgGaS_2 shell. In particular, owing to the relatively low reactivity of the Ga precursor, increasing the Ag precursor under Ga-rich conditions preferentially promoted further growth of the AgGaS_2 shell. STEM–EELS analyses consistently showed increased Ag incorporation and thickening of the AgGaS_2 shell with increasing Ag precursor amount. At the same time, the trap-related emission contribution increased and the photoluminescence quantum yield decreased, suggesting that Ag-driven shell growth could be accompanied by enhanced defect formation. Importantly, in contrast, when the Ag precursor amount was fixed and the Ga precursor amount was increased, no significant increase in QD size or AgGaS_2 shell thickness was observed. This asymmetric response provides compelling experimental support for our interpretation that the outer Ag-deficient $\text{AgIn}_y\text{Ga}_{1-y}\text{S}_2$ shell originates from the relatively lower reactivity of the Ga precursor compared to Ag, leading to preferential Ag incorporation and shell growth when additional Ag is supplied, rather than a homogeneous redistribution of cations. These additional results and discussions have now been incorporated into the revised manuscript and Supplementary Information.

The manuscript has been revised as follows:

Please see Page #10,

“To demonstrate the role of differential precursor reactivity, we performed controlled syntheses in which the Ag precursor amount was increased at a fixed Ga level, a condition under which increasing the Ag precursor amount directly promotes the further growth of the AgGaS_2 shell. STEM–EELS analyses showed the pronounced thickening of the AgGaS_2 shell and consistently increased Ag incorporation, supporting the role of precursor reactivity in the formation of the shell (Supplementary Fig. 8 and Supplementary Table 1). These changes were accompanied by an increased contribution from trap-related emission and a decreased photoluminescence QY, indicating that Ag-driven shell growth directly

impacts the optical properties of the QDs.”

Please see Supplementary Fig. 8.

Supplementary Fig. 8 STEM–EELS analysis of $\text{AgIn}_x\text{Ga}_{1-x}\text{S}_2/\text{AgGaS}_2$ QDs as a function of the Ga/Ag precursor ratio. (a) Representative STEM–EELS results for high-Ga/Ag-ratio (Ag-poor) $\text{AgIn}_x\text{Ga}_{1-x}\text{S}_2/\text{AgGaS}_2$ QDs and low-Ga/Ag-ratio (Ag-rich) $\text{AgIn}_x\text{Ga}_{1-x}\text{S}_2/\text{AgGaS}_2$ QDs. As the Ag precursor amount increases, the overall diameter of the QDs increases, particularly because of the increasing thickness of the AgGaS_2 shell. Owing to the relatively low reactivity of the Ga precursor, increasing the Ag precursor under Ga-rich conditions mainly promotes the growth of the AgGaS_2 shell, accompanied by increased Ag incorporation. (Scale bars: 5 nm.)

Please see Supplementary Table 1,

Ag precursor amount	Ga/Ag (precursor ratio)	$\lambda_{\text{PL max}}$ (band edge)	FWHM (nm)	Trap emission area ratio	QY	QD size measured by TEM (nm)
Low Ag	29	524	35	5.1	77	6.0
	10	520	37	7.8	68	6.8
↓	6	517	37	12.9	57	6.9
High Ag	4	516	38	21.3	37	7.9

Supplementary Table 1. Dependence of the optical properties and size of $\text{AgIn}_x\text{Ga}_{1-x}\text{S}_2/\text{AgGaS}_2$ QDs on the Ga/Ag precursor ratio. As the Ag precursor amount increases, the average particle size of the $\text{AgIn}_x\text{Ga}_{1-x}\text{S}_2/\text{AgGaS}_2$ QDs as measured by TEM increases. Meanwhile, the trap emission increases, and the QY decreases. Owing to the relatively low reactivity of the Ga precursor, increasing the Ag precursor under Ga-rich conditions primarily leads to further growth of the AgGaS_2 shell. The trap emission area ratio was defined as the fraction of the integrated PL intensity in the long-wavelength region ($\lambda > \lambda_{\text{PL max}} + 50$ nm) to the total integrated emission.

Comment 5. A close inspection of the 2D compositional profile in Figure 4d reveals an anomaly. For QD#4, Zn atoms (blue squares) are shown to be present within the QD core, not just in the outer shell. This observation is not discussed by the authors, whose analysis focuses on cation exchange at the shell interface. The authors must address this finding.

Answer to Comment 5: We thank the reviewer for the careful inspection of Fig. 4d. The apparent presence of Zn within the core region of QD #4 should not be interpreted as intrinsic Zn incorporation into the core, but rather arises from the way the 2D compositional profile was constructed.

The 2D compositional profiles in Fig. 4d were extracted uniformly along the analysis (z) direction using a fixed cylindrical region, corresponding to a sampling volume of $3 \times 3 \times 21$ nm³. In the APT specimen, however, individual quantum dots (QDs) do not share an identical center along the z-axis but are spatially arranged in a close-packed, non-collinear manner. While QD#3 and QD#5 are located at similar z positions, QD#4 is laterally and vertically offset within the APT specimen. As a result, the z-directional 2D profile for QD#4 partially intersects the shell region, rather than exclusively sampling the core, resulting in the apparent Zn signal within the “core” region in Fig. 4d. This interpretation is consistent with the iso-surface visualization in the same figure, which resolves the core/shell/shell

architecture and, in particular, shows that Zn atoms are predominantly located at the outer shell of QD#4 rather than within the core. Furthermore, because a fixed-diameter cylindrical profile was used, variations in QD size and spatial offset inevitably lead to differences in whether the extracted volume captures only the core or includes portions of the shell.

For this reason, the individual compositional profiles extracted from individual QD volumes (Fig. 5c and 5f) provide a more direct representation of the true core/shell distribution and do not show significant Zn incorporation into the core. Nevertheless, Fig. 4d is included to demonstrate the overall consistency and reliability of the APT dataset across multiple QDs. We have clarified this point in the revised manuscript to avoid potential misinterpretation.

The manuscript has been revised as follows:

Please see Page #8,

“Because the QDs are closely packed in a non-collinear configuration within the APT specimen and may therefore partially overlap in projection views, 3D reconstructions with C- and H-related ions removed are also provided in Supplementary Fig. 7 to better visualize the individual QDs.”

Please see Page #12,

“The 2D concentration profile along the analysis direction offers additional insights into the internal structure of each QD (Fig. 4d). Although the elemental concentrations vary slightly depending on the profiling path owing to the spatial offset of individual QDs in the APT specimen, the 2D profile confirms the significant presence of Ga atoms in the outer shell, albeit at lower concentrations than Zn atoms.”

Please see Fig. 4,

Fig. 4 3D reconstruction and compositional analysis of $\text{AgIn}_x\text{Ga}_{1-x}\text{S}_2/\text{AgGaS}_2/\text{ZnS}$ QDs using APT.

a Reconstructed 3D atom map of the entire analyzed volume, showing five slightly ellipsoidal QDs. **b**

Slice-view atom maps of Ag, In, Ga, Zn, and S, illustrating their internal spatial distributions. Although most Ag and In atoms are in the core regions, a significant amount of Ga is distributed over the shell region. Zn atoms predominantly exist in the outer shell. **c** Ag and Zn iso-surface and proxigram obtained from Ag iso-surfaces; the proxigram was obtained from five QDs. The Ag and In contents increase toward the QD core, while the content of Zn increases toward the outer shell of the QD. A significant amount of Ga also appears in the outer shell. **d** 2D compositional profile extracted from a cylindrical volume ($3 \times 3 \times 21 \text{ nm}^3$) through the 3D reconstruction, confirming the core/shell/shell structure. (Scale bars: 2 nm in **a–d**). The slight variations in the compositional distributions from QD to QD arise from the spatial offsets of individual QDs within the APT specimen. Error bars in **c**, **d** indicate one-sigma counting statistics.

***Comment 6.** There is a fundamental contradiction in the manuscript's description of the final shell. The text frequently refers to "synthesizing the ZnS shell", "ZnS shell coating", or "shell growth". However, two key pieces of the authors' own data refute this:*

- TEM Data: The authors state the QDs "grow only negligibly" and remain "similar in both size and shape" after the ZnS precursor step. This is inconsistent with the epitaxial growth of a new shell.*
- APT Data: The authors' own excellent APT analysis proves the process is cation exchange, forming a $\text{Zn}_{1-3/2x}\text{Ga}_x\text{S}$ shell.*

The authors must revise the manuscript to consistently describe this process as cation exchange, as their own evidence strongly contradicts the narrative of simple "shell growth."

Answer to Comment 6: We agree with the reviewer that our experimental evidence clearly and consistently supports cation exchange as the dominant mechanism for shell formation rather than the epitaxial growth of a new ZnS shell. As correctly pointed out, the negligible change in quantum dot size and shape observed by TEM, together with the APT results demonstrating the formation of a $\text{Zn}_{1-3/2x}\text{Ga}_x\text{S}$ shell, is inconsistent with conventional epitaxial shell growth. Accordingly, we have revised the manuscript to consistently describe this process as cation exchange throughout the text and have removed or rephrased terminology implying simple ZnS shell growth or coating.

The manuscript has been revised as follows:

“synthesizing the ZnS shell”, “ZnS shell coating”, “shell growth”

→ “ZnS precursor step”, “during the ZnS precursor step to form an outer shell”, “after the ZnS precursor step for the outer shell formation”

Comment 7. The authors claim that the final $Zn_{1-3/2x}Ga_xS$ shell "maintains the charge carrier confinement with a Type-I band alignment". This is a critical conclusion for the QD's performance, but it is not experimentally proven. This claim appears to be an inference based on the band gaps of related bulk materials. Given the novel, complex, and compositionally graded structure revealed by APT, such an assumption is not sufficient. The authors must provide spectroscopic evidence (e.g., time-resolved PL or other measurements) to conclusively demonstrate the Type-I alignment. If such evidence is unavailable, this claim must be significantly toned down and presented as a well-reasoned hypothesis, not a confirmed fact.

Answer to Comment 7: We thank the reviewer for this important comment regarding the band alignment of the $Zn_{1-3/2x}Ga_xS$ shell. We agree that the original wording may have overstated the level of experimental certainty, and we have revised the manuscript to present the Type-I-like carrier confinement as a well-reasoned interpretation rather than a conclusively demonstrated fact.

Direct spectroscopic determination of the band alignment or bandgap of the compositionally graded $Zn_{1-3/2x}Ga_xS$ shell itself is experimentally challenging in the present system. While time-resolved photoluminescence (TR-PL) measurements were performed, the decay dynamics did not allow shell-related decay components to be clearly distinguished. Nevertheless, our interpretation can be informed by indirect experimental observations as well as prior studies on Zn–Ga–S compounds and quantum dots (QDs). Although there are no reports directly measuring the bandgap of a $Zn_{1-3/2x}Ga_xS$ shell layer, previous studies on $Zn_{1-3/2x}Ga_xS$ QDs and related compounds indicate that the bandgap energy of this system lies between that of Ga_2S_3 (~2.5 eV) and ZnS (~3.5 eV), depending on the cation ratio. In particular, $ZnGa_2S_4$ exhibits a bandgap of approximately ~3.18 eV, and literature reports show that $ZnGa_2S_4/ZnS$ heterostructured QDs form a Type-I band alignment, whereas $ZnGa_2S_4/Ga_2S_3$ QDs do not. In addition, although not in the form of QDs, previous studies have reported that nanocrystalline $ZnGa_2S_4$ films exhibit a wide bandgap in the range of approximately 3.75–4.11 eV, classifying them as wide-bandgap semiconductors.

Based on prior studies showing that Zn–Ga–S–based materials generally possess wider bandgaps than $AgInS_2$ -based cores, the $Zn_{1-3/2x}Ga_xS$ shell formed in the present work is expected to provide Type-I-like carrier confinement. This interpretation is further supported by several experimental observations: (i) the quantum yield (QY) remains high (80–90%) after cation exchange and increases to ~92% following the ZnS precursor step, and (ii) the QDs exhibit markedly enhanced ambient-condition stability in QD–acrylate composite films at room temperature. In contrast, a Type-II alignment would typically be associated with reduced radiative efficiency, pronounced red-shifts, and QY degradation upon shell formation, none of which are observed in the present system. Accordingly, we now describe the $Zn_{1-3/2x}Ga_xS$ shell as consistent with or suggestive of Type-I carrier confinement, rather than claiming definitive experimental proof. The revised manuscript has been carefully edited to reflect this interpretation and to avoid overstating the certainty of the band alignment.

The manuscript has been revised as follows:

Please see Page #4.

“Interestingly, during the ZnS precursor step to form an outer shell on the $AgIn_xGa_{1-x}S_2/AgGaS_2$ QDs, a $Zn_{1-3/2x}Ga_xS$ outer shell spontaneously forms instead, driven by Ag^+ and Ga^{3+} to Zn^{2+} cation exchange.

This $Zn_{1-3/2x}Ga_xS$ shell exhibits excellent structural and chemical compatibility with the $AgGaS_2$ inner shell and is expected to induce a Type-I band alignment.”

Please see Page #9,

“Nevertheless, the resulting $AgIn_yGa_{1-y}S_2$ shell is expected to effectively confine charges within the $AgIn_xGa_{1-x}S_2$ core owing to its higher band gap, consistent with a Type-I heterostructure, which promotes radiative recombination.”

Please see Page #16,

“In terms of the band gap energy, the $Zn_{1-3/2x}Ga_xS$ shell typically exhibits a slightly lower band gap than ZnS (~3.5 eV), even at the nanoscale.^{62,65,69-72} The formation of the compositionally mixed $Zn_{1-3/2x}Ga_xS$ layer is therefore expected to adequately preserve the quantum efficiency, with negligible weakening of charge confinement, consistent with Type-I-like band alignment. Moreover, the spontaneous formation of the $Zn_{1-3/2x}Ga_xS$ shell could significantly reduce crystallographic or interfacial defects that might arise from the structural differences between the inner and outer shells. By creating a more seamless interface with the $AgGaS_2$ inner shell, the $Zn_{1-3/2x}Ga_xS$ shell may enhance the stability of the QDs under ambient conditions and contribute to a more stable and efficient green luminescence. This optical stability is particularly advantageous for optoelectronic applications where long-term emission reliability is critical.”

References have been added as follows:

71. Hu, T., Zhu, K., Cheng, H., Teng, Y. & Pan, Z. Core-shell energy band engineering of cyan light-emitting ternary $ZnGa_2S_4@ZnS$ quantum dots toward anti-counterfeiting and bioimaging applications. *J. Mater. Chem. C* **13**, 21797-21811 (2025). [10.1039/D5TC02797G](https://doi.org/10.1039/D5TC02797G).
72. Yadav, A. N. & Singh, K. Investigation of photophysical properties of ternary Zn-Ga-S quantum dots: band gap versus sub-band-gap excitations and emissions. *ACS Omega* **4**, 18327-18333 (2019). [10.1021/acsomega.9b02546](https://doi.org/10.1021/acsomega.9b02546).

Reviewer #2 (Remarks to the Author):

General Comment: The present manuscript simply reported the measurement of AgInGaS QDs with atom probe tomography (APT). Although atom resolution images were obtained, no relationship between the nanostructure of QDs and their photochemical properties were reported. Thus, no novel results enough to publish in highly ranked journals were contained in the present manuscript. Especially, following points were very weak.

Answer to General Comment: We thank the reviewer for the constructive comment. We acknowledge that, in the original version of the manuscript, the broader significance of this study and its connection to photochemical properties were not sufficiently articulated.

The primary novelty of this work lies in the first direct, atomically resolved analysis of 3D nanoscale compositional distributions in quaternary $\text{AgIn}_x\text{Ga}_{1-x}\text{S}_2$ quantum dots (QDs) using APT. Our APT analysis reveals previously inaccessible structural features arising during synthesis, including the spontaneous formation of a $\text{Zn}_{1-3/2x}\text{Ga}_x\text{S}$ outer shell via cation exchange, the development of Ag-deficient shell regions governed by precursor reactivity, and the presence of compositional gradients and nanoscale inhomogeneities within individual QDs.

In response to the reviewer's comment, we have revised the manuscript to more explicitly establish the relationship between these nanoscale structural features and the photochemical properties of the QDs. Specifically, we have added a detailed discussion on the evolution of quantum yield and stability associated with the formation of the $\text{Zn}_{1-3/2x}\text{Ga}_x\text{S}$ shell, as well as the origin of the observed long-wavelength emission. In addition, new experimental evidence has been included to demonstrate how variations in the Ag/Ga precursor ratio affect shell formation behavior and associated optical properties, and to experimentally substantiate the compositional inhomogeneity among individual QDs. Furthermore, relevant literature has been incorporated to support the reliability and validity of our experimental approach and APT analysis. We therefore believe that the revised manuscript goes beyond a simple structural characterization study and provides new insight into how nanoscale compositional evolution during synthesis influences the optical behavior of complex quaternary QDs, which is directly relevant to their rational design and optimization.

Comment 1. As mentioned in the introduction, the authors claimed that they three-dimensionally characterize quaternary QDs ($\text{AgIn}_x\text{Ga}_{1-x}\text{S}_2$ -based heterostructured QDs with core/shell and core/shell/shell structures in this case) on the atomic scale for the first time using APT. If so, the authors must show at first the reliability of the APT method for measurements of QD nanostructures, using well known QDs, such as CdSe QDs and CdSe/ZnS core-shell structured QDs.

Answer to Comment 1: We thank the reviewer for the comment regarding the reliability of APT for quantum dot (QD) nanostructures.

The applicability and reliability of APT for the 3D compositional analysis of colloidal QDs have been previously demonstrated using well-known CdSe-based systems in our previous work (Chae et al. ACS Nano, 2018). In that study, we reported APT analyses of commercial CdSe/ZnS core/shell QDs and systematically addressed specimen preparation strategies, reconstruction considerations, and potential artifacts relevant to APT analysis of colloidal QDs. We further discussed the representativeness and reliability of the resulting datasets. The elemental distributions obtained by APT were directly compared

with transmission electron microscopy (TEM) results, demonstrating that APT can reliably resolve core/shell architectures and provide compositional information that is complementary to, and in some aspects beyond, what is accessible by TEM. Based on this prior validation using CdSe/ZnS core/shell QDs, we consider the methodological reliability of APT for QD nanostructures to be well supported. In the revised manuscript, we now explicitly reference this previous work and discuss known APT-related artifacts relevant to nanostructured materials, together with justification for the reliability of the present measurements.

It is well recognized that APT analyses of nanostructures can be affected by artifacts such as local magnification effects and trajectory aberrations, which may lead to artificially broadened interfaces and apparent intermixing. These effects have been systematically investigated in prior studies. In particular, compositional profiles extracted parallel to nominal interfaces often appear artificially diffuse due to trajectory overlap, whereas compositional analyses performed perpendicular to the interface—corresponding to the analysis (z) direction in APT—have been shown to provide superior reliability in resolving compositional distributions. This behavior has been widely reported for metallic and semiconductor nanostructures, where reduced trajectory overlap preserves more reliable compositional distributions.

In the present study, compositional trends were evaluated using multiple APT analysis approaches, including proxigrams and compositional profiles extracted along different spatial directions. While compositional profiles obtained along the analysis (z) direction—generally considered to provide more reliable compositional distribution in APT—serve as an important reference, the trends observed in lateral (x/y) profiles are qualitatively consistent. Although minor variations in profile broadening are observed depending on the analysis direction, all datasets reveal the same overall compositional tendencies.

To further support the reliability of the APT analysis, we performed additional complementary characterization beyond APT. X-ray photoelectron spectroscopy (XPS) measurements further corroborated the formation of a Ga-containing outer shell after the ZnS precursor step. Although XPS does not provide 3D spatial information, we compared the relative Ag and Ga signals before and after the ZnS shelling process using multiple core-level regions, including Ag *MNN* vs Ga *2p* and Ag *4d* vs Ga *3d*, each of which occupy equivalent binding energy ranges. In all cases, the Ga-to-Ag ratios were consistently higher in the $\text{AgIn}_x\text{Ga}_{1-x}\text{S}_2/\text{AgGaS}_2/\text{ZnS}$ QDs than in the $\text{AgIn}_x\text{Ga}_{1-x}\text{S}_2/\text{AgGaS}_2$ QDs, suggesting the presence of substantial Ga-related components in the outermost shell region after the ZnS precursor step. These XPS results are consistent with the $\text{Zn}_{1-3/2x}\text{Ga}_x\text{S}$ shell formation revealed by APT. In addition, Cl-related signals were detected only after the ZnS precursor step, implying the incorporation of Zn–Cl-related species. TEM further revealed structural and compositional features consistent with the APT observations. Our TEM results show both the overall similarity in elemental distribution trends and the presence of particle-to-particle variability observed by APT. These results provide additional confidence that the APT reconstructions are consistent with the underlying nanostructure. Corresponding clarifications and references have been added to the revised manuscript.

The manuscript has been revised as follows:

Please see Page #7.

“To characterize and quantify the 3D atomic distribution of the $\text{AgIn}_x\text{Ga}_{1-x}\text{S}_2$ -based core/shell structured QDs, APT was employed. For the APT analysis in this work, dried QDs were prepared in powder form

and analyzed following our previously described method.⁴¹ In that study, we demonstrated that APT could faithfully resolve core/shell architectures and 3D compositional distributions in commercially available CdSe/ZnS core/shell QDs, in good agreement with TEM observations. Specimen preparation strategies, potential APT artifacts, and reconstruction considerations specific to CdSe/ZnS core/shell QDs were systematically examined, thereby establishing the applicability of APT to colloidal QD nanostructures. More broadly, the feasibility of APT in the analysis of metallic nanoparticles has been demonstrated in numerous prior studies, including ligand-resolved analyses and combined experiment–simulation approaches.⁴²⁻⁴⁷ On this basis, we report a 3D atomic-scale characterization of $\text{AgIn}_x\text{Ga}_{1-x}\text{S}_2$ -based QDs using APT, which, to the best of our knowledge, represents the first such analysis of quaternary QDs.”

Please see Page #8,

“Figure 2a shows the 3D reconstructed APT atom maps of the $\text{AgIn}_x\text{Ga}_{1-x}\text{S}_2/\text{AgGaS}_2$ QDs, representing a hemispherical APT specimen containing five QDs (see Supplementary Fig. 6 for dashed guides indicating the approximate outlines of individual QDs). The QDs appear as slightly ellipsoidal features in the APT reconstruction, and a slice-view image along the x – z plane is presented in Fig. 2b to visualize their internal elemental distribution. We note that APT analyses of nanostructures may exhibit reconstruction artifacts, such as apparent elongation along the analysis (z) direction and interfacial broadening, primarily arising from local magnification and trajectory aberrations.⁴²⁻⁵² Accordingly, the slightly ellipsoidal morphology observed here is attributed to such effects rather than the intrinsic particle shape. However, compositional distributions along the z -direction have been shown to provide more reliable quantitative information than lateral directions because the reduced trajectory overlap suppresses artificial intermixing.^{42-44,51,52} Under these conditions, compositional distributions and interfacial characteristics closer to the intrinsic distributions can be more faithfully evaluated.”

Please see Page #9,

"The 2D compositional profiles along two different directions (Figs. 3c and 3f) further confirm the formation of the $\text{AgIn}_x\text{Ga}_{1-x}\text{S}_2$ core/ AgGaS_2 shell heterostructure with a compositional gradient. From the extracted profiles, we primarily focus on z -direction compositional analyses, cross-validated by

proxigrams and lateral profiles. Ag and In atoms are highly concentrated at the center of the QD, at more than 20 and 10 at.%, respectively, but their concentrations gradually decrease toward the shell.”

Please see Page #12,

“Figures 5a and 5d show the extracted 3D atom maps and the cross-sectional regions cropped by the green boxes for each QD. According to the z-direction compositional profiles cross-validated by proxigrams and lateral profiles, Ag atoms are more widely distributed than In atoms in the 3D atom maps (Figs. 5b and 5e). However, both Ag and In atoms are at their highest concentrations in the central region of the QDs, with values exceeding 20 and 10 at.%, respectively.”

Please see Page #13,

“Because APT inherently probes only a limited number of individual QDs and may therefore suffer from limited statistical representativeness, the compositional features identified by APT were further examined using complementary techniques. XPS analyses performed on the same samples reveal consistently higher Ga-to-Ag ratios after the Zn precursor step, evaluated from multiple core-level regions, including Ag *MNN* vs Ga *2p* and Ag *4d* vs Ga *3d*. This systematic increase in the Ga-to-Ag ratios supports the formation of Ga-containing outer shells and is consistent with the $Zn_{1-3/2x}Ga_xS$ shell formation revealed by APT (Supplementary Fig. 15 and Supplementary Table 2). Inductively coupled plasma analyses performed on the same samples show overall elemental compositions consistent with the APT results (Supplementary Tables 3 and 4). Taken together, these results indicate that the structural and compositional features identified by APT are not limited to a small number of individual QDs but are representative of overall sample.”

Please see Page #14,

“Consistent with the APT observations, STEM–EDS line profiles collected from multiple QDs reveal noticeable particle-to-particle variations in the relative Ag, In, and Ga distributions (Supplementary Figs. 16 and 17). For each QD, several independent line profiles were extracted and compared, all of which show dot-to-dot compositional fluctuations. These results indicate that the observed compositional inhomogeneity is an intrinsic feature of the synthesized quaternary QDs, thereby supporting the features revealed by APT.”

Please see Page #21.

“APT analysis

APT analysis was conducted using a CAMECA atom probe (Invizo 6000) operating in the deep-UV laser mode ($\lambda = 257.5$ nm) with a pulse repetition rate of 200 kHz and a base temperature of 50 K. A laser pulse energy of 100 pJ was applied, achieving a detection rate of 0.01 atoms pulse⁻¹. The acquired data were reconstructed and analyzed using CAMECA AP Suite 6.3 software. All elemental species constituting the AgIn_xGa_{1-x}S₂-based QDs, as well as C-related ions, were identified from the mass spectra and assigned before reconstruction. The reconstructions were performed using a shank-angle-based protocol. SEM and HAADF-STEM images were utilized to guide the reconstruction and constrain the specimen geometry, thereby enhancing the accuracy of the APT reconstruction process. No artificial geometric constraints were imposed (e.g., forcing spherical particle shapes) during the reconstruction process. The slightly ellipsoidal morphologies observed in the APT datasets were therefore preserved as reconstructed rather than being artificially forced into an idealized geometry and are attributed to known trajectory aberration and local magnification effects.”

Please see Supplementary Fig. 15.

Supplementary Fig. 15 XPS analysis results of AgIn_xGa_{1-x}S₂-based QDs. XPS spectra of **a** Ga 2p, **b** Zn 2p_{3/2}, **c** In 3d, **d** Ag 3d, and **e** S 2s core levels acquired from AgIn_xGa_{1-x}S₂ (AIGS), AgIn_xGa₁₋

$xS_2/AgGaS_2$ (AIGS/AGS), and $AgIn_xGa_{1-x}S_2/AgGaS_2/ZnS$ (AIGS/AGS/ZnS) QDs using an Al K-alpha X-ray source. All QDs contain Ag–S, In–S, Ga–S, and Zn–S chemical bonds, as designed without detectable oxidation. f Valence band spectra of AIGS, AIGS/AGS, and AIGS/AGS/ZnS QDs. The Ga-to-Ag ratio is consistently higher in the AIGS/AGS/ZnS QDs than in the AIGS/AGS QDs. The detailed compositions are provided in **Supplementary Table 2**. These results further support the presence of a Ga-containing outer shell, consistent with the $Zn_{1-3/2x}Ga_xS$ shell identified by APT after the ZnS precursor step.

Please see Supplementary Fig. 16.

Supplementary Fig. 16 Detailed elemental distributions of $AgIn_xGa_{1-x}S_2/AgGaS_2$ QDs measured by STEM–EDS. STEM image and corresponding EDS line profiles of individual $AgIn_xGa_{1-x}S_2/AgGaS_2$ QDs. Although the QDs show similar compositional trends, the elemental compositions vary from particle to particle. Owing to the projected nature and limited spatial resolution of STEM–EDS, these data provide qualitative compositional trends rather than a definitive 3D core/shell/shell assignment. (Scale bars: 7 nm.)

Please see Supplementary Fig. 17,

Supplementary Fig. 17 Detailed elemental distributions of $\text{AgIn}_x\text{Ga}_{1-x}\text{S}_2/\text{AgGaS}_2/\text{ZnS}$ QDs measured by STEM–EDS. STEM image and corresponding EDS line profiles of individual $\text{AgIn}_x\text{Ga}_{1-x}\text{S}_2/\text{AgGaS}_2/\text{ZnS}$ QDs. Zn is distributed over a broader region than Ag and In, indicating the formation of a Zn-containing outer region. Although the QDs show similar overall compositional tendencies, the elemental compositions vary among individual QDs, indicating particle-to-particle variability. Owing to the projected nature and limited spatial resolution of STEM–EDS, these data provide qualitative compositional trends rather than a definitive 3D core/shell/shell assignment. (Scale bars: 7 nm.)

Please see Supplementary Table 2,

Specimen	Atomic composition (%) (XPS)								
	C (in C 1s)	O (in O 1s)	F (in F 1s)	S (in S 2s)	Cl (in Cl 2p)	Zn (in Zn 2p _{3/2})	Ga (in Ga 2p _{3/2})	Ag (in Ag 3d)	In (in In 3d _{5/2})
AIGS	76.29	1.91	0	11.54	0	0	1.1	4.46	4.7
AIGS/AGS	69.37	2.83	0	14.69	0.23	0	5.16	5.9	1.83

AIGS/AGS/ZnS	62.92	6.58	0	13.93	2.07	6.07	4.54	2.85	1.06
-------	------	---	-------	------	------	------	------	------

Specimen	Atomic composition (%) (XPS)			
	S (in S 2s)	Ga (in Ga 2p _{3/2})	Ag (in Ag 3d)	In (in In 3d _{5/2})
AIGS	52.92	5.07	20.44	21.57
AIGS/AGS	53.25	18.71	21.41	6.63
AIGS/AGS/ZnS	62.28	20.28	12.73	4.72

Specimen	Atomic composition (%) (XPS)			
	Ag (in Ag 4d)	Zn (in Zn 3d)	In (in In 4d)	Ga (in Ga 3d)
AIGS	36.1	0	37.7	26.2
AIGS/AGS	37.1	0	10.0	52.9
AIGS/AGS/ZnS	21.3	30.5	5.1	43.1

Supplementary Table 2. Atomic compositions of AgIn_xGa_{1-x}S₂/AgGaS₂ (AIGS/AGS) and AgIn_xGa_{1-x}S₂/AgGaS₂/ZnS (AIGS/AGS/ZnS) QDs measured by XPS. The Ga-to-Ag ratio is consistently higher in the AIGS/AGS/ZnS QDs than in the AIGS/AGS QDs, suggesting the formation of a Ga-containing outer shell. This trend is observed both when comparing Ag *MNN* and Ga *2p* and when comparing Ag *4d* and Ga *3d*, all of which lie in similar binding energy ranges.

References have been added as follows:

41. Chae, B. G. et al. Direct three-dimensional observation of core/shell-structured quantum dots with a composition-competitive gradient. *ACS Nano* **12**, 12109-12117 (2018). [10.1021/acsnano.8b05379](https://doi.org/10.1021/acsnano.8b05379).
42. Jang, K. et al. Three-dimensional atomic mapping of ligands on palladium nanoparticles by atom probe tomography. *Nat. Commun.* **12**, 4301 (2021). [10.1038/s41467-021-24620-9](https://doi.org/10.1038/s41467-021-24620-9).
43. Kim, S. H. et al. Characterization of Pd and Pd@Au core-shell nanoparticles using atom probe tomography and field evaporation simulation. *J. Alloys Compd.* **831**, 154721 (2020).

[10.1016/j.jallcom.2020.154721](https://doi.org/10.1016/j.jallcom.2020.154721).

44. Tedsree, K. et al. Hydrogen production from formic acid decomposition at room temperature using a Ag–Pd core–shell nanocatalyst. *Nat. Nanotechnol.* **6**, 302-307 (2011). [10.1038/nnano.2011.42](https://doi.org/10.1038/nnano.2011.42).
45. Grenier, A. et al. 3D analysis of advanced nano-devices using electron and atom probe tomography. *Ultramicroscopy* **136**, 185-192 (2014). [10.1016/j.ultramic.2013.10.001](https://doi.org/10.1016/j.ultramic.2013.10.001).
46. Khan, M. A., Ringer, S. P. & Zheng, R. Atom probe tomography on semiconductor devices. *Adv. Mater. Interfaces* **3**, 1500713 (2016). [10.1002/admi.201500713](https://doi.org/10.1002/admi.201500713).
47. Hatzoglou, C., Radiguet, B. & Pareige, P. Experimental artefacts occurring during atom probe tomography analysis of oxide nanoparticles in metallic matrix: quantification and correction. *J. Nucl. Mater.* **492**, 279-291 (2017). [10.1016/j.jnucmat.2017.05.008](https://doi.org/10.1016/j.jnucmat.2017.05.008).
48. Philippe, T., Gruber, M., Vurpillot, F. & Blavette, D. Clustering and local magnification effects in atom probe tomography: a statistical approach. *Microsc. Microanal.* **16**, 643-648 (2010). [10.1017/S1431927610000449](https://doi.org/10.1017/S1431927610000449).
49. Lawitzki, R., Stender, P. & Schmitz, G. Compensating local magnifications in atom probe tomography for accurate analysis of nano-sized precipitates. *Microsc. Microanal.* **27**, 1–12 (2021). [10.1017/S1431927621000180](https://doi.org/10.1017/S1431927621000180).
50. Beinke, D., Oberdorfer, C. & Schmitz, G. Towards an accurate volume reconstruction in atom probe tomography. *Ultramicroscopy* **165**, 34-41 (2016). [10.1016/j.ultramic.2016.03.008](https://doi.org/10.1016/j.ultramic.2016.03.008).
51. Takahashi, J. & Kawakami, K. Position artifacts in 3D reconstruction of plate-shaped precipitates in steels depending on the analysis direction of atom probe tomography. *Surf. Interface Anal.* **53**, 982-995 (2021). [10.1002/sia.7001](https://doi.org/10.1002/sia.7001).
52. Maruyama, N., Smith, G. D. W. & Cerezo, A. Interaction of the solute niobium or molybdenum with grain boundaries in α -iron. *Mater. Sci. Eng. A* **353**, 126-132 (2003). [10.1016/S0921-5093\(02\)00678-0](https://doi.org/10.1016/S0921-5093(02)00678-0).”

Comment 2. No relationship between the nanostructure of QDs and their photochemical properties were reported.

Answer to Comment 2: We thank the reviewer for this important comment. We agree that establishing clear correlations between the nanostructure of the QDs and their photochemical properties is essential, and we have made substantial efforts to strengthen this aspect in the revised manuscript.

In the original version, our primary focus was on directly resolving, at the 3D atomic scale, several previously inaccessible structural features in this quaternary system, including the unexpected formation of a $\text{Zn}_{1-3/2x}\text{Ga}_x\text{S}$ outer shell via cation exchange, particle-to-particle compositional variations, compositional gradients, and the retention of Cl-related species originating from the ZnS precursor step. We also briefly discussed how such nanoscale features could influence QD properties. This represents an important methodological and conceptual advance, given the analytical challenges and limited prior reports on quaternary I–III–VI₂ QDs. Nevertheless, we agree that it is critical to explicitly correlate these nanostructural characteristics with the photochemical behavior of the QDs. Accordingly, in response to the reviewer's comment, we have now significantly expanded both the experimental evidence and the discussion linking the resolved nanostructures to the observed photochemical properties.

First, we clarify that despite the formation of a $\text{Zn}_{1-3/2x}\text{Ga}_x\text{S}$ shell instead of an ideal ZnS shell, the QDs exhibit both an increased photoluminescence quantum yield (QY) from 85% to 92% after the ZnS precursor step and improved ambient-condition stability in QD–acrylate composite films (Fig. 1 and Supplementary Fig. 1). Based on prior studies on Zn–Ga–S compounds and QDs, which indicate that Zn–Ga–S systems possess bandgap energies between those of Ga_2S_3 and ZnS depending on composition and can form Type-I heterostructures (e.g., $\text{ZnGa}_2\text{S}_4/\text{ZnS}$), the $\text{Zn}_{1-3/2x}\text{Ga}_x\text{S}$ shell formed here is expected to have a wider bandgap than the $\text{AgIn}_x\text{Ga}_{1-x}\text{S}_2$ core or AgGaS_2 shell, making Type-I-like carrier confinement energetically plausible. Consistent with this interpretation, our QDs retain high QYs (80–90%) and show a further increase to ~92% following the ZnS precursor step. Moreover, they exhibit markedly enhanced ambient-condition stability, as evidenced by the significantly improved retention of QY after 48 h relative to the initial value. In contrast, a Type-II alignment would typically be associated with reduced radiative efficiency and QY degradation, which are not observed in the present system. These results and their interpretation have now been explicitly incorporated into the revised manuscript.

Second, we substantially expanded the discussion of the long-wavelength tail emission that emerges after the ZnS precursor step. PL characteristics show that the trap-related emission contribution increases from ~3.7% in $\text{AgIn}_x\text{Ga}_{1-x}\text{S}_2/\text{AgGaS}_2$ QDs to ~14.9% in $\text{AgIn}_x\text{Ga}_{1-x}\text{S}_2/\text{AgGaS}_2/\text{ZnS}$ QDs (the trap emission area ratio was defined as the fraction of the integrated photoluminescence intensity in the long-wavelength region ($\lambda > \lambda_{\text{PL,max}} + 50$ nm) to the total integrated emission.). Importantly, XRD and TEM confirm that the shell retains a chalcopyrite-related tetragonal structure rather than forming a new zinc-blende-type ZnS heterophase. This demonstrates that the observed tail emission does not originate from the formation of a structurally distinct shell, but is instead associated with defect formation. Consistent with this interpretation, our APT results directly reveal a complex, four-layered nanostructure consisting of an Ag- or In-rich $\text{AgIn}_x\text{Ga}_{1-x}\text{S}_2$ core/Ga-rich AgGaS_2 inner shell/Ag-deficient AgGaS_2 intermediate shell/Zn- and Ga-rich $\text{Zn}_{1-3/2x}\text{Ga}_x\text{S}$ outer shell. After the ZnS precursor

step, APT further shows inward diffusion of Zn^{2+} into the QDs. Because excitons in I–III–VI₂ QDs are preferentially confined to In-rich regions, Zn diffusion and cation exchange in the vicinity of these regions are expected to generate a high density of cation vacancies and related complexes. Such intrinsically non-charge-neutral substitution processes are well known to introduce intragap trap states responsible for donor–acceptor-pair-like recombination, leading to the long-wavelength emission in I–III–VI₂ QDs. We therefore attribute the enhanced tail emission to ~14.9% after the ZnS precursor step primarily to defect formation induced by Zn inward diffusion and cation exchange, rather than to the emergence of a new heterostructural phase. Relevant literature on Ag–In–S and Zn-alloyed Ag–In–S QDs reporting similar trap-assisted long-wavelength emission and our explanation has now been incorporated into the revised manuscript.

Third, to further link nanostructure formation to optical properties, we performed additional controlled syntheses in which the Ag and Ga precursor amounts were systematically varied. When the Ga precursor amount was fixed and the Ag precursor amount was increased, STEM–EELS revealed pronounced thickening of the AgGaS₂ shell and increased Ag incorporation. At the same time, the trap-related emission contribution increased from 5.1% to 21.3% as the Ag precursor amount was increased up to sixfold, while the photoluminescence QY decreased from 77% to 37%, indicating that Ag-driven shell growth could be accompanied by enhanced defect formation (Supplementary Fig. 8 and Supplementary Table. 2). In contrast, when the Ag precursor amount was fixed and the Ga precursor amount was increased, no significant increase in QD size or shell thickness was observed. This asymmetric behavior provides compelling experimental support for our interpretation that the formation of the Ag-deficient outer shell originates from the relatively lower reactivity of the Ga precursor and directly impacts the defect and optical response of the QDs. This interpretation and its implications for the optical properties have now been incorporated into the revised manuscript.

Taken together, these new experiments and expanded discussions explicitly connect the 3D nanostructure revealed by APT to the observed photochemical and optical properties, including QY, stability, and trap-related emission. The corresponding data, analyses, and discussions have been incorporated into the revised manuscript and Supplementary Information.

The manuscript has been revised as follows:

Please see Page #5,

“Notably, the QY increases from 85% to 92% after the ZnS precursor step. Furthermore, the stability of QD–acrylate composite films under ambient conditions was evaluated in terms of the change in the QY after 48 h with respect to the initial QY. As shown in Supplementary Fig. 1, the ZnS shell significantly improves this normalized QY from 66% to 97%. The enhanced optical properties can be attributed to the effective confinement of electrons and holes provided by the wide-band-gap ZnS encapsulating the AgIn_xGa_{1-x}S₂/AgGaS₂. In addition to the dominant green emission, the AgIn_xGa_{1-x}S₂/AgGaS₂/ZnS QDs exhibit a long-wavelength emission tail in the 600–700 nm range (Fig. 1d). The area ratio of this long-wavelength emission, defined as the fraction of the total PL intensity in the long-wavelength region (\$\lambda > \lambda_{\text{PL max}} + 50 \text{ nm}\$ ), increases from ~3.7% in AgIn_xGa_{1-x}S₂/AgGaS₂ QDs to ~14.9%

in $\text{AgIn}_x\text{Ga}_{1-x}\text{S}_2/\text{AgGaS}_2/\text{ZnS}$ QDs. Such long-wavelength emission is generally attributed to defect- or trap-assisted recombination rather than band-edge excitonic transitions in AgInS_2 -based QDs and related I–III–VI₂-based QDs.³⁶⁻⁴⁰»

Please see Page #10.

“To demonstrate the role of differential precursor reactivity, we performed controlled syntheses in which the Ag precursor amount was increased at a fixed Ga level, a condition under which increasing the Ag precursor amount directly promotes the further growth of the AgGaS_2 shell. STEM–EELS analyses showed the pronounced thickening of the AgGaS_2 shell and consistently increased Ag incorporation, supporting the role of precursor reactivity in the formation of the shell (Supplementary Fig. 8 and Supplementary Table 1). These changes were accompanied by an increased contribution from trap-related emission and a decreased photoluminescence QY, indicating that Ag-driven shell growth directly impacts the optical properties of the QDs.”

Please see Page #16.

“The emergence of the $\text{Zn}_{1-3/2x}\text{Ga}_x\text{S}$ outer shell via cation exchange offers a plausible origin for the long-wavelength emission tail observed after the ZnS precursor step. In our system, the substitution of Ag^+ and Ga^{3+} by Zn^{2+} during shell formation inherently introduces cation vacancies and local charge-compensating defects. Consistent with our APT observations, the inward diffusion of Zn^{2+} into the interior of the QDs further induces cation exchange and promotes the formation of such defects. When these defects form in the vicinity of In-rich regions, where carriers are preferentially confined, they can introduce intragap trap states that locally perturb the confinement potential and promote trap-assisted recombination. These defect-related states are commonly associated with donor–acceptor-pair-like recombination pathways that give rise to long-wavelength emission in I–III–VI₂ semiconductors.^{23,38-40,66-68}»

Please see Page #16.

“In terms of the band gap energy, the $\text{Zn}_{1-3/2x}\text{Ga}_x\text{S}$ shell typically exhibits a slightly lower band gap than ZnS (~3.5 eV), even at the nanoscale.^{62,65,69-72} The formation of the compositionally mixed $\text{Zn}_{1-3/2x}\text{Ga}_x\text{S}$

layer is therefore expected to adequately preserve the quantum efficiency, with negligible weakening of charge confinement, consistent with Type-I-like band alignment.”

Please see Fig. 1,

Fig. 1 Comparison of $\text{AgIn}_x\text{Ga}_{1-x}\text{S}_2/\text{AgGaS}_2$ QDs and $\text{AgIn}_x\text{Ga}_{1-x}\text{S}_2/\text{AgGaS}_2/\text{ZnS}$ QDs. a, d Schematic representation of the internal structure, band diagrams, and UV–vis absorption, and PL characteristics. Both QDs exhibit bright green luminescence at ~ 530 nm, with the QY increasing from 85% to 92% after the ZnS precursor step to form the outer shell. In addition, $\text{AgIn}_x\text{Ga}_{1-x}\text{S}_2/\text{AgGaS}_2/\text{ZnS}$ QDs exhibit a long-wavelength emission tail in the ~ 600 – 700 nm range. b, e HAADF–STEM image of $\text{AgIn}_x\text{Ga}_{1-x}\text{S}_2$ -based QDs before and after the ZnS precursor step, which negligibly changed the size of the QDs, with average diameters of 6.14 ± 0.76 and 6.11 ± 1.03 nm, respectively. Inset: high-magnification HAADF–STEM image. c, f STEM–EDS elemental maps of $\text{AgIn}_x\text{Ga}_{1-x}\text{S}_2$ -based QDs before and after the ZnS precursor step.

Please see Supplementary Fig.8,

Supplementary Fig. 8 STEM–EELS analysis of $\text{AgIn}_x\text{Ga}_{1-x}\text{S}_2/\text{AgGaS}_2$ QDs as a function of the Ga/Ag precursor ratio. (a) Representative STEM–EELS results for high-Ga/Ag-ratio (Ag-poor) $\text{AgIn}_x\text{Ga}_{1-x}\text{S}_2/\text{AgGaS}_2$ QDs and low-Ga/Ag-ratio (Ag-rich) $\text{AgIn}_x\text{Ga}_{1-x}\text{S}_2/\text{AgGaS}_2$ QDs. As the Ag precursor amount increases, the overall diameter of the QDs increases, particularly because of the increasing thickness of the AgGaS_2 shell. Owing to the relatively low reactivity of the Ga precursor, increasing the Ag precursor under Ga-rich conditions mainly promotes the growth of the AgGaS_2 shell, accompanied by increased Ag incorporation. (Scale bars: 5 nm.)

Please see Supplementary Table 1.

Ag precursor amount	Ga/Ag (precursor ratio)	$\lambda_{\text{PL max}}$ (band edge)	FWHM (nm)	Trap emission area ratio	QY	QD size measured by TEM (nm)
Low Ag	29	524	35	5.1	77	6.0
	10	520	37	7.8	68	6.8
↓	6	517	37	12.9	57	6.9
High Ag	4	516	38	21.3	37	7.9

Supplementary Table 1. Dependence of the optical properties and size of $\text{AgIn}_x\text{Ga}_{1-x}\text{S}_2/\text{AgGaS}_2$ QDs on the Ga/Ag precursor ratio. As the Ag precursor amount increases, the average particle size of the $\text{AgIn}_x\text{Ga}_{1-x}\text{S}_2/\text{AgGaS}_2$ QDs as measured by TEM increases. Meanwhile, the trap emission increases, and the QY decreases. Owing to the relatively low reactivity of the Ga precursor, increasing the Ag precursor under Ga-rich conditions primarily leads to further growth of the AgGaS_2 shell. The trap emission area ratio was defined as the fraction of the integrated PL intensity in the long-wavelength region ($\lambda > \lambda_{\text{PL max}} + 50$ nm) to the total integrated emission.

References have been added as follows:

36. Dai, M. et al. Tunable photoluminescence from the visible to near-infrared wavelength region of non-stoichiometric AgInS_2 nanoparticles. *J. Mater. Chem.* **22**, 12851-12858 (2012). [10.1039/C2JM31463K](https://doi.org/10.1039/C2JM31463K).

37. Chen, Y. et al. Green and facile synthesis of high-quality water-soluble Ag-In-S/ZnS core/shell quantum dots with obvious bandgap and sub-bandgap excitations. *J. Alloys Compd.* **753**, 364-370 (2018). [10.1016/j.jallcom.2018.04.242](https://doi.org/10.1016/j.jallcom.2018.04.242).

38. Park, S. M. et al. Red Ag-based I–III–VI quantum dots as competitive alternatives to InP emitters. *ACS Energy Lett.* **10**, 3005-3013 (2025). [10.1021/acseenergylett.5c00962](https://doi.org/10.1021/acseenergylett.5c00962).

39. Farid, A. et al. One-pot synthesis of luminescent Ag–Cu–Ga–S/ZnS quantum dots bridging the cyan gap for ultrahigh-color-rendering white-light-emitting diodes. *ACS Appl. Nano Mater.* **8**,

14703-14712 (2025). [10.1021/acsnanm.5c02386](https://doi.org/10.1021/acsnanm.5c02386).

40. Park, S. et al. Suppressing tail emission from $\text{AgIn}_{1-x}\text{Ga}_x\text{S}_2/\text{AgGaS}_2$ quantum dots by GaI_3 -assisted interface reinforcement. *ACS Nano* **19**, 26831-26842 (2025). [10.1021/acsnano.5c07418](https://doi.org/10.1021/acsnano.5c07418).

71. Hu, T., Zhu, K., Cheng, H., Teng, Y. & Pan, Z. Core-shell energy band engineering of cyan light-emitting ternary $\text{ZnGa}_2\text{S}_4@/\text{ZnS}$ quantum dots toward anti-counterfeiting and bioimaging applications. *J. Mater. Chem. C* **13**, 21797-21811 (2025). [10.1039/D5TC02797G](https://doi.org/10.1039/D5TC02797G).

72. Yadav, A. N. & Singh, K. Investigation of photophysical properties of ternary Zn-Ga-S quantum dots: band gap versus sub-band-gap excitations and emissions. *ACS Omega* **4**, 18327-18333 (2019). [10.1021/acsomega.9b02546](https://doi.org/10.1021/acsomega.9b02546).

Comment 3. No definition of the x and y values were indicated. No explanation of $x \gg y$ was made, because each formula only contained x or y .

Answer to Comment 3: We thank the reviewer for pointing out the lack of clarity regarding the definitions of the variables x and y and the meaning of the relation $x \gg y$. We apologize for this ambiguity in the original manuscript.

The notation $x \gg y$ is used to emphasize that In is not completely absent from the outer Ag-deficient shell, but is present at a significantly lower concentration than in the core. This notation therefore serves as a qualitative compositional description highlighting the strong depletion of In in the outer shell region. In the revised manuscript, we now explicitly define x and y as the In fractions in different regions of the heterostructured QDs. Specifically, x denotes the In fraction in the $\text{AgIn}_x\text{Ga}_{1-x}\text{S}_2$ core, whereas y represents the residual In fraction in the Ag-deficient layer formed at the outer surface of the AgGaS_2 shell. Here, the notation $x \gg y$ is intended as a qualitative description rather than a strict mathematical inequality.

The manuscript has been revised as follows:

Please see Page #9,

“In addition, trace amounts of In atoms remain present in the shell region, leading to a shell composition closer to $\text{AgIn}_y\text{Ga}_{1-y}\text{S}_2$, where x denotes the fraction of In in the core and y represents the residual In fraction in the outer layer, with $x \gg y$, rather than a pure AgGaS_2 shell. This is likely due to the outward diffusion of In atoms from the $\text{AgIn}_x\text{Ga}_{1-x}\text{S}_2$ core to vacant sites in AgGaS_2 , driven by the concentration gradient or thermal budget during synthesis. Nevertheless, the resulting $\text{AgIn}_y\text{Ga}_{1-y}\text{S}_2$ shell is expected

to effectively confine charges within the $\text{AgIn}_x\text{Ga}_{1-x}\text{S}_2$ core owing to its higher band gap, consistent with a Type-I heterostructure, which promotes radiative recombination.”

Comment 4. The charge neutrality in single QDs and its spatial modulation should be discussed.

Answer to Comment 4: We thank the reviewer for raising the important point regarding charge neutrality in single quantum dots (QDs) and its possible spatial modulation.

In the present study, the $\text{AgIn}_x\text{Ga}_{1-x}\text{S}_2$ -based QDs are capped with oleylamine ligands, which act as electrically neutral L-type ligands. Accordingly, each individual QD is expected to maintain overall charge neutrality. This is further supported by bulk elemental analysis (e.g., Inductively Coupled Plasma Optical Emission Spectrometry), which confirms that the overall cation-to-anion balance of the synthesized QDs is consistent with charge neutrality.

From a spatial perspective, the APT results are also consistent with local charge balance within the QDs. In the $\text{AgIn}_x\text{Ga}_{1-x}\text{S}_2$ core, S is distributed relatively uniformly, while variations in cation composition occur in a compensatory manner, such that regions enriched in Ag correspond to reduced In and Ga fractions, and vice versa. Similarly, in the shell regions, S remains approximately constant while enrichment of one cation species (e.g., Ga in the $\text{Zn}_{1-3/2x}\text{Ga}_x\text{S}$ shell) is accompanied by a corresponding reduction of other cations. These observations indicate that, despite compositional variations, local electroneutrality may be largely preserved through redistribution of cation species rather than through net charge accumulation. In addition, in $\text{AgIn}_x\text{Ga}_{1-x}\text{S}_2$ -based QDs (e.g., AIGS, AIS/ZnS, and AIZS), the formation of the core and shell is known to involve cation exchange processes. During these processes, cation vacancies can be generated as part of the charge-compensation mechanism required to maintain local charge neutrality within the lattice. This behavior has been widely discussed in the context of I–III–VI₂ QDs, where such vacancies are generally regarded as intrinsic structural defects associated with the synthesis and shell-formation process, rather than as sources of net charging of the QDs. That is, the QDs are globally charge-neutral, and local compositional variations and defects formed during cation exchange can lead to spatial modulation of defect states while maintaining overall charge neutrality.

The manuscript has been revised as follows:

Please see Page # 14,

“The pronounced inhomogeneity in the cores likely originates from the intrinsic complexity of the quaternary Ag–In–Ga–S system. The synthesis of the core simultaneously involves multiple cation precursors with different reactivities, and the cores are formed under a continuous heating process. Previous studies have also shown that $\text{AgIn}_x\text{Ga}_{1-x}\text{S}_2$ core formation proceeds through intermediate phases and cation-exchange processes rather than a simple single-step growth process.^{18,53,59} More specifically, AgGaS_2 or AgInS_2 seeds form initially, followed by the incorporation of In and Ga to

generate the chalcopyrite $\text{AgIn}_x\text{Ga}_{1-x}\text{S}_2$ lattice, which is likely to promote local compositional fluctuations within the core.”

Please see Supplementary Table 3,

Specimen	Atomic ratio (ICP–OES)			
	Ag/S	In/S	Ga/S	Zn/S
AIGS/AGS	0.40	0.12	0.44	0
AIGS/AGS/ZnS	0.13	0.08	0.35	0.46

Supplementary Table 3. Atomic ratios measured of $\text{AgIn}_x\text{Ga}_{1-x}\text{S}_2/\text{AgGaS}_2$ (AIGS/AGS) and $\text{AgIn}_x\text{Ga}_{1-x}\text{S}_2/\text{AgGaS}_2/\text{ZnS}$ (AIGS/AGS/ZnS) determined by ICP–OES.

Please see Supplementary Table 4,

Specimen	Atomic ratio (APT)			
	Ag/S	In/S	Ga/S	Zn/S
AIGS/AGS	0.43	0.13	0.39	0
AIGS/AGS/ZnS	0.15	0.08	0.30	0.48

Supplementary Table 4. Atomic ratios of $\text{AgIn}_x\text{Ga}_{1-x}\text{S}_2/\text{AgGaS}_2$ (AIGS/AGS) and $\text{AgIn}_x\text{Ga}_{1-x}\text{S}_2/\text{AgGaS}_2/\text{ZnS}$ (AIGS/AGS/ZnS) determined by APT.

References have been added as follows:

- Lee, H. J. et al. Coherent heteroepitaxial growth of I-III-VI₂ Ag(In,Ga)S₂ colloidal nanocrystals with near-unity quantum yield for use in luminescent solar concentrators. *Nat. Commun.* **14**, 3779 (2023). [10.1038/s41467-023-39509-y](https://doi.org/10.1038/s41467-023-39509-y).

53. Uematsu, T., Tepakidareekul, M., Hirano, T., Torimoto, T. & Kuwabata, S. Facile high-yield synthesis of Ag–In–Ga–S quaternary quantum dots and coating with gallium sulfide shells for narrow band-edge emission. *Chem. Mater.* **35**, 1094-1106 (2023). [10.1021/acs.chemmater.2c03023](https://doi.org/10.1021/acs.chemmater.2c03023).
54. Lee, S. H. et al. The effects of discrete and gradient mid-shell structures on the photoluminescence of single InP quantum dots. *Nanoscale* **11**, 23251-23258 (2019). [10.1039/C9NR06847C](https://doi.org/10.1039/C9NR06847C).
55. Lim, J. et al. InP@ ZnSeS, core@ composition gradient shell quantum dots with enhanced stability. *Chem. Mater.* **23**, 4459-4463 (2011). [10.1021/cm201550w](https://doi.org/10.1021/cm201550w).
56. Bae, W. K., Char, K., Hur, H. & Lee, S. Single-step synthesis of quantum dots with chemical composition gradients. *Chem. Mater.* **20**, 531-539 (2008). [10.1021/cm070754d](https://doi.org/10.1021/cm070754d).
57. Duan, X. et al. InP quantum dots with a strain-engineered gradient shell for enhanced optical performance and stability. *Nano Lett.* **25**, 13539-13548 (2025). [10.1021/acs.nanolett.5c03042](https://doi.org/10.1021/acs.nanolett.5c03042).
58. Yu, P. et al. Highly efficient green InP-based quantum dot light-emitting diodes regulated by inner alloyed shell component. *Light Sci. Appl.* **11**, 162 (2022). [10.1038/s41377-022-00855-z](https://doi.org/10.1038/s41377-022-00855-z).
59. Xie, X. et al. Narrow-bandwidth I–III–VI semiconductor nanocrystals: synthesis, luminescence and applications in quantum-dot light-emitting diodes. *Adv. Phys. Res.* **3**, 2400071 (2024). [10.1002/apxr.202400071](https://doi.org/10.1002/apxr.202400071).

Comment 5. AgIn_xGa_{1-x}S₂/AgGaS₂ had quite Ag-deficient surface layer. What is the crystal structure of this layer? Ag-doped Ga₂S₃??

Answer to Comment 5: We thank the reviewer for this insightful question regarding the crystal structure of the Ag-deficient surface layer in the AgIn_xGa_{1-x}S₂/AgGaS₂ quantum dots (QDs).

Both the AgIn_xGa_{1-x}S₂ core and the AgGaS₂ shell are well known to crystallize in a chalcopyrite structure with tetragonal symmetry. Our X-ray diffraction (XRD) analysis confirms that the core-only AgIn_xGa_{1-x}S₂ samples and the AgIn_xGa_{1-x}S₂/AgGaS₂ samples exhibit essentially identical diffraction patterns, consistent with a chalcopyrite-type tetragonal structure. This XRD observation indicates that the overall crystal structure is preserved even after shell formation. Consistently, AgInS₂ and AgGaS₂

QDs reported in the literature are also known to adopt the chalcopyrite structure, and $\text{AgIn}_x\text{Ga}_{1-x}\text{S}_2$ QDs can be regarded as chalcopyrite AgInS_2 or AgGaS_2 lattices in which Ga or In partially substitutes the group-III cation sites.

We acknowledge that the presence of an Ag-deficient surface layer may raise the question of whether a different crystalline phase, such as Ga_2S_3 , is formed. However, complete removal of Ag from the lattice is unlikely, and as long as the S sublattice is preserved, Ag deficiency alone does not necessarily lead to a collapse or reconstruction into a fundamentally different crystal structure. Instead, Ag vacancies are expected to introduce defects while maintaining the underlying chalcopyrite framework. Therefore, we interpret the Ag-deficient surface region as retaining a tetragonal chalcopyrite-based structure with a substantial density of Ag vacancies, which can be more appropriately described as a chalcopyrite-like or defective chalcopyrite phase rather than as a distinct Ga_2S_3 -type phase. Consequently, our structural and diffraction data support the assignment of the Ag-deficient surface layer to a chalcopyrite-derived structure with Ag vacancies.

The manuscript has been revised as follows:

Please see Page # 6,

“Moreover, the high-resolution STEM results reveal that the QDs are structurally similar before and after the ZnS precursor step, with all crystal planes showing a lattice spacing of ~ 0.29 nm, which almost corresponds to the interplanar spacing of the (200) planes of chalcopyrite-structured AgGaS_2 . The lattice fringes are clearly evident not only inside the $\text{AgIn}_x\text{Ga}_{1-x}\text{S}_2$ -based QDs but also at the edges, indicating the high crystallinity across the entire QD structure. (Figs. 1b and 1e). X-ray diffraction (XRD) measurements were further performed on the core-only $\text{AgIn}_x\text{Ga}_{1-x}\text{S}_2$, $\text{AgIn}_x\text{Ga}_{1-x}\text{S}_2/\text{AgGaS}_2$, and $\text{AgIn}_x\text{Ga}_{1-x}\text{S}_2/\text{AgGaS}_2/\text{ZnS}$ QDs to corroborate the structural similarity before and after shell formation. All XRD patterns are consistent with a chalcopyrite-based tetragonal structure, and the overall diffraction features remain essentially unchanged, even after the Zn precursor step (Supplementary Fig. 4). This comparative analysis indicates that the crystal structure is preserved throughout the shell formation process.”

Please see Supplementary Fig. 4,

Supplementary Fig. 4 XRD patterns of $\text{AgIn}_x\text{Ga}_{1-x}\text{S}_2$ -based QDs. XRD patterns of $\text{AgIn}_x\text{Ga}_{1-x}\text{S}_2$ (AIGS), $\text{AgIn}_x\text{Ga}_{1-x}\text{S}_2/\text{AgGaS}_2$ (AIGS/AGS), and $\text{AgIn}_x\text{Ga}_{1-x}\text{S}_2/\text{AgGaS}_2/\text{ZnS}$ (AIGS/AGS/ZnS) QDs. All XRD patterns are consistent with a chalcopyrite-based tetragonal structure, with only minor peak shifts attributable to slight lattice parameter variations.

Comment 6. Also, what is the crystal structure of the surface of $\text{AgIn}_x\text{Ga}_{1-x}\text{S}_2/\text{AgGaS}_2/\text{ZnS}$? The authors assumed the ZnGaS alloy layer was formed on the surface. However, no difference of lattice fringes between core and shell in the corresponding HAADF-STEM images (Fig. 1) was observed.

Answer to Comment 6: We thank the reviewer for the question regarding the crystal structure of the surface region in the $\text{AgIn}_x\text{Ga}_{1-x}\text{S}_2/\text{AgGaS}_2/\text{ZnS}$ quantum dots (QDs) and the absence of discernible lattice-fringe differences between the core and shell in the HAADF-STEM images.

The $\text{AgIn}_x\text{Ga}_{1-x}\text{S}_2$ core is well known to crystallize in a chalcopyrite structure with tetragonal symmetry, and AgGaS_2 QDs reported in the literature also adopt the same chalcopyrite structure. In the present work, no obvious difference in lattice fringes is observed between the core and shell regions in the HAADF-STEM images (Fig. 1). Although resolving subtle structural differences in nanometer-scale QDs is inherently challenging, this observation is consistent with our interpretation that both the AgGaS_2 inner shell and the $\text{Zn}_{1-3/2x}\text{Ga}_x\text{S}$ outer shell retain a chalcopyrite-type tetragonal structure. If the core, inner shell, and outer shell share similar crystallographic symmetry, the absence of clear lattice-fringe contrast between these regions is therefore consistent with this interpretation.

To further support this interpretation, we performed X-ray diffraction (XRD) analysis on the core-only $\text{AgIn}_x\text{Ga}_{1-x}\text{S}_2$ samples, the $\text{AgIn}_x\text{Ga}_{1-x}\text{S}_2/\text{AgGaS}_2$ samples, and the $\text{AgIn}_x\text{Ga}_{1-x}\text{S}_2/\text{AgGaS}_2/\text{ZnS}$ samples. All samples exhibit diffraction patterns consistent with a tetragonal chalcopyrite-based structure. Notably, even after the ZnS shelling step, the outer layer is identified as $\text{Zn}_{1-3/2x}\text{Ga}_x\text{S}$ by APT analysis, while the diffraction patterns remain essentially unchanged, aside from minor peak shifts attributable to slight variations in lattice parameters. While the diffraction peaks are inevitably broadened due to the nanocrystalline nature of the QDs, the comparative analysis remains meaningful and indicates that the overall crystal structure is preserved throughout the shell formation process. Therefore, the absence of a clear lattice-fringe contrast between the core and shell in high-resolution TEM images can be attributed to their strong crystallographic similarity. From a local structural perspective, even if partial Ag depletion occurs near the surface, the underlying S sublattice and the tetragonal framework are expected to remain intact. Within this context, the $\text{Zn}_{1-3/2x}\text{Ga}_x\text{S}$ outer layer can be described as adopting a chalcopyrite-type tetragonal structure, which provides a plausible explanation for the absence of distinct lattice-fringe contrast between the core and shell regions in the HAADF-STEM images. We have added the corresponding XRD results to the Supplementary Information to support the structural similarity of the QDs before and after shell formation.

The manuscript has been revised as follows:

Please see Page # 6,

“Moreover, the high-resolution STEM results reveal that the QDs are structurally similar before and after the ZnS precursor step, with all crystal planes showing a lattice spacing of ~ 0.29 nm, which almost corresponds to the interplanar spacing of the (200) planes of chalcopyrite-structured AgGaS_2 . The lattice fringes are clearly evident not only inside the $\text{AgIn}_x\text{Ga}_{1-x}\text{S}_2$ -based QDs but also at the edges, indicating the high crystallinity across the entire QD structure. (Figs. 1b and 1e). X-ray diffraction (XRD) measurements were further performed on the core-only $\text{AgIn}_x\text{Ga}_{1-x}\text{S}_2$, $\text{AgIn}_x\text{Ga}_{1-x}\text{S}_2/\text{AgGaS}_2$, and $\text{AgIn}_x\text{Ga}_{1-x}\text{S}_2/\text{AgGaS}_2/\text{ZnS}$ QDs to corroborate the structural similarity before and after shell formation. All XRD patterns are consistent with a chalcopyrite-based tetragonal structure, and the overall diffraction features remain essentially unchanged, even after the Zn precursor step (Supplementary Fig. 4). This comparative analysis indicates that the crystal structure is preserved throughout the shell formation process.”

Please see Page #15,

“This process forms a compositionally mixed $\text{Zn}_{1-3/2x}\text{Ga}_x\text{S}$ outer shell with a chalcopyrite structure, which is the overall crystallographic framework of the AgGaS_2 inner shell.^{18,53,60-65} Fortuitously, the formation of $\text{Zn}_{1-3/2x}\text{Ga}_x\text{S}$ with a chalcopyrite-like structure, which is the same as that of AgGaS_2 , is crystallographically preferable over zincblende-structured ZnS.”

Please see Supplementary Fig. 4.

Supplementary Fig. 4 XRD patterns of $\text{AgIn}_x\text{Ga}_{1-x}\text{S}_2$ -based QDs. XRD patterns of $\text{AgIn}_x\text{Ga}_{1-x}\text{S}_2$ (AIGS), $\text{AgIn}_x\text{Ga}_{1-x}\text{S}_2/\text{AgGaS}_2$ (AIGS/AGS), and $\text{AgIn}_x\text{Ga}_{1-x}\text{S}_2/\text{AgGaS}_2/\text{ZnS}$ (AIGS/AGS/ZnS) QDs. All XRD patterns are consistent with a chalcopyrite-based tetragonal structure, with only minor peak shifts attributable to slight lattice parameter variations.

References have been added as follows:

18. Lee, H. J. et al. Coherent heteroepitaxial growth of I-III-VI₂ Ag(In,Ga)S₂ colloidal nanocrystals with near-unity quantum yield for use in luminescent solar concentrators. *Nat. Commun.* **14**, 3779 (2023). [10.1038/s41467-023-39509-y](https://doi.org/10.1038/s41467-023-39509-y).
53. Uematsu, T., Tepakidarekul, M., Hirano, T., Torimoto, T. & Kuwabata, S. Facile high-yield synthesis of Ag–In–Ga–S quaternary quantum dots and coating with gallium sulfide shells for narrow band-edge emission. *Chem. Mater.* **35**, 1094-1106 (2023). [10.1021/acs.chemmater.2c03023](https://doi.org/10.1021/acs.chemmater.2c03023).
60. Yan, D. et al. High photoluminescence Ag-In-Ga-S quantum dots based on ZnX₂-treated surface passivation. *Nano Res.* **17**, 7533-7541 (2024). [10.1007/s12274-024-6724-0](https://doi.org/10.1007/s12274-024-6724-0).

61. Jang, J. S., Borse, P. H., Lee, J. S., Choi, S. H. & Kim, H. G. Indium induced band gap tailoring in $\text{AgGa}_{1-x}\text{In}_x\text{S}_2$ chalcopyrite structure for visible light photocatalysis. *J. Chem. Phys.* **128** (2008). doi.org/10.1063/1.2900984.
62. Asadullayeva, S. G., Ismayilova, N. A., Musayev, M. A. & Abbasov, I. I. Optical and electronic properties of defect chalcopyrite ZnGa_2S_4 . *Int. J. Mod. Phys. B* **38**, 2450007 (2024). [10.1142/S0217979224500073](https://doi.org/10.1142/S0217979224500073).
63. Bai, L., Lin, Z., Wang, Z., Chen, C. & Lee, M. H. Mechanism of linear and nonlinear optical effects of chalcopyrite AgGaX_2 (X=S, Se, and Te) crystals. *J. Chem. Phys.* **120**, 8772-8778 (2004). [10.1063/1.1687338](https://doi.org/10.1063/1.1687338).
64. Kaga, H. & Kudo, A. Cosubstituting effects of copper(I) and gallium(III) for ZnGa_2S_4 with defect chalcopyrite structure on photocatalytic activity for hydrogen evolution. *J. Catal.* **310**, 31-36 (2014). [10.1016/j.jcat.2013.08.025](https://doi.org/10.1016/j.jcat.2013.08.025).
65. Sahariya, J., Kumar, P. & Soni, A. Structural and optical investigations of ZnGa_2X_4 (X = S, Se) compounds for solar photovoltaic applications. *Mater. Chem. Phys.* **199**, 257-264 (2017). [10.1016/j.matchemphys.2017.07.003](https://doi.org/10.1016/j.matchemphys.2017.07.003).

Thus, the present manuscript does not contain novel results. The manuscript may not be suitable for the publication in the preset journal.

Reviewer #3 (Remarks to the Author):

General Comment: The authors present an important study on the atomistic level for Quantum Dots using APT. The interface design and the intermixing of the components is still a great uncertainty for quantum dots, therefore, a knowledge of the distribution of the components is very important. APT is very promising for obtain more important and exciting results about the intermixing which can use for a better tuning of the synthesis of QDs. All in all, the manuscript is very important contribution in this field. But there are some points which should be clarified before publication:

Answer to General Comment: We sincerely thank the reviewer for the positive assessment of our work and for recognizing the importance of APT to achieve an atomistic-level understanding of compositional distributions in $\text{AgIn}_x\text{Ga}_{1-x}\text{S}_2$ -based quantum dots. We fully agree with the reviewer's perspective on the potential of APT to guide improved synthesis strategies. In response to the reviewer's comments, we have made every effort to further enhance the reliability of our analysis and to clarify data interpretation and visualization throughout the manuscript. All comments raised by the reviewer have been carefully addressed, and the manuscript has been revised accordingly. Detailed responses to each comment are provided below.

Comment 1. In Figs. 2, 4, and 6 different particles are shown in (a), in Figs. 2 and 4 there are five. It is unclear if all particles of them were further evaluated or only one of them was evaluated.

Answer to Comment 1: We thank the reviewer for seeking clarification. In all APT datasets shown in Figs. 2, 4, and 6, five quantum dots (QDs) were analyzed within a single reconstructed volume. Figure 2 presents the APT results for the $\text{AgIn}_x\text{Ga}_{1-x}\text{S}_2/\text{AgGaS}_2$ QDs, while Figures 4 and 6 correspond to the $\text{AgIn}_x\text{Ga}_{1-x}\text{S}_2/\text{AgGaS}_2/\text{ZnS}$ QDs.

Figures 4 and 6 are based on the same APT dataset and include the same five QDs; however, Figure 6 uses a different visualization to highlight the spatial distribution of Cl. No Cl signal was detected in the $\text{AgIn}_x\text{Ga}_{1-x}\text{S}_2/\text{AgGaS}_2$ QDs shown in Fig. 2. Therefore, the analysis was not limited to a single particle, and all five QDs in each reconstructed dataset were evaluated consistently.

Comment 2. In Figs. 4 and 6 the five particles are not homogeneous, in QD 4 and QD 5 (Fig.4) Ag seems be not so present, in Fig. 6 In was inhomogeneously distributed.

Answer to Comment 2: We thank the reviewer for the careful observation. The apparent compositional inhomogeneities observed in Figs. 4 and 6, such as the reduced apparent Ag signal in QD#4 and QD#5 in Fig. 4 or the seemingly inhomogeneous In distribution in Fig. 6, are largely influenced by geometric and visualization-related effects rather than from intrinsic compositional variations among the quantum dots (QDs).

In the APT specimen, the QDs are not arranged in a single line or perfectly aligned along the analysis (z) direction, but instead exhibit different lateral positions, with each QD occupying a distinct position in the x-y plane. As a result, sliced-view images such as Fig. 4b cannot intersect all QDs at an identical central plane. Similarly, the 2D compositional profiles in Fig. 4d were extracted uniformly along the z

direction from a fixed cylindrical sampling volume ($3 \times 3 \times 21 \text{ nm}^3$). Consequently, in some QDs the profiles partially sample the shell region, whereas in others they primarily intersect the core, which can give rise to apparent dot-to-dot variations in the compositional profiles.

To evaluate the intrinsic compositional uniformity more accurately, we therefore extracted and analyzed the full 3D volumes of individual QDs, as shown in Fig. 5. These analyses confirm that, while minor variations exist, there are no significant compositional anomalies, such as a pronounced depletion of Ag in specific QDs. Moreover, the sliced 3D atom maps in Fig. 4b provide the most direct visualization of the internal spatial distributions, clearly showing that Ag and In are appropriately distributed within each QD at their respective positions.

Figure 6 is based on the same APT dataset as Fig. 4 and was visualized differently to emphasize the spatial distribution of Cl. In this figure, the entire 3D reconstruction volume is displayed rather than a sliced view. Consequently, depending on the projection direction, QDs located along the line of sight can partially overlap along the projection direction and obscure one another, making some elemental distributions—such as In—appear more inhomogeneous. Nevertheless, these different visualization modes were intentionally employed to illustrate both the internal spatial distributions and the overall compositional distribution tendencies of the QDs. To avoid potential misinterpretation of the compositional maps, we have clarified these visualization and geometry related effects in the revised manuscript.

The manuscript has been revised as follows:

Please see Page #8,

“Because the QDs are closely packed in a non-collinear configuration within the APT specimen and may therefore partially overlap in projection views, 3D reconstructions with C- and H-related ions removed are also provided in Supplementary Fig. 7 to better visualize the individual QDs.”

Please see Page #12,

“The 2D concentration profile along the analysis direction offers additional insights into the internal structure of each QD (Fig. 4d). Although the elemental concentrations vary slightly depending on the profiling path owing to the spatial offset of individual QDs in the APT specimen, the 2D profile confirms the significant presence of Ga atoms in the outer shell, albeit at lower concentrations than Zn atoms.”

Please see Fig. 2,

Fig. 2 3D reconstruction and compositional analysis of $\text{AgIn}_x\text{Ga}_{1-x}\text{S}_2/\text{AgGaS}_2$ QDs using APT. **a** Reconstructed 3D atom map of the entire analyzed volume, showing five slightly ellipsoidal QDs. **b** Slice-view atom maps of Ag, In, Ga, and S, illustrating their internal spatial distributions. Ga and S atoms are more broadly distributed than Ag and In. **c** Ag iso-surface map and proxigram showing 3D compositional trends; the proxigram was obtained from five QDs. The Ag and In contents increase toward the QD core, while the Ga content increases toward its outer surface, confirming the core/shell heterostructure. (Scale bars: 2 nm in **a–c**.) Error bars in **c** indicate one-sigma counting statistics.

Please see Fig. 4,

Fig. 4 3D reconstruction and compositional analysis of $\text{AgIn}_x\text{Ga}_{1-x}\text{S}_2/\text{AgGaS}_2/\text{ZnS}$ QDs using APT.

a Reconstructed 3D atom map of the entire analyzed volume, showing five slightly ellipsoidal QDs. **b** Slice-view atom maps of Ag, In, Ga, Zn, and S, illustrating their internal spatial distributions. Although most Ag and In atoms are in the core regions, a significant amount of Ga is distributed over the shell region. Zn atoms predominantly exist in the outer shell. **c** Ag and Zn iso-surface and proxigram obtained from Ag iso-surfaces; the proxigram was obtained from five QDs. The Ag and In contents increase toward the QD core, while the content of Zn increases toward the outer shell of the QD. A significant amount of Ga also appears in the outer shell. **d** 2D compositional profile extracted from a cylindrical

volume ($3 \times 3 \times 21 \text{ nm}^3$) through the 3D reconstruction, confirming the core/shell/shell structure. (Scale bars: 2 nm in **a–d**). The slight variations in the compositional distributions from QD to QD arise from the spatial offsets of individual QDs within the APT specimen. Error bars in **c, d** indicate one-sigma counting statistics.

Comment 3. Was the Cl correlated with some of the cations? I am missing a statement about this question?

Answer to Comment 3: We thank the reviewer for this question regarding the correlation of Cl with specific cations. As discussed in the manuscript, Cl originates from the ZnCl_2 precursor used during the ZnS shell formation process, and this point has now been emphasized more clearly in the revised text.

The APT results show that Cl is predominantly localized in the outer shell region and is not detected in the $\text{AgIn}_x\text{Ga}_{1-x}\text{S}_2/\text{AgGaS}_2$ quantum dots (QDs) prior to the ZnS precursor step. Importantly, Cl exhibits a strong spatial correlation with Zn, as clearly shown in Fig. 6, whereas no comparable correlation is observed with other cations. This is consistent with the shell composition revealed by APT, where the outer shell is identified as $\text{Zn}_{1-3/2x}\text{Ga}_x\text{S}$. Because no additional Ga precursor is introduced during the shell formation step, the observed Cl distribution is attributed to Zn-related species, indicating that Cl is correlated with Zn rather than with Ag or In.

To further support the reliability of the APT results, we performed complementary X-ray photoelectron spectroscopy (XPS) measurements. While XPS does not provide 3D spatial information, it clearly shows that Cl is nearly absent in the $\text{AgIn}_x\text{Ga}_{1-x}\text{S}_2/\text{AgGaS}_2$ QDs and becomes distinctly detectable only after the ZnS precursor step, that is, in the $\text{AgIn}_x\text{Ga}_{1-x}\text{S}_2/\text{AgGaS}_2/\text{ZnS}$ QDs. The Cl 2p core-level spectra reveal the presence of metal–Cl chemical states exclusively in the $\text{AgIn}_x\text{Ga}_{1-x}\text{S}_2/\text{AgGaS}_2/\text{ZnS}$ samples, further supporting the strong Zn–Cl association observed by APT. Given that no significant metal–Cl component is observed in the $\text{AgIn}_x\text{Ga}_{1-x}\text{S}_2/\text{AgGaS}_2$ QDs, this Cl signal is most reasonably attributed to Zn–Cl–related chemical states introduced during the ZnS precursor step. This observation is fully consistent with the APT data and supports the conclusion that Cl is introduced during the Zn-based shell formation process. As discussed in the manuscript, although Cl is clearly associated with the Zn-containing outer shell, its precise role, whether acting as a trap related defect or as an inorganic ligand, cannot be conclusively determined within the scope of the present study. The revised manuscript has been updated accordingly, and the corresponding XPS results have been added to the Supplementary Information.

The manuscript has been revised as follows:

Please see Page #17,

“As shown in Fig. 6 and Supplementary Fig. 19, Cl atoms are detected only in the $\text{AgIn}_x\text{Ga}_{1-x}\text{S}_2/\text{AgGaS}_2/\text{ZnS}$ QDs and are localized in the outer shell region in only trace amounts. The Cl atoms, derived from the ZnCl_2 precursor used in the ZnS precursor step, show a strong spatial correlation with

Zn atoms in the APT analysis, whereas no comparable correlation is observed with Ag, In, or Ga atoms. XPS measurement further confirms that Cl is absent in $\text{AgIn}_x\text{Ga}_{1-x}\text{S}_2$ cores and nearly absent in $\text{AgIn}_x\text{Ga}_{1-x}\text{S}_2/\text{AgGaS}_2$ QDs, whereas the Cl signal is pronounced in the $\text{AgIn}_x\text{Ga}_{1-x}\text{S}_2/\text{AgGaS}_2/\text{ZnS}$ QDs, exhibiting metal–Cl chemical states attributable to Zn–Cl-related species (Supplementary Fig. 20 and Supplementary Table 2).”

Please see Supplementary Fig. 15,

Supplementary Fig. 20 XPS Cl $2p$ core-level spectra of $\text{AgIn}_x\text{Ga}_{1-x}\text{S}_2$ -based QDs. XPS spectra of Cl $2p$ core-level acquired from $\text{AgIn}_x\text{Ga}_{1-x}\text{S}_2$ (AIGS), $\text{AgIn}_x\text{Ga}_{1-x}\text{S}_2/\text{AgGaS}_2$ (AIGS/AGS), and $\text{AgIn}_x\text{Ga}_{1-x}\text{S}_2/\text{AgGaS}_2/\text{ZnS}$ (AIGS/AGS/ZnS) QDs. Cl-related signals are observed only in the AIGS/AGS/ZnS QDs, indicating the presence of metal–Cl chemical states associated with the Zn-based shell introduced during the ZnS precursor step.

Please see Supplementary Table 2.

Specimen	Atomic composition (%) (XPS)								
	C (in C 1s)	O (in O 1s)	F (in F 1s)	S (in S 2s)	Cl (in Cl 2p)	Zn (in Zn 2p _{3/2})	Ga (in Ga 2p _{3/2})	Ag (in Ag 3d)	In (in In 3d _{5/2})
AIGS	76.29	1.91	0	11.54	0	0	1.1	4.46	4.7
AIGS/AGS	69.37	2.83	0	14.69	0.23	0	5.16	5.9	1.83
AIGS/AGS/ZnS	62.92	6.58	0	13.93	2.07	6.07	4.54	2.85	1.06

Specimen	Atomic composition (%) (XPS)			
	S (in S 2s)	Ga (in Ga 2p _{3/2})	Ag (in Ag 3d)	In (in In 3d _{5/2})
AIGS	52.92	5.07	20.44	21.57
AIGS/AGS	53.25	18.71	21.41	6.63
AIGS/AGS/ZnS	62.28	20.28	12.73	4.72

Specimen	Atomic composition (%) (XPS)			
	Ag (in Ag 4d)	Zn (in Zn 3d)	In (in In 4d)	Ga (in Ga 3d)
AIGS	36.1	0	37.7	26.2
AIGS/AGS	37.1	0	10.0	52.9
AIGS/AGS/ZnS	21.3	30.5	5.1	43.1

Supplementary Table 2. Atomic compositions of $\text{AgIn}_x\text{Ga}_{1-x}\text{S}_2/\text{AgGaS}_2$ (AIGS/AGS) and $\text{AgIn}_x\text{Ga}_{1-x}\text{S}_2/\text{AgGaS}_2/\text{ZnS}$ (AIGS/AGS/ZnS) QDs measured by XPS. The Ga-to-Ag ratio is consistently higher in the AIGS/AGS/ZnS QDs than in the AIGS/AGS QDs, suggesting the formation of a Ga-containing outer shell. This trend is observed both when comparing Ag *MNN* and Ga *2p* and when comparing Ag *4d* and Ga *3d*, all of which lie in similar binding energy ranges.

Comment 4. The general problem of all microscopic studies are the statistical relevance. Some comments to this issue would be very helpful for the readership of this manuscript.

Answer to Comment 4: We thank the reviewer for raising the important issue of statistical relevance, which is a general limitation of all microscopy-based techniques. As correctly pointed out, microscopic analyses inherently probe a limited number of particles compared to bulk characterization techniques that are more representative of the overall sample, and this limitation has now been explicitly acknowledged in the revised manuscript.

The primary objective of this work is to directly visualize and elucidate key physical and chemical phenomena in $\text{AgIn}_x\text{Ga}_{1-x}\text{S}_2$ -based quantum dots (QDs), namely compositional gradients, precursor reactivity, and cation exchange-driven shell formation, using APT. A central contribution of this study is to demonstrate that such nanoscale processes can be directly accessed and analyzed using APT, thereby providing methodological guidance for future studies of complex QD systems.

To strengthen the statistical relevance beyond the limited number of QDs analyzed by APT and TEM, we performed complementary characterizations using X-ray photoelectron spectroscopy (XPS). XPS is a surface-sensitive technique that probes elemental composition and chemical bonding states within the top few nanometers of a sample, and therefore provides averaged information from a large number of QDs rather than 3D distributions within individual particles. XPS spectra acquired from different core-level regions confirm that the QDs consist of Ag–S, In–S, Ga–S, and Zn–S chemical bonds as designed, without detectable oxidation. Although XPS cannot resolve 3D spatial distributions, its surface sensitivity makes it suitable for inferring compositional changes near the outer region of the QDs. To assess the compositional trends, we compared the relative Ag and Ga signals before and after the ZnS shelling process using multiple core-level regions, including Ag *MNV* vs Ga *2p* and Ag *4d* vs Ga *3d*, each of which occupies an equivalent binding energy range. In all cases, the Ga-to-Ag ratios were consistently higher in the $\text{AgIn}_x\text{Ga}_{1-x}\text{S}_2/\text{AgGaS}_2/\text{ZnS}$ QDs than in the $\text{AgIn}_x\text{Ga}_{1-x}\text{S}_2/\text{AgGaS}_2$ QDs. This systematic increase allows us to infer that substantial Ga-related components are present in the outermost shell region after the ZnS precursor step. This XPS result is fully consistent with the APT observations that reveal Ga-containing outer shells, and suggests that the compositional features identified by APT are not limited to a small number of individual QDs but are representative of the overall sample.

In addition, the overall elemental compositions obtained from APT are consistent with inductively coupled plasma optical emission spectrometry measurements, supporting the representativeness of the analyzed volumes. Finally, X-ray diffraction (XRD) measurements confirm that the crystal structure remains similar before and after shell formation, with core-only $\text{AgIn}_x\text{Ga}_{1-x}\text{S}_2$, $\text{AgIn}_x\text{Ga}_{1-x}\text{S}_2/\text{AgGaS}_2$, and $\text{AgIn}_x\text{Ga}_{1-x}\text{S}_2/\text{AgGaS}_2/\text{ZnS}$ QDs all exhibiting a comparable chalcopyrite-based tetragonal structure. This structural consistency measured by XRD, also observed by transmission electron microscopy, further supports the representativeness of the microscopic observations. We now explicitly discuss the general issue of statistical representativeness inherent to microscopy-based analyses in the revised manuscript. In addition, we strengthen the interpretation by corroborating the APT results with multiple complementary characterization techniques.

The manuscript has been revised as follows:

Please see Page # 13,

“Because APT inherently probes only a limited number of individual QDs and may therefore suffer from limited statistical representativeness, the compositional features identified by APT were further examined using complementary techniques. XPS analyses performed on the same samples reveal

consistently higher Ga-to-Ag ratios after the Zn precursor step, evaluated from multiple core-level regions, including Ag *MNN* vs Ga *2p* and Ag *4d* vs Ga *3d*. This systematic increase in the Ga-to-Ag ratios supports the formation of Ga-containing outer shells and is consistent with the $Zn_{1-3/2x}Ga_xS$ shell formation revealed by APT (Supplementary Fig. 15 and Supplementary Table 2). Inductively coupled plasma analyses performed on the same samples show overall elemental compositions consistent with the APT results (Supplementary Tables 3 and 4). Taken together, these results indicate that the structural and compositional features identified by APT are not limited to a small number of individual QDs but are representative of overall sample.”

Please see Supplementary Fig. 15.

Supplementary Fig. 15 XPS analysis results of $AgIn_xGa_{1-x}S_2$ -based QDs. XPS spectra of **a** Ga *2p*, **b** Zn *2p*_{3/2}, **c** In *3d*, **d** Ag *3d*, and **e** S *2s* core levels acquired from $AgIn_xGa_{1-x}S_2$ (AIGS), $AgIn_xGa_{1-x}S_2/AgGaS_2$ (AIGS/AGS), and $AgIn_xGa_{1-x}S_2/AgGaS_2/ZnS$ (AIGS/AGS/ZnS) QDs using an Al K-alpha X-ray source. All QDs contain Ag–S, In–S, Ga–S, and Zn–S chemical bonds, as designed without detectable oxidation. **f** Valence band spectra of AIGS, AIGS/AGS, and AIGS/AGS/ZnS QDs. The Ga-to-Ag ratio is consistently higher in the AIGS/AGS/ZnS QDs than in the AIGS/AGS QDs. The detailed compositions are provided in **Supplementary Table 2**. These results further support the presence of a

Ga-containing outer shell, consistent with the $Zn_{1-3/2x}Ga_xS$ shell identified by APT after the ZnS precursor step.

Please see Supplementary Table 3,

Specimen	Atomic ratio (ICP–OES)			
	Ag/S	In/S	Ga/S	Zn/S
AIGS/AGS	0.40	0.12	0.44	0
AIGS/AGS/ZnS	0.13	0.08	0.35	0.46

Supplementary Table 3. Atomic ratios measured of $AgIn_xGa_{1-x}S_2/AgGaS_2$ (AIGS/AGS) and $AgIn_xGa_{1-x}S_2/AgGaS_2/ZnS$ (AIGS/AGS/ZnS) determined by ICP–OES.

Please see Supplementary Table 4,

Specimen	Atomic ratio (APT)			
	Ag/S	In/S	Ga/S	Zn/S
AIGS/AGS	0.43	0.13	0.39	0
AIGS/AGS/ZnS	0.15	0.08	0.30	0.48

Supplementary Table 4. Atomic ratios of $AgIn_xGa_{1-x}S_2 /AgGaS_2$ (AIGS/AGS) and $AgIn_xGa_{1-x}S_2/AgGaS_2/ZnS$ (AIGS/AGS/ZnS) determined by APT.

Comment 5. It would be helpful, to mark the single particles in Fig. 2, 4 and 6, then the reader can see the borders of the particles.

Answer to Comment 5: We thank the reviewer for this helpful suggestion. We agree that marking individual particles can assist readers in identifying the boundaries of the quantum dots (QDs) in the

3D reconstructions.

However, in the present APT datasets, the QDs are not aligned along a single common axis but are arranged in a non-collinear, close-packed configuration within the specimen. As a result, depending on the viewing direction, some particles partially overlap in projection views, making it nontrivial to unambiguously define particle boundaries within the main reconstruction figures.

To address the reviewer's suggestion while taking this limitation into account, we have prepared additional images in which the approximate outlines of individual QDs are indicated with dashed guides based on the 3D ion maps. Although the particles are not perfectly spherical, simplified spherical guides for visual reference were used to provide a clear and intuitive visualization of particle separation. To avoid introducing visual bias, such as a misleading visual impression or encouraging over-interpretation of the reconstructions in the main figures, these annotated images have been added to the Supplementary Information. The revised manuscript now explicitly refers to them.

The manuscript has been revised as follows:

Please see Page #8,

“Figure 2a shows the 3D reconstructed APT atom maps of the $\text{AgIn}_x\text{Ga}_{1-x}\text{S}_2/\text{AgGaS}_2$ QDs, representing a hemispherical APT specimen containing five QDs (see Supplementary Fig. 6 for dashed guides indicating the approximate outlines of individual QDs). The QDs appear as slightly ellipsoidal features in the APT reconstruction, and a slice-view image along the x - z plane is presented in Fig. 2b to visualize their internal elemental distribution.”

Please see Page #11,

“Following the ZnS precursor step to form the outermost shell, APT was conducted to investigate the compositional evolution of $\text{AgIn}_x\text{Ga}_{1-x}\text{S}_2/\text{AgGaS}_2/\text{ZnS}$ QDs. Figure 4a shows a reconstructed APT 3D atom map containing five nearly spherical $\text{AgIn}_x\text{Ga}_{1-x}\text{S}_2/\text{AgGaS}_2/\text{ZnS}$ QDs (see Supplementary Fig. 9 for dashed guides indicating the approximate outlines of individual QDs). To further clarify the presence of these five QDs within the reconstructed volume and their internal structures, slice-view images along the x - z planes are provided in Fig. 4b.”

Please see Supplementary Fig. 6,

Supplementary Fig. 6 APT 3D reconstructions of $\text{AgIn}_x\text{Ga}_{1-x}\text{S}_2/\text{AgGaS}_2$ QDs with delineated particle boundaries. APT 3D reconstructions showing multiple $\text{AgIn}_x\text{Ga}_{1-x}\text{S}_2/\text{AgGaS}_2$ QDs within the analyzed volume. Dashed ovals mark the boundaries between individual QDs. In the APT specimen, the QDs are arranged in a close-packed, non-collinear configuration, which can lead to partial overlap in the projection; the similar elemental compositions of neighboring QDs further reduce the visual contrast between adjacent particles. (Scale bars: 2 nm.)

Please see Supplementary Fig. 9.

Supplementary Fig. 9 APT 3D reconstructions of $\text{AgIn}_x\text{Ga}_{1-x}\text{S}_2/\text{AgGaS}_2/\text{ZnS}$ QDs with delineated particle boundaries. APT 3D reconstructions showing multiple $\text{AgIn}_x\text{Ga}_{1-x}\text{S}_2/\text{AgGaS}_2/\text{ZnS}$ QDs within the analyzed volume. Dashed guides mark the boundaries between individual QDs. In the APT specimen, the QDs are arranged in a close-packed, non-collinear configuration, leading to partial overlap in the projection. (Scale bars: 2 nm.)

Reviewer #4 (Remarks to the Author):

General Comment: The paper highlights an additional pathway to uncover the complex internal structure of quaternary AgInGaS core-shelled quantum dots using atom probe tomography (APT). It is a well-written paper, providing elemental compositional profiles across nanocrystals to obtain detailed information about the compositional variations of core-shell and core-shell-shell nanocrystals. Furthermore, insights are provided about relative precursor reactivity and potential substitution reactions occurring, as to why and how some of these layers were formed. I recommend publication in nature communications, with minor revisions.

Answer to General Comment: We sincerely thank the reviewer for the positive assessment of our manuscript and for recommending its publication in Nature Communications. We particularly appreciate the reviewer's recognition of the value of atom probe tomography in revealing detailed compositional profiles and internal structural variations in $\text{AgIn}_x\text{Ga}_{1-x}\text{S}_2$ -based quaternary core-shell and core-shell-shell quantum dots, as well as the insights into precursor reactivity and possible substitution reactions during synthesis. We agree with the reviewer's comments aimed at further strengthening the manuscript through additional clarification and literature support. In response, we have revised the manuscript accordingly, as detailed in the point-by-point responses below.

Comment 1. The authors make broad claims about how the compositional gradient would improve basically every QD property (quantum yield, stability, longevity, etc). The authors should provide some literature references that have shown this or indicate that this should be the case.

Answer to Comment 1: We thank the reviewer for pointing out that our original statements regarding the impact of compositional gradients may appear overly broad.

We acknowledge that our original wording may not have sufficiently conveyed the intended level of caution regarding the impact of compositional gradients on quantum dot (QD) properties. Accordingly, the manuscript has been revised to adopt a more cautious and clearly articulated tone, emphasizing that a compositional gradient is expected to be beneficial based on prior studies, rather than asserting guaranteed improvements. More specifically, numerous previous studies have reported that introducing compositionally graded shells or intermediate (mid-)shell layers can lead to enhanced photoluminescence quantum yield (QY) and improved photostability in colloidal QDs. These improvements are commonly attributed to reduced lattice mismatch at core/shell interfaces and suppressed defect formation. For example, compositionally graded or alloyed shell structures have been demonstrated to improve optical performance and stability in InP-based and related QD systems. Based on established literature precedents, it is reasonable to expect that the compositionally graded shells observed in our QDs could contribute to improved QY and stability. In response to the reviewer's comment, the revised manuscript has been updated to reflect this more conservative interpretation and to explicitly cite relevant literature supporting the potential benefits of compositional gradients, without overstating their general applicability or certainty.

The manuscript has been revised as follows:

Please see Page #13,

“Interestingly, the compositional distributions of Ag, In, and Ga are gradual rather than discrete across the layers. From a structural perspective, this compositional gradient is expected to offer advantages such as minimizing lattice mismatch between the core and inner shell and effectively reducing the likelihood of internal defect formation. Such structural features—often referred to as compositional-gradient shells or gradient alloy shells—have been widely reported in InP- and CdSe-based QDs to contribute to enhancing both the QY and stability.⁵⁴⁻⁵⁸”

References have been added as follows:

54. Lee, S. H. et al. The effects of discrete and gradient mid-shell structures on the photoluminescence of single InP quantum dots. *Nanoscale* **11**, 23251-23258 (2019). [10.1039/C9NR06847C](https://doi.org/10.1039/C9NR06847C).

55. Lim, J. et al. InP@ ZnSeS, core@ composition gradient shell quantum dots with enhanced stability. *Chem. Mater.* **23**, 4459-4463 (2011). [10.1021/cm201550w](https://doi.org/10.1021/cm201550w).

56. Bae, W. K., Char, K., Hur, H. & Lee, S. Single-step synthesis of quantum dots with chemical composition gradients. *Chem. Mater.* **20**, 531-539 (2008). [10.1021/cm070754d](https://doi.org/10.1021/cm070754d).

57. Duan, X. et al. InP quantum dots with a strain-engineered gradient shell for enhanced optical performance and stability. *Nano Lett.* **25**, 13539-13548 (2025). [10.1021/acs.nanolett.5c03042](https://doi.org/10.1021/acs.nanolett.5c03042).

58. Yu, P. et al. Highly efficient green InP-based quantum dot light-emitting diodes regulated by inner alloyed shell component. *Light Sci. Appl.* **11**, 162 (2022). [10.1038/s41377-022-00855-z](https://doi.org/10.1038/s41377-022-00855-z).

Comment 2. A prominent tail is seen in the PL spectra after the addition of a ZnS shell (or alloyed ZnGaS layer as the authors later conclude). Could the others discuss the origin of this tail? Is it related to the compositional gradients observed in the QDs.

Answer to Comment 2: We thank the reviewer for pointing out the pronounced long-wavelength (tail) emission observed after the addition of the ZnS shell (or alloyed $Zn_{1-3/2x}Ga_xS$ layer).

Long-wavelength emission following ZnS shell coating has been widely reported in AgInS₂- and related I-III-VI₂-based quantum dots (QDs) and is generally attributed to defect- or trap-related recombination pathways rather than band-edge excitonic transitions. Multiple mechanisms may contribute to such

emission, including unpassivated intrinsic defects, newly formed interfacial trap states, and defects introduced during cation exchange. To date, however, the precise origin of this tail emission has not been conclusively established in the literature.

In the present system, we attribute the emergence of the long-wavelength emission primarily to defect formation during the ZnS precursor step and the associated cation exchange process. Our transmission electron microscopy and X-ray diffraction analyses indicate that the shell retains a chalcopyrite-related tetragonal structure rather than forming an incoherent zinc-blende-type ZnS phase, suggesting that $\text{Zn}_{1-3/2x}\text{Ga}_x\text{S}$ is formed via cation exchange through substitution of Ag^+ and Ga^{3+} by Zn^{2+} . This intrinsically non-charge-neutral process can generate a substantial density of cation vacancies, which act as intragap trap states in I–III–VI₂ semiconductors. Such defect-related intragap states are commonly associated with donor–acceptor-pair-like recombination responsible for long-wavelength emission in I–III–VI₂ QDs. Similar trap-assisted tail emissions have been reported in $\text{AgInS}_2/\text{ZnS}$ and Zn-alloyed Ag–In–S QDs, where they are attributed to defect-mediated recombination rather than band-edge transitions. Consistent with our APT observations, the inward migration of Zn^{2+} species into the particle interior can further promote the formation of cation vacancies and associated defect complexes. When these defects are introduced in the vicinity of In-rich domains, where charge carriers are preferentially localized, they can give rise to intragap trap states that locally perturb carrier confinement and facilitate trap-assisted recombination.

With respect to the compositional gradients observed in the QDs, we note that these gradients are unlikely to be the direct origin of the tail emission. Rather, the compositional gradients arise from the cation exchange-driven shell formation process, which can locally modulate the chemical environment and defect distribution within the QDs. In this sense, the compositional gradients may indirectly influence the spatial distribution and density of defect states, particularly near the core/shell and inner/outer shell regions, thereby contributing to the prominence of defect-related tail emission, without being its primary cause. We also attempted to further elucidate the nature of this emission using time-resolved photoluminescence (TR-PL) measurements. However, the long-wavelength feature could not be reliably decomposed into distinct lifetime components, indicating that it originates from a broad distribution of energetically and spatially heterogeneous trap states. Accordingly, the revised manuscript now explicitly discusses that the observed tail emission is consistent with defect-related recombination processes associated with the ZnS precursor step.

The manuscript has been revised as follows:

Please see Page #5,

“In addition to the dominant green emission, the $\text{AgIn}_x\text{Ga}_{1-x}\text{S}_2/\text{AgGaS}_2/\text{ZnS}$ QDs exhibit a long-wavelength emission tail in the 600–700 nm range (Fig. 1d). The area ratio of this long-wavelength emission, defined as the fraction of the total PL intensity in the long-wavelength region ($\lambda > \lambda_{\text{PL max}} + 50$ nm), increases from ~3.7% in $\text{AgIn}_x\text{Ga}_{1-x}\text{S}_2/\text{AgGaS}_2$ QDs to ~14.9% in $\text{AgIn}_x\text{Ga}_{1-x}\text{S}_2/\text{AgGaS}_2/\text{ZnS}$ QDs. Such long-wavelength emission is generally attributed to defect- or trap-assisted recombination rather than band-edge excitonic transitions in AgInS_2 -based QDs and related I–III–VI₂-based QDs.³⁶⁻⁴⁰”

Please see Page #6,

“X-ray diffraction (XRD) measurements were further performed on the core-only $\text{AgIn}_x\text{Ga}_{1-x}\text{S}_2$, $\text{AgIn}_x\text{Ga}_{1-x}\text{S}_2/\text{AgGaS}_2$, and $\text{AgIn}_x\text{Ga}_{1-x}\text{S}_2/\text{AgGaS}_2/\text{ZnS}$ QDs to corroborate the structural similarity before and after shell formation. All XRD patterns are consistent with a chalcopyrite-based tetragonal structure, and the overall diffraction features remain essentially unchanged, even after the Zn precursor step (Supplementary Fig. 4). This comparative analysis indicates that the crystal structure is preserved throughout the shell formation process.”

Please see Page #15,

“The emergence of the $\text{Zn}_{1-3/2x}\text{Ga}_x\text{S}$ outer shell via cation exchange offers a plausible origin for the long-wavelength emission tail observed after the ZnS precursor step. In our system, the substitution of Ag^+ and Ga^{3+} by Zn^{2+} during shell formation inherently introduces cation vacancies and local charge-compensating defects. Consistent with our APT observations, the inward diffusion of Zn^{2+} into the interior of the QDs further induces cation exchange and promotes the formation of such defects. When these defects form in the vicinity of In-rich regions, where carriers are preferentially confined, they can introduce intragap trap states that locally perturb the confinement potential and promote trap-assisted recombination. These defect-related states are commonly associated with donor–acceptor-pair-like recombination pathways that give rise to long-wavelength emission in I–III–VI₂ semiconductors.^{23,38-40,66-68}”

Please see Fig. 1,

“Fig. 1 Comparison of $\text{AgIn}_x\text{Ga}_{1-x}\text{S}_2/\text{AgGaS}_2$ QDs and $\text{AgIn}_x\text{Ga}_{1-x}\text{S}_2/\text{AgGaS}_2/\text{ZnS}$ QDs. a, d Schematic representation of the internal structure, band diagrams, and UV–vis absorption, and PL characteristics. Both QDs exhibit bright green luminescence at ~530 nm, with the QY increasing from 85% to 92% after the ZnS precursor step to form the outer shell. In addition, $\text{AgIn}_x\text{Ga}_{1-x}\text{S}_2/\text{AgGaS}_2/\text{ZnS}$ QDs exhibit a long-wavelength emission tail in the ~600–700 nm range.”

Please see Supplementary Fig. 4,

Supplementary Fig. 4 XRD patterns of $\text{AgIn}_x\text{Ga}_{1-x}\text{S}_2$ -based QDs. XRD patterns of $\text{AgIn}_x\text{Ga}_{1-x}\text{S}_2$ (AIGS), $\text{AgIn}_x\text{Ga}_{1-x}\text{S}_2/\text{AgGaS}_2$ (AIGS/AGS), and $\text{AgIn}_x\text{Ga}_{1-x}\text{S}_2/\text{AgGaS}_2/\text{ZnS}$ (AIGS/AGS/ZnS) QDs. All XRD patterns are consistent with a chalcopyrite-based tetragonal structure, with only minor peak shifts attributable to slight lattice parameter variations.

References have been added as follows:

23. Kim, J. H. et al. Synthesis of widely emission-tunable Ag–Ga–S and its quaternary derivative quantum dots. *Chem. Eng. J.* **347**, 791-797 (2018). [10.1016/j.cej.2018.04.167](https://doi.org/10.1016/j.cej.2018.04.167).
36. Dai, M. et al. Tunable photoluminescence from the visible to near-infrared wavelength region of non-stoichiometric AgInS_2 nanoparticles. *J. Mater. Chem.* **22**, 12851-12858 (2012). [10.1039/C2JM31463K](https://doi.org/10.1039/C2JM31463K).
37. Chen, Y. et al. Green and facile synthesis of high-quality water-soluble Ag-In-S/ZnS core/shell quantum dots with obvious bandgap and sub-bandgap excitations. *J. Alloys Compd.* **753**, 364-370 (2018). [10.1016/j.jallcom.2018.04.242](https://doi.org/10.1016/j.jallcom.2018.04.242).
38. Park, S. M. et al. Red Ag-based I–III–VI quantum dots as competitive alternatives to InP

emitters. *ACS Energy Lett.* **10**, 3005-3013 (2025). [10.1021/acsenerylett.5c00962](https://doi.org/10.1021/acsenerylett.5c00962).

39. Farid, A. et al. One-pot synthesis of luminescent Ag–Cu–Ga–S/ZnS quantum dots bridging the cyan gap for ultrahigh-color-rendering white-light-emitting diodes. *ACS Appl. Nano Mater.* **8**, 14703-14712 (2025). [10.1021/acsanm.5c02386](https://doi.org/10.1021/acsanm.5c02386).

40. Park, S. et al. Suppressing tail emission from AgIn_{1-x}Ga_xS₂/AgGaS₂ quantum dots by GaI₃-assisted interface reinforcement. *ACS Nano* **19**, 26831-26842 (2025). [10.1021/acsnano.5c07418](https://doi.org/10.1021/acsnano.5c07418).

66. Shen, F. et al. Photophysics and photovoltaic properties of Zn-alloyed Ag-In-S quantum dots sensitized solar cells. *J. Alloys Compd.* **922**, 166296 (2022). [10.1016/j.jallcom.2022.166296](https://doi.org/10.1016/j.jallcom.2022.166296).

67. Rivaux, C. et al. Continuous flow aqueous synthesis of highly luminescent AgInS₂ and AgInS₂/ZnS quantum dots. *J. Phys. Chem. C* **126**, 20524-20534 (2022). [10.1021/acs.jpcc.2c06849](https://doi.org/10.1021/acs.jpcc.2c06849).

68. Kameyama, T. et al. Controlling the electronic energy structure of ZnS–AgInS₂ solid solution nanocrystals for photoluminescence and photocatalytic hydrogen evolution. *J. Phys. Chem. C* **119**, 24740-24749 (2015). [10.1021/acs.jpcc.5b07994](https://doi.org/10.1021/acs.jpcc.5b07994).

Comment 3. In Figures 2c and 4c: is the iso-surface proxigram shown just for one quantum dot? How does it differ between the 5 QDs identified?

Answer to Comment 3: The iso-surface proxigrams shown in Figures 2c and 4c were extracted by averaging the data from all five quantum dots (QDs) identified in each APT dataset, rather than from a single QD. As shown in the individual 2D concentration profiles in Figures 3c and 3f, as well as Figures 5c and 5f, the absolute compositions vary slightly from one QD to another; however, the overall compositional trends and distribution profiles are highly consistent across the five QDs. Proxigrams extracted from a single QD can be relatively noisy due to limited ion counts, whereas averaging over multiple QDs improves statistical robustness and more clearly reveals the general reproducible compositional trends.

Because proxigram analysis is particularly useful for capturing and visualizing the compositional tendencies in nanoscale heterostructures, we included the proxigrams in Figures 2c and 4c to represent the common behavior among the analyzed QDs. This approach allows the key compositional features to be conveyed clearly while minimizing dot-to-dot variability. We have now explicitly clarified in the figure captions that the proxigrams represent averaged profiles obtained from five QDs.

The manuscript has been revised as follows:

Please see Fig. 2,

c Ag iso-surface map and proxigram showing 3D compositional trends; the proxigram was obtained from five QDs. The Ag and In contents increase toward the QD core, while the Ga content increases toward its outer surface, confirming the core/shell heterostructure.

Please see Fig. 4,

c Ag and Zn iso-surface and proxigram obtained from Ag iso-surfaces; the proxigram was obtained from five QDs. The Ag and In contents increase toward the QD core, while the content of Zn increases toward the outer shell of the QD. A significant amount of Ga also appears in the outer shell.

Comment 4. What happens if quantum dots aggregate? Is it possible to differentiate using the APT technique aggregated regions of QDs from individual qds that have clearly defined domains?

Answer to Comment 4: We thank the reviewer for raising this important question regarding quantum dot (QD) aggregation and the capability of APT to distinguish aggregated regions from individual QDs with well-defined domains.

In general, APT analysis of colloidal QDs is technically challenging, primarily because QDs are not embedded in a rigid matrix, which complicates specimen preparation for APT. To address this issue, we prepared APT specimens from QDs in a dried powder form. In this state, the organic ligands surrounding the QDs contract during drying, resulting in QDs being positioned in close proximity to one another. However, this proximity does not indicate irreversible aggregation, as the dried QDs readily disperse uniformly when re-dissolved in solvent.

As shown by the APT reconstructions, the QDs do not appear as a single aggregated mass within the reconstructed volume but remain distinguishable as individual particles with separable volumes. This allows us to perform volume extraction and analyze each QD individually. To further support this interpretation, we provide complementary transmission electron microscopy images of the QDs prepared in the same dried powder form, which show that individual QDs remain spatially distinguishable despite their close packing.

If the QDs were present as a single, fully merged structure, the APT data would not be expected to reveal distinct core/shell/shell architectures for individual particles, nor would ligand-derived C and H signals be observed separating adjacent QDs, as seen in our datasets. In that case, a single continuous compositional profile would be expected, rather than the discrete, particle-resolved structures observed here. Therefore, while QDs are closely packed in the dried powder state, APT enables us to differentiate individual QDs from aggregated regions within the present datasets by resolving distinct compositional domains and interparticle boundaries.

The manuscript has been revised as follows:

Please see Page #7,

“For the APT analysis in this work, dried QDs were prepared in powder form and analyzed following our previously described method.⁴¹ In that study, we demonstrated that APT could faithfully resolve core/shell architectures and 3D compositional distributions in commercially available CdSe/ZnS core/shell QDs, in good agreement with TEM observations.”

Please see Page #21,

“APT specimen preparation

Needle-shaped specimens for APT analysis were prepared from dried QDs in powder form using the focused ion beam (FIB) lift-out method (Helios5 HX, Thermo Fisher Scientific). Initially, a 100 nm-thick Pt layer was deposited using an electron beam, followed by depositing a 1 μm -thick Pt layer with a Ga-ion beam to passivate a region of interest measuring 12 μm \times 1.7 μm on the sample surface. The target region was then extracted and mounted onto a sharpened W tip. The specimens on the W tip were further shaped into a needle-like geometry using an annular milling pattern (30 kV, 80 pA). To refine the samples and reduce damage induced by the Ga-ion beam, the region of interest was thinned again with an annular pattern at 5 kV and 8 pA. The morphology and dispersion of the QDs prepared by this procedure were confirmed by HAADF-STEM (Supplementary Fig. 21).”

Please see Supplementary Fig. 21,

Supplementary Fig. 21 Morphology and dispersion of $\text{AgIn}_x\text{Ga}_{1-x}\text{S}_2/\text{AgGaS}_2/\text{ZnS}$ QDs in dried powder form. HAADF-STEM images of $\text{AgIn}_x\text{Ga}_{1-x}\text{S}_2/\text{AgGaS}_2/\text{ZnS}$ QDs in dried powder form,

prepared by FIB, showing the representative particle morphology and overall dispersion. The QDs exhibit sizes and shapes comparable to those observed in the solution-dispersed QDs. (Scale bars: 10 nm.)

References have been added as follows:

41. Chae, B. G. et al. Direct three-dimensional observation of core/shell-structured quantum dots with a composition-competitive gradient. *ACS Nano* **12**, 12109-12117 (2018). [10.1021/acsnano.8b05379](https://doi.org/10.1021/acsnano.8b05379).

Comment 5. For the core-shell-shell structure (with ZnS), is the first shell still Ag-deficient like with the core-shell structure (as mentioned in line 76)?

Answer to Comment 5: We thank the reviewer for the question regarding whether the first shell remains Ag-deficient after the ZnS precursor step.

Based on our APT results, the first shell remains Ag-deficient even after the ZnS precursor step. As shown in the 2D concentration profiles in Figures 5c and 5f, the region immediately inside the Zn-based outer shell, identified as the $Zn_{1-3/2x}Ga_xS$ shell, exhibits a relatively higher Ga concentration and a correspondingly lower Ag concentration. This compositional trend is consistent with that observed for the Ag-deficient shell in the core/shell structure prior to the ZnS precursor step, as shown in Figures 3c and 3f. These observations indicate that the Ag-deficient nature of the first shell is preserved during the subsequent Zn-based shelling process, rather than being eliminated or homogenized. Accordingly, the final core/shell/shell architecture consists of an $AgIn_xGa_{1-x}S_2$ core, an $AgGaS_2$ inner shell, an Ag-deficient $AgGaS_2$ layer, and a $Zn_{1-3/2x}Ga_xS$ outer shell.

Comment 6. Since the nanocrystal size does not change after ZnS shelling, does this mean the core-size decreases after addition of the ZnS shell? Or does the thickness of the first shell ($AgGaS_2$ shell) decrease instead? Is it possible to estimate the diameter of the core and the thickness of two shells using this technique?

Answer to Comment 6: We thank the reviewer for the question regarding whether the unchanged nanocrystal size after the ZnS precursor step implies a reduction in the core size or a change in the thickness of the first shell, and whether the core and shell thicknesses can be estimated using APT.

Based on our transmission electron microscopy (TEM) and APT results, the $AgIn_xGa_{1-x}S_2$ core does not decrease in size after the ZnS precursor step. Instead, a portion of the $AgGaS_2$ inner shell is converted into the $Zn_{1-3/2x}Ga_xS$ outer shell through a cation exchange process, rather than through epitaxial growth of a new ZnS layer. Consequently, the overall nanocrystal diameter remains nearly unchanged, while a $Zn_{1-3/2x}Ga_xS$ outer shell forms at the expense of the part of the $AgGaS_2$ inner shell. To further corroborate

this interpretation, we additionally performed XRD measurements in this revision. The core-only $\text{AgIn}_x\text{Ga}_{1-x}\text{S}_2$, $\text{AgIn}_x\text{Ga}_{1-x}\text{S}_2/\text{AgGaS}_2$, and $\text{AgIn}_x\text{Ga}_{1-x}\text{S}_2/\text{AgGaS}_2/\text{ZnS}$ quantum dots (QDs) all exhibit diffraction patterns consistent with a chalcopyrite-based tetragonal structure, and the overall diffraction features remain essentially unchanged after the Zn precursor step, aside from minor peak shifts. These XRD results further support that the overall crystallographic framework is preserved during the ZnS precursor step.

Regarding dimensional analysis, APT is well suited for resolving 3D compositional distributions and identifying compositional gradients and interfaces within individual QDs. However, determining the absolute thickness of each layer with high precision using APT alone remains challenging. This limitation arises because different materials exhibit different evaporation fields, which can introduce local magnification effects and distort reconstructed dimensions. Despite these effects, numerous studies have demonstrated that APT provides reliable information on compositional distributions in nanostructures, particularly along the analysis (z) direction. Therefore, while APT reliably captures the relative compositional trends and the core/shell/shell architecture, precise and absolute measurements of the core diameter and individual shell thicknesses remain subject to uncertainty. In principle, TEM can provide complementary size information; however, because TEM images represent projected views through the nanocrystal, accurately determining the thickness of individual shell layers remains inherently challenging.

The manuscript has been revised as follows:

Please see Page #8,

“The QDs appear as slightly ellipsoidal features in the APT reconstruction, and a slice-view image along the x–z plane is presented in Fig. 2b to visualize their internal elemental distribution. We note that APT analyses of nanostructures may exhibit reconstruction artifacts, such as apparent elongation along the analysis (z) direction and interfacial broadening, primarily arising from local magnification and trajectory aberrations.⁴²⁻⁵² Accordingly, the slightly ellipsoidal morphology observed here is attributed to such effects rather than the intrinsic particle shape. However, compositional distributions along the z-direction have been shown to provide more reliable quantitative information than lateral directions because the reduced trajectory overlap suppresses artificial intermixing.^{42-44,51,52} Under these conditions, compositional distributions and interfacial characteristics closer to the intrinsic distributions can be more faithfully evaluated.”

References have been added as follows:

42. Jang, K. et al. Three-dimensional atomic mapping of ligands on palladium nanoparticles by atom probe tomography. *Nat. Commun.* **12**, 4301 (2021). [10.1038/s41467-021-24620-9](https://doi.org/10.1038/s41467-021-24620-9).
43. Kim, S. H. et al. Characterization of Pd and Pd@ Au core-shell nanoparticles using atom probe

tomography and field evaporation simulation. *J. Alloys Compd.* **831**, 154721 (2020). [10.1016/j.jallcom.2020.154721](https://doi.org/10.1016/j.jallcom.2020.154721).

44. Tedsree, K. et al. Hydrogen production from formic acid decomposition at room temperature using a Ag–Pd core–shell nanocatalyst. *Nat. Nanotechnol.* **6**, 302-307 (2011). [10.1038/nnano.2011.42](https://doi.org/10.1038/nnano.2011.42).

45. Grenier, A. et al. 3D analysis of advanced nano-devices using electron and atom probe tomography. *Ultramicroscopy* **136**, 185-192 (2014). [10.1016/j.ultramic.2013.10.001](https://doi.org/10.1016/j.ultramic.2013.10.001).

46. Khan, M. A., Ringer, S. P. & Zheng, R. Atom probe tomography on semiconductor devices. *Adv. Mater. Interfaces* **3**, 1500713 (2016). [10.1002/admi.201500713](https://doi.org/10.1002/admi.201500713).

47. Hatzoglou, C., Radiguet, B. & Pareige, P. Experimental artefacts occurring during atom probe tomography analysis of oxide nanoparticles in metallic matrix: quantification and correction. *J. Nucl. Mater.* **492**, 279-291 (2017). [10.1016/j.jnucmat.2017.05.008](https://doi.org/10.1016/j.jnucmat.2017.05.008).

48. Philippe, T., Gruber, M., Vurpillot, F. & Blavette, D. Clustering and local magnification effects in atom probe tomography: a statistical approach. *Microsc. Microanal.* **16**, 643-648 (2010). [10.1017/S1431927610000449](https://doi.org/10.1017/S1431927610000449).

49. Lawitzki, R., Stender, P. & Schmitz, G. Compensating local magnifications in atom probe tomography for accurate analysis of nano-sized precipitates. *Microsc. Microanal.* **27**, 1–12 (2021). [10.1017/S1431927621000180](https://doi.org/10.1017/S1431927621000180).

50. Beinke, D., Oberdorfer, C. & Schmitz, G. Towards an accurate volume reconstruction in atom probe tomography. *Ultramicroscopy* **165**, 34-41 (2016). [10.1016/j.ultramic.2016.03.008](https://doi.org/10.1016/j.ultramic.2016.03.008).

51. Takahashi, J. & Kawakami, K. Position artifacts in 3D reconstruction of plate-shaped precipitates in steels depending on the analysis direction of atom probe tomography. *Surf. Interface Anal.* **53**, 982-995 (2021). [10.1002/sia.7001](https://doi.org/10.1002/sia.7001).

52. Maruyama, N., Smith, G. D. W. & Cerezo, A. Interaction of the solute niobium or molybdenum with grain boundaries in α -iron. *Mater. Sci. Eng. A* **353**, 126-132 (2003). [10.1016/S0921-](https://doi.org/10.1016/S0921-)

Comment 7. As mentioned in line 235 to 245, why was the compositional inhomogeneity more pronounced in the core versus the shell for the samples presented? Also, why does this inhomogeneity get more noticeable/severe after shelling with ZnS?

Answer to Comment 7: We thank the reviewer for the insightful question regarding the origin of the more pronounced compositional inhomogeneity observed in the core compared to the shell, and its apparent enhancement after the ZnS precursor step.

The relatively stronger compositional inhomogeneity observed in the core is primarily attributed to the intrinsic complexity of quaternary Ag–In–Ga–S core formation. During core synthesis, multiple precursors with different chemical reactivities are simultaneously present, and the formation of the core does not follow a simple, single-step nucleation-and-growth process. In particular, the $\text{AgIn}_x\text{Ga}_{1-x}\text{S}_2$ core is synthesized by heating a mixed precursor solution from low temperature, a process during which local clustering and compositionally heterogeneous regions are likely to form due to differences in precursor reactivity and diffusion kinetics. Previous studies have shown that $\text{AgIn}_x\text{Ga}_{1-x}\text{S}_2$ core formation does not proceed via a simple nucleation-and-growth mechanism, but rather involves intermediate phases and cation exchange processes. For example, several reports indicate that AgGaS_2 or AgInS_2 seeds form initially, followed by the incorporation of In and Ga through cation exchange to generate the chalcopyrite $\text{AgIn}_x\text{Ga}_{1-x}\text{S}_2$ lattice. This multistep and dynamically perturbed growth pathway is therefore likely to increase the likelihood of local compositional fluctuations within the core. Such effects are intrinsically more pronounced in quaternary systems than in binary or ternary quantum dots (QDs), explaining why compositional inhomogeneity is stronger in the core than in the subsequently formed shells. In addition, the ZnS precursor step involves an additional thermal treatment of the pre-formed cores, which can further promote cation diffusion and redistribution within the QDs, thereby amplifying pre-existing compositional fluctuations to some extent.

Another contributing factor to the observed compositional inhomogeneity, particularly when comparing multiple QDs, is inherent particle-to-particle variability. During quaternary QDs formation, individual QDs tend to exhibit a broader distribution of compositions compared to simpler systems. As supporting evidence, we present complementary STEM–EDS compositional profiles. Although STEM–EDS does not provide 3D information in this experiment, it enables a rough assessment of compositional distributions across a larger number of QDs. We analyzed the elemental distributions of six individual QDs from both the $\text{AgIn}_x\text{Ga}_{1-x}\text{S}_2/\text{AgGaS}_2$ and $\text{AgIn}_x\text{Ga}_{1-x}\text{S}_2/\text{AgGaS}_2/\text{ZnS}$ samples. While the overall compositional tendencies, including the core/shell and core/inner-shell/outer-shell architectures, are consistent, the relative compositions of the constituent elements vary from particle to particle. These results confirm clear particle-to-particle compositional variations, which are consistent with the trends observed by APT. These points have been addressed and discussed in the revised manuscript and the Supplementary Information.

The manuscript has been revised as follows:

Please see Page #14,

“Interestingly, directly comparing two individual QDs reveals compositional differences, even among

co-synthesized QDs. These differences are particularly pronounced in the core region, with noticeable variations in the Ag, In, and Ga contents; some QDs exhibit cores with a very low Ga content (Figs. 4d and 5c). Similar results were observed in the $\text{AgIn}_x\text{Ga}_{1-x}\text{S}_2/\text{AgGaS}_2$ QDs, albeit to a lesser extent. The pronounced inhomogeneity in the cores likely originates from the intrinsic complexity of the quaternary Ag–In–Ga–S system. The synthesis of the core simultaneously involves multiple cation precursors with different reactivities, and the cores are formed under a continuous heating process. Previous studies have also shown that $\text{AgIn}_x\text{Ga}_{1-x}\text{S}_2$ core formation proceeds through intermediate phases and cation-exchange processes rather than a simple single-step growth process.^{18,53,59} More specifically, AgGaS_2 or AgInS_2 seeds form initially, followed by the incorporation of In and Ga to generate the chalcopyrite $\text{AgIn}_x\text{Ga}_{1-x}\text{S}_2$ lattice, which is likely to promote local compositional fluctuations within the core. Furthermore, the subsequent ZnS precursor step introduces an additional thermal treatment, which can promote cation diffusion and redistribution, thereby amplifying pre-existing compositional fluctuations within the cores.”

Please see Page #14.

“Consistent with the APT observations, STEM–EDS line profiles collected from multiple QDs reveal noticeable particle-to-particle variations in the relative Ag, In, and Ga distributions (Supplementary Figs. 16 and 17). For each QD, several independent line profiles were extracted and compared, all of which show dot-to-dot compositional fluctuations. These results indicate that the observed compositional inhomogeneity is an intrinsic feature of the synthesized quaternary QDs, thereby supporting the features revealed by APT. A deviation from the ideal stoichiometry in the core can change the emission wavelengths. Thus, such compositional inhomogeneity in the quaternary semiconductor cores may contribute to the inhomogeneous spectral broadening of the UV–Vis absorption and PL bands for both QDs, along with other factors such as the size distribution. Our APT analysis directly shows compositional inhomogeneity among the cores and highlights the importance of controlling such variations, along with defects, to improve the color purity of the green emission.”

Please see Supplementary Fig. 16.

Supplementary Fig. 16 Detailed elemental distributions of $\text{AgIn}_x\text{Ga}_{1-x}\text{S}_2/\text{AgGaS}_2$ QDs measured by STEM-EDS. STEM image and corresponding EDS line profiles of individual $\text{AgIn}_x\text{Ga}_{1-x}\text{S}_2/\text{AgGaS}_2$ QDs. Although the QDs show similar compositional trends, the elemental compositions vary from particle to particle. Owing to the projected nature and limited spatial resolution of STEM-EDS, these data provide qualitative compositional trends rather than a definitive 3D core/shell/shell assignment. (Scale bars: 7 nm.)

Please see Supplementary Fig. 17,

Supplementary Fig. 17 Detailed elemental distributions of $\text{AgIn}_x\text{Ga}_{1-x}\text{S}_2/\text{AgGaS}_2/\text{ZnS}$ QDs measured by STEM–EDS. STEM image and corresponding EDS line profiles of individual $\text{AgIn}_x\text{Ga}_{1-x}\text{S}_2/\text{AgGaS}_2/\text{ZnS}$ QDs. Zn is distributed over a broader region than Ag and In, indicating the formation of a Zn-containing outer region. Although the QDs show similar overall compositional tendencies, the elemental compositions vary among individual QDs, indicating particle-to-particle variability. Owing to the projected nature and limited spatial resolution of STEM–EDS, these data provide qualitative compositional trends rather than a definitive 3D core/shell/shell assignment. (Scale bars: 7 nm.)

References have been added as follows:

18. Lee, H. J. et al. Coherent heteroepitaxial growth of I-III-VI₂ Ag(In,Ga)S₂ colloidal nanocrystals with near-unity quantum yield for use in luminescent solar concentrators. *Nat. Commun.* **14**, 3779 (2023). [10.1038/s41467-023-39509-y](https://doi.org/10.1038/s41467-023-39509-y).
53. Uematsu, T., Tepakidarekul, M., Hirano, T., Torimoto, T. & Kuwabata, S. Facile high-yield synthesis of Ag–In–Ga–S quaternary quantum dots and coating with gallium sulfide shells for narrow

band-edge emission. *Chem. Mater.* **35**, 1094-1106 (2023). [10.1021/acs.chemmater.2c03023](https://doi.org/10.1021/acs.chemmater.2c03023).

59. Xie, X. et al. Narrow-bandwidth I–III–VI semiconductor nanocrystals: synthesis, luminescence and applications in quantum-dot light-emitting diodes. *Adv. Phys. Res.* **3**, 2400071 (2024). [10.1002/apxr.202400071](https://doi.org/10.1002/apxr.202400071).

Comment 8. X-ray diffraction pattern not included. Can the XRD of the QDs before and after ZnS shelling be included as supporting information? This would further corroborate the high-resolution STEM imaging showing that the particles are structurally similar before and after coating with ZnS (lines 114 to 119).

Answer to Comment 8: We thank the reviewer for this valuable suggestion. To further corroborate the structural similarity of the quantum dots (QDs) before and after ZnS shelling, we have performed X-ray diffraction (XRD) analysis on the core-only $\text{AgIn}_x\text{Ga}_{1-x}\text{S}_2$ samples, the $\text{AgIn}_x\text{Ga}_{1-x}\text{S}_2/\text{AgGaS}_2$ samples, and the $\text{AgIn}_x\text{Ga}_{1-x}\text{S}_2/\text{AgGaS}_2/\text{ZnS}$ samples.

All samples exhibit diffraction patterns consistent with a chalcopyrite-based tetragonal structure. Importantly, the overall diffraction features remain largely unchanged after the ZnS precursor step, except for minor peak shifts attributable to slight lattice parameter variations associated with shell formation. Although the diffraction peaks are broadened due to the nanocrystalline nature of the QDs, the comparative analysis indicates that the overall crystal structure is preserved throughout the shell formation process. The corresponding XRD patterns have now been included in the Supplementary Information, and the revised manuscript has been updated to reflect this additional structural characterization in the Supplementary Information.

The manuscript has been revised as follows:

Please see Page #6.

“X-ray diffraction (XRD) measurements were further performed on the core-only $\text{AgIn}_x\text{Ga}_{1-x}\text{S}_2$, $\text{AgIn}_x\text{Ga}_{1-x}\text{S}_2/\text{AgGaS}_2$, and $\text{AgIn}_x\text{Ga}_{1-x}\text{S}_2/\text{AgGaS}_2/\text{ZnS}$ QDs to corroborate the structural similarity before and after shell formation. All XRD patterns are consistent with a chalcopyrite-based tetragonal structure, and the overall diffraction features remain essentially unchanged, even after the Zn precursor step (Supplementary Fig. 4). This comparative analysis indicates that the crystal structure is preserved throughout the shell formation process.”

Please see Supplementary Fig. 4.

Supplementary Fig. 4 XRD patterns of $\text{AgIn}_x\text{Ga}_{1-x}\text{S}_2$ -based QDs. XRD patterns of $\text{AgIn}_x\text{Ga}_{1-x}\text{S}_2$ (AIGS), $\text{AgIn}_x\text{Ga}_{1-x}\text{S}_2/\text{AgGaS}_2$ (AIGS/AGS), and $\text{AgIn}_x\text{Ga}_{1-x}\text{S}_2/\text{AgGaS}_2/\text{ZnS}$ (AIGS/AGS/ZnS) QDs. All XRD patterns are consistent with a chalcopyrite-based tetragonal structure, with only minor peak shifts attributable to slight lattice parameter variations.

Comment 9. Do the authors have elemental analysis (such as ICP-OES or ICP-MS) of the sample to compare/confirm the elemental composition obtained from atomic probe tomography (APT)?

Answer to Comment 9: We thank the reviewer for the suggestion to include elemental analysis to validate the compositional results obtained from APT.

We have performed inductively coupled plasma optical emission spectroscopy (ICP-OES) measurements on the samples. It should be noted that the ICP-OES results represent bulk-averaged elemental compositions rather than information from individual quantum dots. In contrast, APT enables the extraction of compositional information from the reconstructed analysis volume, as well as from selected local regions. Despite this fundamental difference in sampling volume, we find that the overall compositional trends obtained from ICP-OES are consistent with those derived from APT. To facilitate direct comparison and to improve transparency, the ICP-OES results have been added to the Supplementary Information.

The manuscript has been revised as follows:

Please see Page #13,

“Because APT inherently probes only a limited number of individual QDs and may therefore suffer from limited statistical representativeness, the compositional features identified by APT were further examined using complementary techniques. XPS analyses performed on the same samples reveal consistently higher Ga-to-Ag ratios after the Zn precursor step, evaluated from multiple core-level regions, including Ag *MNN* vs Ga *2p* and Ag *4d* vs Ga *3d*. This systematic increase in the Ga-to-Ag ratios supports the formation of Ga-containing outer shells and is consistent with the $Zn_{1-3/2x}Ga_xS$ shell formation revealed by APT (Supplementary Fig. 15 and Supplementary Table 2). Inductively coupled plasma analyses performed on the same samples show overall elemental compositions consistent with the APT results (Supplementary Tables 3 and 4). Taken together, these results indicate that the structural and compositional features identified by APT are not limited to a small number of individual QDs but are representative of overall sample.”

Please see Supplementary Table 3,

Specimen	Atomic ratio (ICP–OES)			
	Ag/S	In/S	Ga/S	Zn/S
AIGS/AGS	0.40	0.12	0.44	0
AIGS/AGS/ZnS	0.13	0.08	0.35	0.46

Supplementary Table 3. Atomic ratios measured of $AgIn_xGa_{1-x}S_2/AgGaS_2$ (AIGS/AGS) and $AgIn_xGa_{1-x}S_2/AgGaS_2/ZnS$ (AIGS/AGS/ZnS) determined by ICP–OES.

Please see Supplementary Table 4,

Specimen	Atomic ratio (APT)			
	Ag/S	In/S	Ga/S	Zn/S

AIGS/AGS	0.43	0.13	0.39	0
AIGS/AGS/ZnS	0.15	0.08	0.30	0.48

Supplementary Table 4. Atomic ratios of $\text{AgIn}_x\text{Ga}_{1-x}\text{S}_2$ / AgGaS_2 (AIGS/AGS) and $\text{AgIn}_x\text{Ga}_{1-x}\text{S}_2$ / AgGaS_2 / ZnS (AIGS/AGS/ZnS) determined by APT.

Comment 10. Method section (line 330) indicates that core-only AgInGaS were synthesized as well. How does the size, morphology, crystal structure and elemental composition between the core-only and core-shell (and core-shell-shell) nanocrystals differ? Was APT conducted on the core-only QDs? If so, are the elemental composition trends seen for the core-only, similar to the cores of the shelled QDs? Is inhomogeneity between different nanocrystals seen also for the core-only samples? Or is it a by-product of shelling?

Answer to Comment 10: We thank the reviewer for the detailed questions regarding the size, morphology, crystal structure, and elemental composition of the core-only $\text{AgIn}_x\text{Ga}_{1-x}\text{S}_2$ quantum dots (QDs), and whether APT analysis was performed on these samples.

APT analysis of colloidal QDs is technically very challenging, and this limitation becomes particularly significant in the case of core-only QDs. Because of their small size and the absence of a rigid embedding matrix, specimen preparation for APT is highly nontrivial. More importantly, reliable APT analysis of individual QDs requires sufficient compositional contrast to distinguish neighboring particles within the reconstructed volume. In systems such as $\text{AgIn}_x\text{Ga}_{1-x}\text{S}_2$ / AgGaS_2 or $\text{AgIn}_x\text{Ga}_{1-x}\text{S}_2$ / AgGaS_2 / ZnS , the presence of chemically distinct shell layers provides such contrast, enabling the identification and extraction of individual QDs for analysis. In contrast, for core-only $\text{AgIn}_x\text{Ga}_{1-x}\text{S}_2$ QDs, which consist of a single composition without a surrounding shell or matrix, differentiating adjacent particles in APT reconstructions becomes extremely difficult. This situation is similar to analyzing nanoparticles without a surrounding matrix in APT, where particle segmentation becomes unreliable and subsequent compositional interpretation can be significantly reduced. This limitation represents one of the major challenges in applying APT to colloidal QDs and is one of the key reasons why APT studies on colloidal QDs remain scarce in the literature to date.

Nevertheless, the size, size distribution, crystal structure, and overall elemental composition of the core-only $\text{AgIn}_x\text{Ga}_{1-x}\text{S}_2$ QDs were thoroughly characterized using complementary techniques. X-ray diffraction (XRD) analysis clearly shows that the core-only $\text{AgIn}_x\text{Ga}_{1-x}\text{S}_2$ QDs crystallize in a chalcopyrite-based tetragonal structure, which is also preserved after AgGaS_2 and ZnS shelling processes. Transmission electron microscopy further confirms that the core-only QDs exhibit a round-cornered polyhedral or spherical shape with an average diameter of 3.58 ± 0.59 nm and clear lattice fringes, indicating high crystallinity consistent with the XRD results.

We suggest that the compositional inhomogeneity discussed in this work is not solely a by-product of shelling, but rather largely reflects intrinsic characteristics of quaternary core formation. This

interpretation is consistent with the intrinsically complex formation mechanism of quaternary $\text{AgIn}_x\text{Ga}_{1-x}\text{S}_2$ cores, in which multiple precursors with different reactivities participate in a non-simple, multistep growth process involving intermediate phases and cation exchange reactions. Such dynamically perturbed core formation pathways are well known to promote local compositional fluctuations in quaternary I–III–VI₂ QDs. The shelling process itself may modify local compositional distributions during shell formation, and the observed compositional inhomogeneity may also reflect inherent particle-to-particle variability, as discussed in our response to Comment #7. Accordingly, the compositional features revealed by APT are interpreted as arising from a combination of intrinsic heterogeneity established during quaternary core formation and additional structural and compositional modifications during shell synthesis. These features are therefore not interpreted as being solely induced by shelling, nor as artifacts of APT visualization. The additional results have been incorporated into the revised manuscript and the Supporting Information.

The manuscript has been revised as follows:

Please see Page #6.

“Interestingly, after the ZnS precursor step for the outer shell formation, the QDs grow only negligibly, despite the addition of more precursors, and remain similar in both size and shape, with an average diameter of 6.11 ± 1.03 nm (Supplementary Fig. 2; see also Supplementary Fig. 3 for HAADF–STEM images of the core-only $\text{AgIn}_x\text{Ga}_{1-x}\text{S}_2$ QDs). Moreover, the high-resolution STEM results reveal that the QDs are structurally similar before and after the ZnS precursor step, with all crystal planes showing a lattice spacing of ~ 0.29 nm, which almost corresponds to the interplanar spacing of the (200) planes of chalcopyrite-structured AgGaS_2 .”

Please see Page #6.

“X-ray diffraction (XRD) measurements were further performed on the core-only $\text{AgIn}_x\text{Ga}_{1-x}\text{S}_2$, $\text{AgIn}_x\text{Ga}_{1-x}\text{S}_2/\text{AgGaS}_2$, and $\text{AgIn}_x\text{Ga}_{1-x}\text{S}_2/\text{AgGaS}_2/\text{ZnS}$ QDs to corroborate the structural similarity before and after shell formation. All XRD patterns are consistent with a chalcopyrite-based tetragonal structure, and the overall diffraction features remain essentially unchanged, even after the Zn precursor step (Supplementary Fig. 4). This comparative analysis indicates that the crystal structure is preserved throughout the shell formation process.”

Please see Page #14.

“Previous studies have also shown that $\text{AgIn}_x\text{Ga}_{1-x}\text{S}_2$ core formation proceeds through intermediate phases and cation-exchange processes rather than a simple single-step growth process.^{18,53,59} More specifically, AgGaS_2 or AgInS_2 seeds form initially, followed by the incorporation of In and Ga to

generate the chalcopyrite $\text{AgIn}_x\text{Ga}_{1-x}\text{S}_2$ lattice, which is likely to promote local compositional fluctuations within the core. Furthermore, the subsequent ZnS precursor step introduces an additional thermal treatment, which can promote cation diffusion and redistribution, thereby amplifying pre-existing compositional fluctuations within the cores.”

Please see Supplementary Fig. 3.

Supplementary Fig. 3 HAADF–STEM image of core-only $\text{AgIn}_x\text{Ga}_{1-x}\text{S}_2$ QDs. a Low-magnification and **b** high-magnification HAADF–STEM images of core-only $\text{AgIn}_x\text{Ga}_{1-x}\text{S}_2$ QDs. The $\text{AgIn}_x\text{Ga}_{1-x}\text{S}_2$ QDs are crystalline and exhibit an average diameter of 3.58 ± 0.59 nm. (Scale bars: 20 nm in **a** and 5nm in **b**.)

Please see Supplementary Fig. 4.

Supplementary Fig. 4 XRD patterns of $\text{AgIn}_x\text{Ga}_{1-x}\text{S}_2$ -based QDs. XRD patterns of $\text{AgIn}_x\text{Ga}_{1-x}\text{S}_2$ (AIGS), $\text{AgIn}_x\text{Ga}_{1-x}\text{S}_2/\text{AgGaS}_2$ (AIGS/AGS), and $\text{AgIn}_x\text{Ga}_{1-x}\text{S}_2/\text{AgGaS}_2/\text{ZnS}$ (AIGS/AGS/ZnS) QDs. All XRD patterns are consistent with a chalcopyrite-based tetragonal structure, with only minor peak shifts attributable to slight lattice parameter variations.

References have been added as follows:

18. Lee, H. J. et al. Coherent heteroepitaxial growth of I-III-VI₂ Ag(In,Ga)S₂ colloidal nanocrystals with near-unity quantum yield for use in luminescent solar concentrators. *Nat. Commun.* **14**, 3779 (2023). [10.1038/s41467-023-39509-y](https://doi.org/10.1038/s41467-023-39509-y).
53. Uematsu, T., Tepakidarekul, M., Hirano, T., Torimoto, T. & Kuwabata, S. Facile high-yield synthesis of Ag–In–Ga–S quaternary quantum dots and coating with gallium sulfide shells for narrow band-edge emission. *Chem. Mater.* **35**, 1094-1106 (2023). [10.1021/acs.chemmater.2c03023](https://doi.org/10.1021/acs.chemmater.2c03023).
59. Xie, X. et al. Narrow-bandwidth I–III–VI semiconductor nanocrystals: synthesis, luminescence and applications in quantum-dot light-emitting diodes. *Adv. Phys. Res.* **3**, 2400071 (2024). [10.1002/apxr.202400071](https://doi.org/10.1002/apxr.202400071).